# Co-infection mathematical model for HIV/ AIDS and tuberculosis with optimal control in Ethiopia

**Tigabu Kasie Ayele** [1,2]*, **Emile Franc Doungmo Goufo**[2], **Stella Mugisha**[2]

**1** Department of Mathematics, College of Natural and Applied Science, Addis Ababa Science and Technology University, Addis Ababa, Ethiopia, **2** Department of Mathematical Sciences, College of Science, Engineering and Technology, University of South Africa, South Africa

* tigabu.kassa@aastu.edu.et, 67122884@mylife.unisa.ac.za

**Data Availability Statement:** All relevant data are within the manuscript.

**Funding:** The author(s) received no specific funding for this work.

## Abstract

The co-epidemics of HIV/AIDS and Tuberculosis (TB) outbreak is one of a serious disease in Ethiopia that demands integrative approaches to combat its transmission. In contrast, epidemiological co-infection models often considered a single latent case and recovered individuals with TB. To bridge this gap, we presented a new optimal HIV-TB co-infection model that considers both high risk and low risk latent TB cases with taking into account preventive efforts of both HIV and TB diseases, case finding for TB and HIV/AIDS treatment. This study aimed to develop optimal HIV/AIDS-TB co-infection mathematical model to explore the best cost-effective measure to mitigate the disease burden. The model is analysed analytically by firstly segregating TB and HIV only sub models followed by the full TB-HIV co-infection model. The Disease Free Equilibrium (DFE) and Endemic Equilibrium (EE) points are found and the basic reproduction number $R_0$ is obtained using the next generation matrix method (NGM). Based on the threshold value $R_0$, the stabilities of equilibria for each sub-model are analysed. The DFE point is locally asymptotically stable when $R_0 < 1$ and unstable when $R_0 > 1$. The EE point is also asymptotically stable when $R_0 > 1$ and does not exist otherwise. At $R_0 = 1$, the existence of backward bifurcation phenomena is discussed. To curtail the cost and disease fatality, an optimal control model is formulated via time based controlling efforts. The optimal mathematical model is analysed both analytically and numerically. The numerical results are presented for two or more control measures at a time. In addition, the Incremental Cost-Effectiveness Ratio(ICER) has identified the best strategy which is crucial in limited resource. Hence, the model outcomes illustrated that applying HIV/AIDS prevention efforts and TB case finding concurrently is the most cost-effective strategy to offer substantial relief from the burden of the pandemic in the community. All results found in this study have significant public health lessons. We anticipated that the results will notify evidence based approaches to control the disease. Thus, this study will aids in the fight against HIV/AIDS, TB, and their co-infection policy-makers and other concerned organizations.

**Competing interests:** The authors have declared that no competing interests exist.

# 1 Introduction

The two infectious agents HIV and Mycobacterium tuberculosis (Mtb) have been co-existing in humans for decades. We have enough evidence that one of these infections accelerates the progression of the other [1–4]. Thus, we have to pay a great attention about the subject towards exploring HIV-TB co-epidemics.

Tuberculosis (TB) is an infectious disease caused by a bacterium called Mtb which can be spread from person to person. It affects the lungs and is called Pulmonary TB but can also affect other parts of the body like brain, glands, kidney and bones is then called Extra-Pulmonary TB [5, 6]. It is a curable disease, but may cause death unless infectious individuals are treating at the right time. The symptoms are depending on the body parts wherever the TB bacteria grow. The symptoms are 3 weeks cough or longer, chest pain, night sweat, and weight loss [7]. The TB bacteria spreads in to the air when TB infectious people sneeze, cough, speak or sing [8–10].

Human Immunodeficiency Virus (HIV) is a lenti-virus responsible for the Acquired Immunodeficiency Syndrome (AIDS) [11]. HIV is transmitting by infected blood, semen, vaginal secretion, and breastfeeding (for infants) from HIV+ mother without treatment. This virus affects white blood cells of the immune system by diminishing the quantity. Gradually, it destroys their functions and the immune system cannot resist other opportunistic infections. This leads HIV progression to the AIDS stage. AIDS is the supreme disease phase of HIV infection [11, 12]. The symptoms are tiredness, fever, night sweats, muscle aches, rapid weight loss, sore throat, etc. . . [7, 13]. Nowadays, there is no cure and vaccine for HIV, but the proper treatment with Antiretroviral Therapy (ART) individuals can control the virus to prolong their life [14].

The two pathogens Mtb and HIV may come either by co-infection or super-infection. One can increase the effect of the other and they can accelerate the deterioration of immune system function. People living with HIV are 20 up to 30 times at a higher risk of developing active TB disease than their counterparts [15, 16]. The prevalence rate of TB in HIV+ people is increasing because of exogenous re-infection and endogenous re-activation [17]. Individuals infected by HIV lead to a haggled immune system, consequently their susceptibility to Mtb infection is increasing. It is difficult to diagnose TB in HIV infected people [18]. TB also remains the primary agent of death in HIV infected people, counting for around 1 in 3 deaths associated with AIDS [19].

Globally, there are approximately 16 million people infected by HIV-TB co-infection in 2017 [20]. Recently, WHO 2020 report showed that an estimated number of over 14 million people are co-infected [21]. The report stated that South-East-Asia and Sub-Saharan Africa were taking the highest portion. More than 90% of TB deaths occurred in the two regions [22]. TB is also the highest cause of death from TB–AIDS related deaths, of which 95% happened in developing nations [23, 24].

The burden of TB-HIV co-infection is high in Sub-Saharan Africa (SSA) [25]. As stated by WHO report of 2019, about 84% of the total number of TB-HIV co-infection cases occurred in the region. SSA has 12% of the global population, it has accounted for 30% of the 9 million TB incidence cases and more than 270 000 deaths related to TB [26]. In this region, the HIV prevalence is high which grips to over 50% of the patients were dually infected.

As one of the Sub-Saharan Africa countries, Ethiopia is severely affected by the TB-HIV co-epidemic. As of 2019, an incidence rate of TB was 345 per 100 000 of which 33% are living with HIV [27]. Moreover, the WHO 2020 report stated that Ethiopia is one of the high TB and TB-HIV burden countries in the globe [28]. In 2019, over 23% of people are HIV+ from active

TB individuals in Ethiopia [27]. Co-infection of TB with HIV accelerates the possibility of progressing from latent to active stage [29].

HIV and TB have different nature and diverse treatment outcomes. Thus, an integrative treatment program is urgently needs for TB-HIV co-infection [1, 30]. TB can be cured with appropriate treatment for a period of six up to nine months. Mostly, the recommendations for people infected by this co-infection disease is to begin TB treatment immediately. Hereafter, they can start ART treatment on the appropriate starting time suggested by physicians. Initiating ART at or afterwards the start of TB therapy may cause Immune Reconstitution Inflammatory Syndrome (IRIS) [18]. This happens when a high pill burden of antibiotics and ART are existing. If IRIS occurs, it will worse TB infection. This leads TB treatment can be complicated. And also, delaying ART until after TB therapy is completed may increase HIV transmission risk and death caused by HIV. Hence, it is serious to differentiate the actual time where dual treatment is given for HIV-TB infected people.

Mathematical models are essential to explore the co-epidemics disease dynamics and to provide better insights about preventive and controlling regimes. Existing models on HIV–TB co-infection are reviewed in the following way.

Navjot et al. [31] formulated TB-HIV co-infection model to study the role of screening and treatment. They considered active sexual adult people in the model. Their result showed that increasing the rate of screening TB leads to decreasing TB infectious people. The researchers recommend that strong cooperation between the TB and HIV intervention regime is necessary to control the disease. M. M. Ojo et al. [32]. investigated impact of vaccination on the dynamics of tuberculosis using mathematical modelling. They discussed the stabilities of equilbrum points and exhibited the occurrence of backward bifurcation in the model. The authors explored the influence of vaccination rate, vaccine efficacy, and effective contact rate on the active TB infectious individuals. Their findings showed that rising the vaccination rate of susceptible people reduces the TB disease burden. Nevertheless, the vaccine efficacy must remain above 25% to minimize the disease problem effectively. They concluded that, reducing the contact rate with TB infectious and expanding vaccination with high vaccine efficiency will diminish the TB disease prolifically. Similarly, the dynamics of Tuberculosis (TB) Outbreak have been examined using mathematical modelling approach, where effective use of treatment rates and isolation [33]. Fatmawati et al. [14] studied the effect of antibiotics and ART optimally to control the transmission dynamics of HIV-TB co-epidemic. Their numerical result showed that coupling of ART and anti-TB optimal control is the most effective strategy to fight against the disease. However, they suggested that antibiotics are better than ART when only one control is used. Grace et.al [34] explored the impact of HIV on TB infection via considering ART and TB treatment in Kenya. Their result suggested that testing and administering latent TB, both ART and TB treatment, HIV testing for all TB patients and vice versa are very crucial for Kenyan people. Roeger et al. [23] introduced a deterministic model of TB-HIV co-infection. They analysed the model and their numerical result suggested that the presence of HIV leads to increase the cases of co-infectious individuals even if TB reproduction is less than unity. The authors endorsed that; more effort should be given for reducing HIV prevalence to control TB infection in the co-infection populace. Awoke et al. [35] proposed TB-HIV/AIDS co-epidemics model with behavioural modification. They extended the model into an optimal control problem by considering behavioural modification as preventive measures and treatment efforts as controlling strategies. Their numerical result showed that applying both preventive and control measures can reduce the disease and cost burden. The authors declared that the cost of applying preventive effort is very small as compared to treatment, but the cost of administering the infection is huge when the rate of disease transmission is high.

They conclude that applying both prevention and treatment efforts at a time is a best effective strategy.

More recently, the modeling of TB-HIV and related diseases transmission dynamics are investigated [36–39]. Hadipour et al. [40] investigated TB–HIV co-epidemics treatment controls. They used a mathematical model along with an optimum sliding mode controller. The researchers applied a multi-objective genetic optimization algorithm to find the optimal values of the control coefficients. Their result showed that when controls are applied new infections, disease deaths, and total burden values are reduced rather than without control. A. Ahmad et al. [41] considered TB-HIV co-infection model for mathematical analysis and numerical simulation. They used non-standard finite difference technique with Mickens approach $\phi(h) = h + O(h^2)$ to analysis the model numerically. The researchers presented the graphical scenarios of each compartment for equilibrium points of the model by varying the step size $h$. Their result displayed that when the step size is increasing slightly, the susceptibility of individuals with HIV, TB, and their co-infection are increasing, otherwise decreasing. Finally, their result showed that the disease burden is decreased when they projected the time duration. Aggarwal and Raj [42] formulated and analysed a fractional order model in Caputo sense for TB-HIV co-epidemics along with recurrent TB and exogenous reinfection. They justified that, when the reproduction number for TB is less than unity, the existence of the backward bifurcation occurred in a certain domain. The researchers considered the memory effect in the Caputo fractional order model and they analysed the model numerically. They showed that the rate of state trajectories convergence to the equilibrium points is more for higher order derivatives rather than small order. The authors displayed that the number of TB and HIV infectious have lower increment for a smaller fractional order. Finally, they conclude that the fractional order of derivative plays a key role against the disease prevalence along with integration of memory effect. Moreover, the two diseases HIV and TB can also be co-infected with the current COVID-19 global pandemic. M.M. Ojo et al. [43] established a co-infection deterministic mathematical model for TB and COVID-19, to investigate their co-infection nature and each disease impact in the community. They performed various simulations to investigate the effect of transmission rates and threshold quantities on the co-infection disease. Their results addressed that the threshold quantities (both effective and invasion reproduction numbers) are the defining factors for disease incursion in the populace. The authors demonstrated that the disease with the maximum invasion reproduction number would dominate but does not drive the other towards elimination. Their numerical analysis also presented that the rising of co-infection transmission rate would upsurge the TB prevalence. Finally, they endorsed the strategies which are prioritizing to diminution the diseases in the population. K. G. Mekonen et al. [44] studied the optimal control of TB and COVID-19 co-epidemics model. Their analytical result revealed that, the rising of TB infected individuals has a positive impact on the transmission of COVID-19 disease and vice versa. They incorporated prevention efforts against TB and COVID-19, treatment for TB, and medical care for COVID-19 infection in the model. Their numerical results showed that the prevalence of co-epidemic disease can be reduced when applying all controlling strategies at a time. N. Ringa et al. [45] formulated a mathematical model of HIV-COVID-19 co-epidemics disease with optimal control. The researchers incorporated HIV and COIVD-19 prevention mechanisms and treatment for COVID-19 in the model. They analysed the model with and without optimal control analytically. The authors also analysed the model numerically based on the data taken from South Africa. Their result showed that the HIV preventive effort and treatment of COVID-19 can minimize the co-infection disease burden.

As we have seen from literatures made by different scholars, TB and HIV/AIDS co-epidemic is a public health concern particularly in developing countries with limited resource.

In this study, we developed a TB-HIV co-infection mathematical model based on the TB model [46] via HIV/AIDS cohorts taking in to account high risk exposed stage (E) and low risk latent TB by treatment (L). Law-risk and high-risk latent TB infected individuals co-infected with HIV are considered in the model formulation. To the best of our knowledge, no other study has examined this possibility. Likewise, we formulated a co-infection model with optimal control consisting high and low risk latent TB stages which are currently responsible for rising TB infectious cases in Ethiopia [19, 47–49]. Similarly, Guo et al. formulated age-structured HIV-TB co-infection model to investigate how to regulate the TB transmission in china [50]. They considered only detection and treatment of latent TB cases and educational campaign to mitigate the TB disease. Whereas, our model examined four controlling strategies which are preventive efforts of TB, preventive efforts of HIV/AIDS, case finding for TB, and HIV treatment to reduce the TB-HIV co-infectious individuals. These strategies were not examined on a TB-HIV co-epidemic model comprises HIV-TB co-infectious people who cannot be fully recovered from TB, which motivates us to undertake this investigation and fill the gap. The other motivation is also the concept of optimal control theory which is applicable to study controlling mechanisms of a disease. The optimal model analysis can be used to identify an effective strategy to minimize the number of infectious individuals and concurrently to reduce the cost incurred during implementation of strategies. To validate our model, the model solution is fitted to the real data collected from Ethiopia. Hence, we developed and analyzed an optimal HIV-TB co-epidemic model with the four suggested interventions using real data, which is the novelty of this work and makes it different from other approaches in the literature.

Thus, the purpose of this study is to establish optimal HIV/AIDS- TB co-infection mathematical model to explore the best cost-effective measure to mitigate the disease burden. Thus, we used time based controlling efforts and we applied optimal combinations of two or more strategies at a time. As shown in all plots (Figs 4–15), the co-infected individuals are dramatically decreased. The elaboration will be presented in the numerical simulations.

The main contribution of this study is to propose the best cost-effective controlling mechanism in the fight against the TB-HIV/AIDS co-epidemic disease in Ethiopia. We chose the country, Ethiopia, because it is one of the $14^{th}$ countries in the world affected by HIV-TB co-infection disease, and also one of the $8^{th}$ most affected countries in the African continent [28, 51].

The rest of paper is divided as follows: In the following section, the model construction is presented. Section 3 covers the analytical findings of the model properties. In section 4 the proposed model is extended and analyzed analytically through controlling variables while in section 5 the numerical results of the model are presented. The cost-effectiveness of the proposed strategies and conclusions are displayed in section 6 and 7 respectively.

## 2 Model formulation

We incorporated the following model assumptions to develop a TB-HIV/AIDS co-infection model.

- Individuals infected with TB cannot fully recover, but to latent TB [46, 52]. The TB bacteria cannot removed 100% from the TB infected individual's body.

- Individuals co-infected with AIDS and active TB are very ill. They have not transmitted HIV virus due to sexual intercourse [53]. They are protected in treatment and their allure for sex is almost insignificant.

- HIV infected individuals under ART treatment are aware of transmitting the disease [7, 27]. The present model will consider those people who are strictly under care and closely monitored.

- The model does not consider vertical transmission of HIV-AIDS and immigrant individuals [54]. HIV can also transmit vertically from HIV infected mother to their new infants. The disease transmission may occur during pregnancy or breastfeeding. Thus, the model which is formulated in this study has not contemplated inflow of HIV-infected individuals. This endorses that the susceptible class comprises new birth people who are vulnerable to HIV infection.

The theory of such an epidemic disease progress in a large population. Initially, the populations diversity can divide into subgroups considering with the nature and the stages of the disease. The sub-groups of populations are named as compartments.

We developed a TB-HIV co-infection model by mixing HIV/AIDS (susceptible, HIV infection with and without AIDS symptoms, and treated individuals from HIV infection) with TB model [46]. Hence, the model allocated the human populations into the following epidemiological compartments. Namely, susceptible individuals (S), exposed (or a high-risk latent TB) (E) that is infected but not infectious individuals, infectious TB (I), low-risk latent TB (L), HIV-infected individuals with no clinical symptoms of AIDS (H), HIV-infected people under treatment for HIV infection (T), HIV-infected individuals with AIDS clinical symptoms (A), exposed (or a high-risk latent TB) co-infected with HIV ($H_E$), low risk latent TB individuals co-infected with HIV ($H_L$), HIV-infected individuals (pre-AIDS) co-infected with active TB disease ($H_I$), HIV-infected individuals with AIDS symptoms co-infected with active TB ($A_I$) and low risk latent TB individuals infected by HIV-infection with AIDS symptoms ($A_L$).

Thus, the total population at time $t$, denoted by $N(t)$, is given by:

$$N(t) = S(t) + E(t) + I(t) + L(t) + H(t) + A(t) + T(t) + H_E(t) + H_L(t) + H_I(t) + A_I(t) + A_L(t). \quad (1)$$

The susceptible population is increased by the recruitment of individuals (new births) at a rate $\pi$. These people acquire TB and HIV infection at a variable rate or force of infection: $\lambda_T(t) = \frac{\beta_1[I(t)+H_I(t)+A_I(t)]}{N(t)}$ and $\lambda_H(t) = \frac{\beta_2[H(t)+H_E(t)+H_L(t)+H_I(t)+\eta(A(t)+A_L(t))]}{N(t)}$ respectively.

The parameters $\beta_1$ and $\beta_2$ are rates for TB transmission and HIV transmission respectively. Whereas, the modification parameter $\eta$ represents the relative infectiousness of people with AIDS symptoms compared to HIV infected people without AIDS symptoms. HIV-infected people (pre-AIDS) are less infectious than people with AIDS symptoms because they have lower viral load and positive relationship among infectiousness and viral load [55]. In general, all parameters involved in the model formulation are described in Table 1.

**Table 1. Descriptions of the parameters.**

| Parameters | Descriptions |
|---|---|
| $\pi$ | Recruitment rate |
| $\mu$ | Per capita natural mortality rate |
| $\beta_1$ | TB transmission rate |
| $\beta_2$ | HIV transmission rate |
| $\eta$ | Rate of infectiousness level |
| $k$ | Per capita progression rate from class E to I |
| $\alpha$ | Treatment rate of E |
| $\sigma$ | The relapse rate due to tubercle bacilli reactivation |
| $1-p$ | Successful treatment rate of I |
| $\gamma$ | TB treatment rate |
| $\omega_i, i = 1, 2, 5, 6$ | Rate of recruitment to receive HIV treatment for $H$, $A$, $H_L$, and $A_L$ respectively |
| $\omega_4, \omega_7$ | Rate of recruitment to receive both HIV and TB treatment for $H_I$ and $A_I$ respectively |
| $\omega_3$ | Rate of recruitment to receive HIV treatment and treatment of high risk latent TB |
| $\omega, \theta, \epsilon_1, \epsilon_2$ | Modification parameters |
| $\delta$ | Progression rate from H to A |
| $\epsilon$ | Per capita progression rate of TB from class $H_E$ to $H_I$ |
| $\phi$ | Fraction of individuals from $H_I$ class that receive treatments for TB only |
| $\sigma_1$ | HIV Progression rate from $H_E$ to $A_I$ |
| $\delta_1$ | TB Progression rate from $H_E$ to $A_I$ |
| $\theta_1$ | The relapse rate due to tubercle bacilli reactivation |
| $\theta_2$ | The recruitment rate of individuals from $H_E$ to $H_L$ due to treatment of latent TB |
| $(1-\psi)$ | Successful TB treatment rate of $H_I$ |
| $\psi_1$ | Progression rate from $H_I$ to $A_I$ |
| $\varphi$ | Rate of failure to properly adhere to HIV treatment rules |
| $\theta_3$ | Progression rate from $H_L$ to $A_L$ |
| $\tau$ | Complete treatment rate of TB from $A_I$ to $A_L$ |
| $d_i\{i = 1, 2, 3\}$ | Per capita TB, HIV, and TB-HIV co-infection induced death rate |

The co-infection transmission dynamics flow diagram in Fig 1 can be described by the following deterministic system of non-linear ODE.

$$
\begin{cases}
\frac{dS}{dt} = \pi - (\lambda_H + \lambda_T + \mu)S, \\
\frac{dE}{dt} = \lambda_T S + \gamma p I + \sigma L - (k + \alpha + \epsilon_1 \lambda_H + \mu)E, \\
\frac{dI}{dt} = kE - (\gamma + \epsilon_2 \lambda_H + d_1 + \mu)I, \\
\frac{dL}{dt} = (1-p)\gamma I + \alpha E - (\sigma + \lambda_H + \mu)L, \\
\frac{dH}{dt} = \lambda_H(S + L) - (\theta\lambda_T + \delta + \omega_1 + \mu)H, \\
\frac{dA}{dt} = \delta H + \varphi T - (\omega_2 + \omega\lambda_T + \mu + d_2)A, \\
\frac{dH_E}{dt} = \epsilon_1 \lambda_H E + \theta\lambda_T H + \psi\gamma H_I + \theta_1 H_L - (\epsilon + \omega_3 + \theta_2 + \sigma_1 + \delta_1 + \mu)H_E, \\
\frac{dH_I}{dt} = \epsilon_2 \lambda_H I + \epsilon H_E - (\psi\gamma + (1-\psi)\phi\gamma + (1-\phi)\omega_4 + \psi_1 + \mu + d_1)H_I, \\
\frac{dH_L}{dt} = (1-\psi)\phi\gamma H_I + \theta_2 H_E - (\theta_1 + \theta_3 + \omega_5 + \mu)H_L, \\
\frac{dT}{dt} = \omega_1 H + \omega_2 A + \omega_3 H_E + \omega_4(1-\phi)H_I + \omega_5 H_L + \omega_6 A_L + \omega_7 A_I - (\varphi + \mu)T, \\
\frac{dA_L}{dt} = \tau A_I + \theta_3 H_L - (\omega_6 + \mu + d_2)A_L, \\
\frac{dA_I}{dt} = (\sigma_1 + \delta_1)H_E + \psi_1 H_I + \omega\lambda_T A - (\omega_7 + \mu + d_3 + \tau)A_I,
\end{cases}
\tag{2}
$$

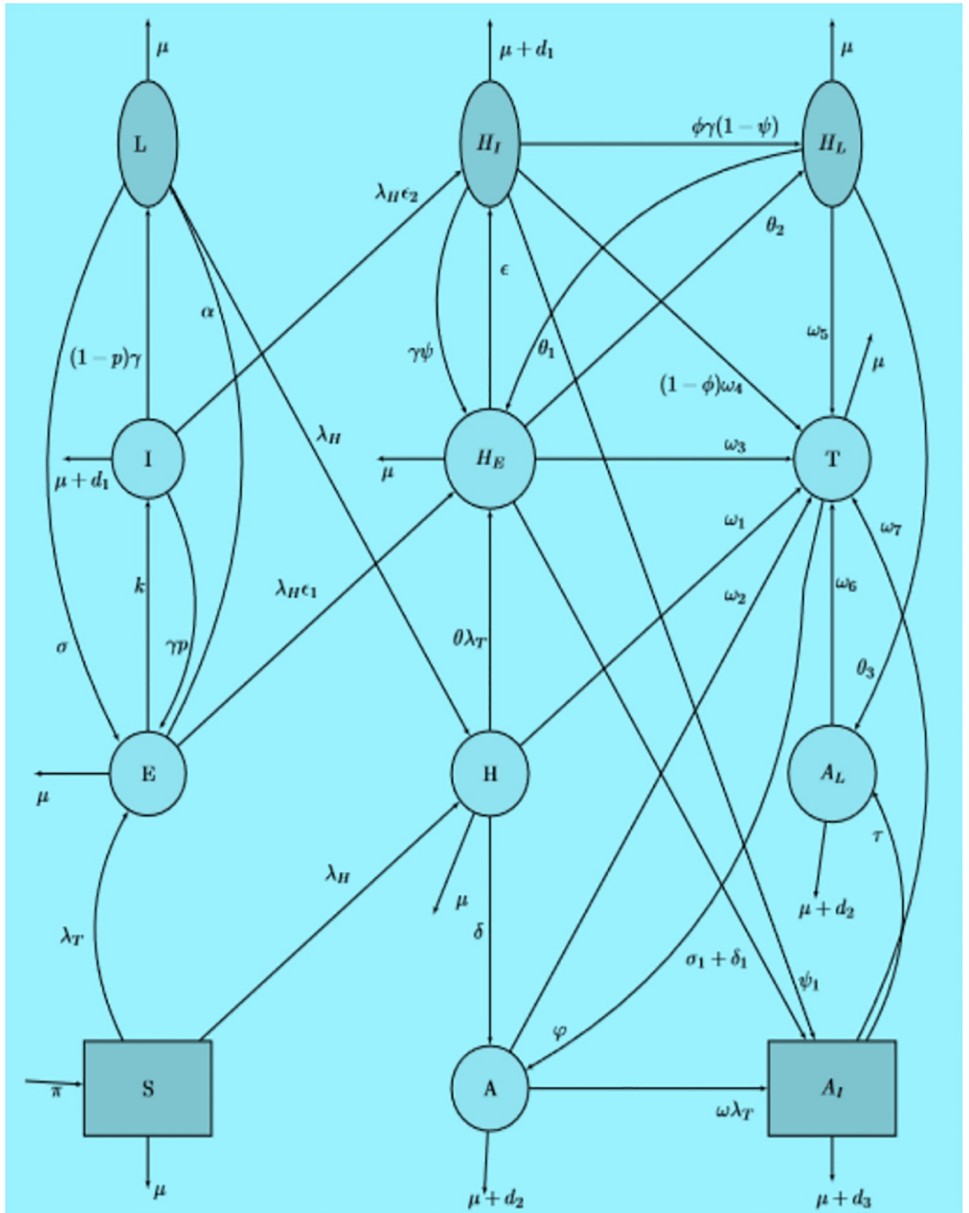

**Fig 1. Flow diagram of the TB-HIV/AIDS co-infection disease transmission.**

with inital conditions are:

$$S(0) > 0, E(0) \geq 0, I(0) \geq 0, L(0) \geq 0, H(0) \geq 0, A(0) \geq 0, H_E(0) \geq 0, H_I(0) \geq 0,$$

$$H_L(0) \geq 0, T(0) \geq 0, A_L(0) \geq 0, \text{and } A_I(0) \geq 0.$$

(3)

## 3 Model analysis

### 3.1 Positivity of the solutions

For the model (2) to be epidemiologically meaningful, we have to prove that the solutions of all state variables are positive. The system of equation (2) expresses that human population in different compartments. Each state variable and parameter of the model are positive. Thus, we state the next theorem.

**Theorem 1** *let* $\Omega = \{(S, E, I, L, H, A, H_E, H_I, H_L, T, A_L, A_I) \in \mathbb{R}_+^{12} : S(0) > 0, E(0) > 0, I(0) > 0, L(0) > 0, H(0) > 0, A(0) > 0, H_E(0) > 0, H_I(0) > 0, H_L(0) > 0, T(0) > 0, A_I(0) > 0, A_L(0) > 0\}$ *then the solutions* $(S(t), E(t), I(t), L(t), H(t), A(t), H_E(t), H_I(t), H_L(t), T(t), A_L(t), A_I(t))$ *of* (2) *are positive for* $\forall t \geq 0$.

**Proof:**

Consider the system (2) and let us take the first equation

$$\frac{dS(t)}{dt} = \pi - (\lambda_H + \lambda_T + \mu)S,$$

$$\frac{dS(t)}{dt} = \pi - \left(\frac{\beta_2[H(t) + H_E(t) + H_L(t) + H_I(t) + \eta(A(t) + A_L(t))]}{N(t)} + \frac{\beta_1[I(t) + H_I(t) + A_I(t)]}{N(t)} + \mu\right)S,$$

$$\frac{dS(t)}{dt} \geq -\left(\frac{\beta_2[H(t) + H_E(t) + H_L(t) + H_I(t) + \eta(A(t) + A_L(t))]}{N(t)} + \frac{\beta_1[I(t) + H_I(t) + A_I(t)]}{N(t)} + \mu\right)S, \quad (4)$$

$$\frac{dS(t)}{S} \geq -(\beta_2[H(t) + H_E(t) + H_L(t) + H_I(t) + \eta(A(t) + A_L(t))] + \beta_1[I(t) + H_I(t) + A_I(t)] + \mu)dt.$$

Integrating the last inequality of (4) using the method of separable of the variables and using initial condition when solving a state variable $S(t)$, we obtain:

$$S(t) \geq S(0)exp^{-\int(\beta_2[H(t)+H_E(t)+H_L(t)+H_I(t)+\eta(A(t)+A_L(t))]+\beta_1[I(t)+H_I(t)+A_I(t)]+\mu)dt} > 0, \quad (\text{since } S(0) > 0). \quad (5)$$

Let us take the second equation

$$\frac{dE}{dt} = \lambda_T S + \gamma pI + \sigma L - (k + \alpha + \epsilon_1\lambda_H + \mu)E,$$

$$\frac{dE}{dt} \geq -(k + \alpha + \epsilon_1\lambda_H + \mu)E,$$

$$\frac{dE}{dt} \geq -(k + \alpha + \epsilon_1\beta_2[H(t) + H_E(t) + H_L(t) + H_I(t) + \eta(A(t) + A_L(t))] + \mu)E, \quad (6)$$

$$E(t) \geq E(0)exp^{-\int(\beta_2[H(t)+H_E(t)+H_L(t)+H_I(t)+\eta(A(t)+A_L(t))]+\mu)dt} > 0, \quad (\text{since } E(0) > 0).$$

Similarly, let us take the rest ten equations of model (2), we have:

$$
\begin{aligned}
\frac{dI}{dt} &\geq -(\gamma + \epsilon_2 \lambda_H + d_1 + \mu)I, \\[2mm]
\frac{dL}{dt} &\geq -(\sigma + \lambda_H + \mu)L, \\[2mm]
\frac{dH}{dt} &\geq -(\theta \lambda_T + \delta + \omega_1 + \mu)H, \\[2mm]
\frac{dA}{dt} &\geq -(\omega_2 + \omega \lambda_T + \mu + d_2)A, \\[2mm]
\frac{dH_E}{dt} &\geq -(\epsilon + \omega_3 + \theta_2 + \sigma_1 + \delta_1 + \mu)H_E, \\[2mm]
\frac{dH_I}{dt} &\geq -(\psi\gamma + (1-\psi)\phi\gamma + (1-\phi)\omega_4 + \psi_1 + \mu + d_1)H_I, \\[2mm]
\frac{dH_L}{dt} &\geq -(\theta_1 + \theta_3 + \omega_5 + \mu)H_L, \\[2mm]
\frac{dT}{dt} &\geq -(\varphi + \mu)T, \\[2mm]
\frac{dA_L}{dt} &\geq -(\omega_6 + \mu + d_2)A_L, \\[2mm]
\frac{dA_I}{dt} &\geq -(\omega_7 + \mu + d_3 + \tau)A_I.
\end{aligned}
\tag{7}
$$

Thus, we obtained the following solutions of inequalities.

$$
\begin{aligned}
I(t) &\geq I(0)exp^{-\int (\gamma + \epsilon_2\beta_2[H(t)+H_E(t)+H_L(t)+H_I(t)+\eta(A(t)+A_L(t))]+d_1+\mu)\,dt} \geq 0, \\[2mm]
L(t) &\geq L(0)exp^{-\int (\sigma + \beta_2[H(t)+H_E(t)+H_L(t)+H_I(t)+\eta(A(t)+A_L(t))]+\mu)dt} \geq 0, \\[2mm]
H(t) &\geq H(0)exp^{-\int (\theta\beta_1[I(t)+H_I(t)+A_I(t)]+\delta+\omega_1+\mu)dt} \geq 0, \\[2mm]
A(t) &\geq A(0)exp^{-\int (\omega_2 + \omega\beta_1[I(t)+H_I(t)+A_I(t)]+\mu+d_2)dt} \geq 0, \\[2mm]
H_E(t) &\geq H_E(0)exp^{-\int (\epsilon+\omega_3+\theta_2+\sigma_1+\delta_1+\mu)dt} \geq 0, \\[2mm]
H_I t &\geq H_I(0)exp^{-\int (\psi\gamma+(1-\psi)\phi\gamma+(1-\phi)\omega_4+\psi_1+\mu+d_1)dt} \geq 0, \\[2mm]
H_L(t) &\geq H_L(0)exp^{-\int (\theta_1+\theta_3+\omega_5+\mu)dt} \geq 0, \\[2mm]
T(t) &\geq T(0)exp^{-\int (\varphi+\mu)dt} \geq 0, \\[2mm]
A_L(t) &\geq A_L(0)exp^{-\int (\omega_6+\mu+d_2)dt} \geq 0, \\[2mm]
A_I(t) &\geq A_I(0)exp^{-\int (\omega_7+\mu+d_3+\tau)dt} \geq 0.
\end{aligned}
\tag{8}
$$

Therefore, the solution of each state variable with positive initial value is positive. This completes the proof.

## 3.2 Invariance region

In this portion, we assured that the solutions of the system (2) is bounded with non-negative initial values.

**Lemma 1** *Let $\Omega$ be the biological feasible region such that*

$\Omega = \{(S, E, I, L, H, A, H_E, H_I, H_L, T, A_L, A_I) \in \mathbb{R}_+^{12} : N \leq \frac{\pi}{\mu}\}$. *Then $\Omega$ is positively invariant set for the system* (2) *and attracts all positive solutions.*

**Proof**:

We showed this clue by adding all the equations in (2) to get the rate of change of $N(t)$.

$$\frac{dN}{dt} = \pi - \mu N(t) - d_1 I - d_1 H_I - d_2 A - d_2 A_L - d_3 A_I. \tag{9}$$

Hence, Eq (9) which is a first order ODE can be written as:

$$\frac{dN}{dt} \leq \pi - \mu N(t). \tag{10}$$

Which yields,

$$N(t) \leq \frac{\pi}{\mu} + e^{-\mu t}\left(N(0) - \frac{\pi}{\mu}\right). \tag{11}$$

Here, $0 < N(0) \leq \frac{\pi}{\mu}$, then we derived $0 < N(t) \leq \frac{\pi}{\mu}, \forall\, t \geq 0$. This indicates that the TB-HIV co-infection model (2) is positive invariance and bounded $\forall > 0$. Hence, the system is well-posed and biologically realistic.

Hereafter, we analyzed each sub model before exploring the full co-infection model.

## 3.3 HIV-only model

If the compartments $E = L = I = H_E = H_I = H_L = A_L = A_I = 0$, then the model (2) purely represents only HIV/AIDS model. Hence, the system (2) is expressed as:

$$\begin{cases} \dfrac{dS}{dt} = \pi - (\lambda_H + \mu)S, \\[2mm] \dfrac{dH}{dt} = \lambda_H S - (\delta + \omega_1 + \mu)H, \\[2mm] \dfrac{dA}{dt} = \delta H + \varphi T - (\omega_2 + \mu + d_2)A, \\[2mm] \dfrac{dT}{dt} = \omega_1 H + \omega_2 A - (\varphi + \mu)T, \end{cases} \tag{12}$$

where $\lambda_H(t) = \frac{\beta_2[H(t) + \eta A(t)]}{N(t)}$ and $N(t) = S(t) + H(t) + A(t) + T(t)$.

Let $\Omega_A$ be the set such that $\Omega_A = \{S, H, A, T \in \mathbb{R}_+^4 : N \leq \frac{\pi}{\mu}\}$. Then similar to lemma (1) we can show that $\Omega_A$ is positively invariant and attracting. So, the HIV only model is contemplated in $\Omega_A$.

**3.3.1 Local stability of disease free equilbrum.** In the absence of HIV infection, we obtained the DFE point of HIV only sub-model (12) by equating the right-hand side of the equation (12) to be zero and is given by $E_0 = (\frac{\pi}{\mu}, 0, 0, 0)$.

**Theorem 2** *The DFE point of system* (12) *is locally asymptotically stable (LAS) if* $R_H < 1$ *and unstable if* $R_H > 1$.

**Proof:**

Firstly, find the basic reproduction number $R_H$ which is the spectral radius of the matrix $FV^{-1}$, where $F$ is the matrix of new infection terms which is $F = \begin{bmatrix} \frac{\beta_2 \pi}{N\mu} & \frac{\beta_2 \eta \pi}{N\mu} & 0 \\ 0 & 0 & 0 \\ 0 & 0 & 0 \end{bmatrix} =$

$\begin{bmatrix} \beta_2 & \beta_2\eta & 0 \\ 0 & 0 & 0 \\ 0 & 0 & 0 \end{bmatrix}$ and $V$ is the matrix of remaining transfer terms expressed as: $V =$

$\begin{bmatrix} \delta + \omega_1 + \mu & 0 & 0 \\ -\delta & \omega_2 + d_2 + \mu & -\varphi \\ -\omega_1 & -\omega_2 & (\varphi + \mu) \end{bmatrix}$ at DFE point [56].

The inverse of $V$ is calculated and which is expressed by:

$$V^{-1} = \begin{bmatrix} \dfrac{1}{A} & 0 & 0 \\ \dfrac{\delta}{AB} - \dfrac{\varphi(\delta\omega_2 + \omega_1 B)}{AB(\varphi\omega_2 - CB)} & \dfrac{1}{B} - \dfrac{\omega_2\varphi}{B(\varphi\omega_2 - CB)} & \dfrac{-\varphi}{\varphi\omega_2 - CB} \\ \dfrac{\delta\omega_2 + \omega_1 B}{A(\varphi\omega_2 - CB)} & \dfrac{-\omega_2}{\varphi\omega_2 - CB} & \dfrac{-B}{\varphi\omega_2 - CB} \end{bmatrix},$$

where $A = \delta + \omega_1 + \mu$, $B = \omega_2 + d_2 + \mu$, $C = \varphi + \mu$. The representations $A$, $B$, $C$ are used for the entire work.

The product of $F$ and $V^{-1}$ gives:

$$FV^{-1} = \frac{\beta_2}{(\varphi\omega_2 - CB)} \begin{bmatrix} \dfrac{(\varphi\omega_2 - CB) - \eta(\delta C + \omega_1\varphi)}{A} & -C\eta & -\eta\varphi \\ 0 & 0 & 0 \\ 0 & 0 & 0 \end{bmatrix}.$$

Hereafter, the eigenvalue of $FV^{-1}$ can be calculated by the determinant of the matrix below.

$$\begin{vmatrix} \left(\dfrac{\beta_2}{(\varphi\omega_2 - CB)}\right)\dfrac{(\varphi\omega_2 - CB) - \eta(\delta C + \omega_1\varphi)}{A} - \lambda & -C\eta\left(\dfrac{\beta_2}{(\varphi\omega_2 - CB)}\right) & -\eta\varphi\left(\dfrac{\beta_2}{(\varphi\omega_2 - CB)}\right) \\ 0 & -\lambda & 0 \\ 0 & 0 & -\lambda \end{vmatrix}$$

$$= 0.$$

This implies,

$$\lambda^2 \left[\frac{\beta_2}{(\varphi\omega_2 - CB)}\left(\frac{(\varphi\omega_2 - CB) - \eta(\delta C + \omega_1\varphi)}{A}\right) - \lambda\right] = 0.$$

Therefore, the dominant eigenvalue is the basic reproduction number $R_H$ such that:

$$
\begin{aligned}
R_H &= \frac{\varphi\omega_2\beta_2 - \beta_2 CB - \beta_2\eta\delta C - \varphi\beta_2\eta\omega_1}{A(\varphi\omega_2 - BC)}, \\
&= \frac{\beta_2(\varphi\omega_2 - CB - \eta\delta C - \eta\omega_1\varphi)}{A(\varphi\omega_2 - CB)}, \\
&= \frac{\beta_2}{A}\left[1 + \frac{\eta(\delta C + \omega_1\varphi)}{CB - \varphi\omega_2}\right], \\
&= \frac{\beta_2}{(\delta + \omega_1 + \mu)}\left[1 + \frac{\eta(\delta\varphi + \delta\mu + \omega_1\varphi)}{\varphi d_2 + \varphi\mu + \omega_2\mu + d_2\mu + \mu^2}\right].
\end{aligned}
\tag{13}
$$

Now, the Jacobian matrix of the system (12) at DFE point is given by:

$$
J\left(\frac{\pi}{\mu}, 0, 0, 0\right) = \begin{bmatrix}
-\mu & -\beta_2\frac{\pi}{\mu N(t)} & -\beta_2\eta & 0 \\
0 & \beta_2\frac{\pi}{\mu N(t)} - (\delta + \omega_1 + \mu) & \beta_2\eta\frac{\pi}{\mu N(t)} & 0 \\
0 & \delta & -(\omega_2 + d_2 + \mu) & \varphi \\
0 & \omega_1 & \omega_2 & -(\varphi + \rho)
\end{bmatrix}.
$$

Secondly, we calculated the eigenvalues of this matrix as follows.

$$
\begin{vmatrix}
-\mu - \lambda & -\beta_2\frac{\pi}{\mu N(t)} & -\beta_2\eta\frac{\pi}{\mu N(t)} & 0 \\
0 & [\beta_2\frac{\pi}{\mu N(t)} - (\delta + \omega_1 + \mu)] - \lambda & \beta_2\eta\frac{\pi}{\mu N(t)} & 0 \\
0 & \delta & -(\omega_2 + d_2 + \mu) - \lambda & \varphi \\
0 & \omega_1 & \omega_2 & -(\varphi + \rho) - \lambda
\end{vmatrix} = 0.
$$

$$
\Rightarrow \begin{vmatrix}
-\mu - \lambda & -\beta_2\frac{\pi}{\mu N(t)} & -\beta_2\eta\frac{\pi}{\mu N(t)} & 0 \\
0 & [\beta_2\frac{\pi}{\mu N(t)} - A] - \lambda & \beta_2\eta\frac{\pi}{\mu N(t)} & 0 \\
0 & \delta & -B - \lambda & \varphi \\
0 & \omega_1 & \omega_2 & -C - \lambda
\end{vmatrix} = 0.
$$

Finally, we obtained a third order polynomial equation as:

$$
(-\mu - \lambda)[\lambda^3 + \lambda^2\left(A + B + C - \frac{\beta\pi}{N\mu}\right) + \lambda(BA + BC + AC - B\frac{\beta\pi}{N\mu} - \delta\frac{\beta\pi}{N\mu} - \omega_2\varphi - C\frac{\beta\pi}{N\mu}) +
\tag{14}
$$
$$
ABC - CB\frac{\beta\pi}{N\mu} - \delta\frac{\beta\pi}{N\mu} + \varphi\omega_2\frac{\beta\pi}{N\mu} - \omega_2\varphi A - \omega_1\varphi\eta\frac{\beta\pi}{N\mu}] = 0.
$$

The Eq (14) has the form $A_0\lambda^3 + A_1\lambda^2 + A_2\lambda + A_3 = 0$. Hence, this polynomial equation has strictly negative real parts, if $\Delta_1 > 0$, $\Delta_2 > 0$, and $\Delta_3 > 0$ by Routh-Hurwitz stability criteria.

Here, $\Delta_0 = A_0 > 0, \Delta_1 = \begin{vmatrix} A_1 & A_0 \\ A_3 & A_2 \end{vmatrix} = A_1 A_2 - A_0 A_3 > 0,$

$\Delta_2 = \begin{vmatrix} A_1 & A_0 & 0 \\ A_3 & A_2 & A_1 \\ A_5 & A_4 & A_3 \end{vmatrix} = \begin{vmatrix} A_1 & A_0 & 0 \\ A_3 & A_2 & A_1 \\ 0 & 0 & A_3 \end{vmatrix} = A_3(A_1 A_2 - A_0 A_3) = A_3 \Delta_1 > 0,$ otherwise $E_0$ is unstable.

Hence, $\Delta_0 = A_0 = 1 > 0, \Delta_1 = (A + B + C - \frac{\beta\pi}{N\mu})(BA + BC + AC - B\frac{\beta\pi}{N\mu} - \delta\frac{\beta\pi}{N\mu} - \omega_2\varphi - C\frac{\beta\pi}{N\mu}) - (ABC - CB\frac{\beta\pi}{N\mu} - \delta\frac{\beta\pi}{N\mu} + \varphi\omega_2\frac{\beta\pi}{N\mu} - \omega_2\varphi A - \omega_1\varphi\eta\frac{\beta\pi}{N\mu}) > 0,$ and $\Delta_2 = A_3 \Delta_1$. The expression $\Delta_2$ is already positive.

Thus, the value $A_3 = ABC - CB\frac{\beta\pi}{N\mu} - \delta\frac{\beta\pi}{N\mu} + \varphi\omega_2\frac{\beta\pi}{N\mu} - \omega_2\varphi A - \omega_1\varphi\eta\frac{\beta\pi}{N\mu} = ABC - CB\beta_2 - \delta\beta_2 + \varphi\omega_2\beta_2 - \omega_2\varphi A - \omega_1\varphi\eta\beta_2,$ since $N = \frac{\pi}{\mu}$ at DFE point.

Therefore, $A_3 = A(\varphi\omega_2 - BC)[1 - \frac{\beta_2(\varphi\omega_2 - CB - \eta\delta C - \eta\omega_1\varphi)}{A(\varphi\omega_2 - CB)}] = (d_2\varphi + \mu^2 + \mu\varphi + \mu d_2 + \omega_2\varphi) \times [1 - R_H] > 0,$ iff $1 - R_H > 0$ which implies $R_H < 1$. This completed the proof.

Hence, the epidemiological interpretation of Theorem (2) is that the number of HIV-infected people will gradually become lower and lower if the new infection generated by a person during his/her infection period is less than one. In addition, the initial size of the sub-population of the system (12) are in the basin of attraction of $E_0$.

**3.3.2 Global stability of the DFE point, $E_0$.** The global stability of the DFE point can be explored by using the method in [57]. The model (12) can be expressed by:

$$\begin{cases} \dfrac{dX_s}{dt} = Q(X_s - X_{DFE,s}) + Q_1 X_i, \\ \dfrac{dX_i}{dt} = Q_2 X_i, \end{cases} \quad (15)$$

where the vectors $X_s$ and $X_i$ are representing the non-transferring and transferring compartments. If $Q$ has real negative eigenvalues and $Q_2$ is a Metzler matrix, then the DFE point is globally asymptotically stable (GAS).

Thus,

$$X_s = (S, T)^T, X_i = (I, A)^T,$$

$$X_s - X_{DFE,s} = \begin{bmatrix} S \\ T \end{bmatrix} - \begin{bmatrix} \frac{\pi}{\mu} \\ 0 \end{bmatrix} = \begin{bmatrix} S - \frac{\pi}{\mu} \\ T \end{bmatrix}. \quad (16)$$

Now, the following matrices are constructed from $X_s$ and $X_i$ vectors.

$$Q = \begin{bmatrix} -\mu & 0 \\ 0 & -(\varphi + \mu) \end{bmatrix},$$

$$Q_1 = \begin{bmatrix} -\beta_2 \frac{S}{N(t)} & -\beta_2 \eta \frac{S}{N(t)} \\ \omega_1 & \omega_2 \end{bmatrix}, Q_2 = \begin{bmatrix} -(\delta + \omega_1 + \mu) & 0 \\ \delta & -(\omega_2 + d_2 + \mu) \end{bmatrix}. \quad (17)$$

As a result, the eigenvalues of $Q$ are negative and real implies that the system $\frac{dX_s}{dt} = Q(X_s - X_{DFE,s}) + Q_1 X_i$ is GAS at DFE point.

**3.3.3 Endemic equilibrium.** The endemic equilibrium (EE) point can be obtained by make it zero for the right side of the equations (12), where the disease is persist in the population.

Thus,

$$\frac{dS}{dt} = \pi - (\lambda_H + \mu)S = 0,$$

$$\frac{dH}{dt} = \lambda_H S - (\delta + \omega_1 + \mu)H = 0,$$

$$\frac{dA}{dt} = \delta H + \varphi T - (\omega_2 + \mu + d_2)A = 0,$$

$$\frac{dT}{dt} = \omega_1 H + \omega_2 A - (\varphi + \mu)T = 0.$$

(18)

Therefore, the solutions are:

$S^* = \frac{\pi}{\lambda_H^* + \mu}$, $H^* = \frac{\lambda_H^* S^*}{\delta + \omega_1 + \mu}$, $A^* = \frac{\delta H^* + \varphi T^*}{\omega_2 + \mu + d_2}$, and $T^* = \frac{\omega_1 H^* + \omega_2 A^*}{\varphi + \mu}$,

where $\lambda_H^*(t) = \frac{\beta_2 [H^*(t) + \eta A^*(t)]}{N^*(t)}$ and $N^*(t) = S^*(t) + H^*(t) + A^*(t) + T^*(t)$.

Therefore, the EE point say $E_1 = \{S^*, H^*, A^*, T^*\}$.

**Lemma 2** *A unique endemic equilibrium point $E_1$ exist if $R_H > 1$.*

**Proof.** If the disease is persist in the community, $\frac{dH}{dt} > 0, \frac{dA}{dt} > 0, \frac{dT(t)}{dt} > 0$. So, the system (12) becomes,

$$\begin{cases} \frac{dH}{dt} = \lambda_H S - (\delta + \omega_1 + \mu)H > 0, \\ \frac{dA}{dt} = \delta H + \varphi T - (\omega_2 + \mu + d_2)A > 0, \\ \frac{dT}{dt} = \omega_1 H + \omega_2 A - (\varphi + \mu)T > 0. \end{cases}$$

(19)

The system (19) becomes,

$$\begin{cases} (\delta + \omega_1 + \mu)H < \lambda_H S = \left(\frac{\beta_2 [H(t) + \eta A(t)]}{N(t)}\right)S, \\ (\omega_2 + \mu + d_2)A < \delta H + \varphi T, \\ (\varphi + \mu)T < \omega_1 H + \omega_2 A. \end{cases}$$

(20)

From the fact that $\frac{S}{N(t)} \leq 1$. Thus, the system (20) gives:

$$\begin{cases} (\delta + \omega_1 + \mu)H < \beta_2 [H(t) + \eta A(t)], \\ (\omega_2 + \mu + d_2)A < \delta H + \varphi T, \\ (\varphi + \mu)T < \omega_1 H + \omega_2 A. \end{cases}$$

(21)

Next, adding the second and the third inequalities of (21) by multiplying $(\varphi + \mu)$ and $\varphi$ respectively. We have the expression as:

$$A < \frac{\delta(\varphi + \mu) + \omega_1 \varphi}{(\varphi + \mu)(\omega_2 + \mu + d_2) - \omega_2 \varphi} H.$$

(22)

Substitute (22) to the first inequality of (21), we obtained the following inequality.

$$(\delta + \omega_1 + \mu)H(t) < \beta_2\left[H(t) + \eta\left(\frac{\delta(\varphi + \mu) + \omega_1\varphi}{(\varphi + \mu)(\omega_2 + \mu + d_2) - \omega_2\varphi}\right)H(t)\right],$$

$$AH(t) < \beta_2\left[H(t) + \eta\left(\frac{\delta C + \omega_1\varphi}{CB - \omega_2\varphi}\right)\right]H(t),$$

$$1 < \frac{\beta_2}{A}\left[1 + \eta\left(\frac{\delta C + \omega_1\varphi}{CB - \omega_2\varphi}\right)\right], \tag{23}$$

$$1 < \frac{\beta_2}{A}\left[1 + \eta\left(\frac{\delta C + \omega_1\varphi}{CB - \omega_2\varphi}\right)\right] = R_H,$$

$$R_H > 1.$$

This completes the proof.

**3.3.4 Global stability of the EE point.** **Theorem 3** *The endemic equilibrium ($E_1$) of model* (12) *is globally asymptotically stable (GAS) if $R_H > 1$.*

**Proof:**

We applied the procedure of Lyapunov functions.

Set the Lyapunov function as:

$G = (S - S^* - S^*\ln\frac{S}{S^*}) + (H - H^* - H^*\ln\frac{H}{H^*}) + (A - A^* - A^*\ln\frac{A}{A^*}) + (T - T^* - T^*\ln\frac{T}{T^*})$.

Such form of Lyapunov function has been stated in [58, 59].

Now, $\frac{dG}{dt} = G'\} = (\frac{S-S^*}{S})S' + (\frac{H-H^*}{H})H' + (\frac{A-A^*}{A})A'\} + (\frac{T-T^*}{T})T'\}$.

$= (\frac{S-S^*}{S})[\pi - (\lambda_H + \mu)S] + (\frac{H-H^*}{H})[\lambda_H S - (\delta + \omega_1 + \mu)H] + (\frac{A-A^*}{A})[\delta H + \varphi T - (\omega_2 + \mu + d_2)A] + (\frac{T-T^*}{T})[\omega_1 H + \omega_2 A - (\varphi + \mu)T]$.

$= (1 - \frac{S^*}{S})[\pi - (\lambda_H + \mu)S] + (1 - \frac{H^*}{H})[\lambda_H S - (\delta + \omega_1 + \mu)H] + (1 - \frac{A^*}{A})[\delta H + \varphi T - (\omega_2 + \mu + d_2)A] + (1 - \frac{T^*}{T})[\omega_1 H + \omega_2 A - (\varphi + \mu)T]$.

$= [\pi - (\lambda_H + \mu)S] - \frac{S^*}{S}[\pi - (\lambda_H + \mu)S] + [\lambda_H S - (\delta + \omega_1 + \mu)H] - \frac{H^*}{H}[\lambda_H S - (\delta + \omega_1 + \mu)H] + [\delta H + \varphi T - (\omega_2 + \mu + d_2)A] - \frac{A^*}{A}[\delta H + \varphi T - (\omega_2 + \mu + d_2)A] + [\omega_1 H + \omega_2 A - (\varphi + \mu)T] - \frac{T^*}{T}[\omega_1 H + \omega_2 A - (\varphi + \mu)T]$.

$= [\pi + S^*(\lambda_H + \mu) + \lambda_H S + H^*(\delta + \omega_1 + \mu) + \delta H + \varphi T + A^*(\omega_2 + \mu + d_2) + \omega_1 H + \omega_2 A + T^*(\varphi + \mu)] - [(\lambda_H + \mu)S + \frac{S^*}{S}\pi + (\delta + \omega_1 + \mu)H + \frac{H^*}{H}\lambda_H S + (\omega_2 + \mu + d_2)A + \frac{A^*}{A}(\delta H + \varphi T) + (\varphi + \mu)T + \frac{T^*}{T}(\omega_1 H + \omega_2 A)]$.

Thus, $G' = \frac{dG}{dt} = X - Y$, where

$X = [\pi + S^*(\lambda_H + \mu) + \lambda_H S + H^*(\delta + \omega_1 + \mu) + \delta H + \varphi T + A^*(\omega_2 + \mu + d_2) + \omega_1 H + \omega_2 A + T^*(\varphi + \mu)]$ and $Y = [(\lambda_H + \mu)S + \frac{S^*}{S}\pi + (\delta + \omega_1 + \mu)H + \frac{H^*}{H}\lambda_H S + (\omega_2 + \mu + d_2)A + \frac{A^*}{A}(\delta H + \varphi T) + (\varphi + \mu)T + \frac{T^*}{T}(\omega_1 H + \omega_2 A)]$.

Here $X$ and $y$ are positive, the derivative $\frac{dG}{dt} = X - Y < 0$, when $< Y$ and $\frac{dG}{dt} = 0$, when $S = S^*$, $H = H^*$, $A = A^*$, and $T = T^*$ in $\Omega$.

Hence, the largest compact invariant set in $\{(S^*, H^*, A^*, T^*) \in \Omega : \frac{dG}{dt} = 0\}$ is the singleton EE point $E_1$. By LaSalle's invariant principle [60], the EE point $E_1$ is GAS in $\Omega$ for $R_H > 1$.

The epidemiological interpretation of Theorem (3) says that there is at least a certain number of infected population if the new infection generated by an individual during his/her infection period is more than one.

## 3.4 TB-only model

The sub-model of (2) with no HIV/AIDS disease, that is, $H_I$, $H_E$, $H$, $A$, $H_L$, $T$, $A_L$, $A_I = 0$, is expressed by:

$$\begin{cases} \frac{dS}{dt} = \pi - (\lambda_T + \mu)S, \\ \frac{dE}{dt} = \lambda_T S + \gamma p I + \sigma L - (k + \alpha + \mu)E, \\ \frac{dI}{dt} = kE - (\gamma + d_1 + \mu)I, \\ \frac{dL}{dt} = (1-p)\gamma I + \alpha E - (\sigma + \mu)L, \end{cases} \quad (24)$$

where $\lambda_T = \frac{\beta_1 I(t)}{N(t)}$ and $N(t) = S(t) + E(t) + I(t) + L(t)$.

The model (24) was formulated and analysed in [46]. The basic reproduction number of this model is calculated by the usual approach.

$R_T = \beta_1 \left[ \frac{k(\sigma + \mu)}{(k+\alpha+\mu)(\gamma+d_1+\mu)(\sigma+\mu) - kp\gamma\mu - \alpha\sigma(\gamma+d_1+\mu) - k\sigma\gamma} \right]$. Moreover, the existence, uniqueness, and stability of equilibra point are proven in [61].

## 3.5 Analysis of the full model

In this section, we analysed the full HIV-TB co-infection model (2). The DFE point is calculated and is given by:

$$\varepsilon_0 = (S^0, E^0, I^0, L^0, H^0, A^0, H_E^0, H_I^0, H_L^0, T^0, A_L^0, A_I^0) = \left( \frac{\pi}{\mu}, 0, 0, 0, 0, 0, 0, 0, 0, 0, 0, 0 \right). \quad (25)$$

The associated matrices $F$ and $V$ are expressed as follows.

$F = \begin{bmatrix} F_1 & F_2 \end{bmatrix}$, where

$$F_1 = \begin{bmatrix} 0 & 0 & 0 & 0 & 0 \\ 0 & \frac{\beta_1 S}{N} & 0 & 0 & 0 \\ 0 & 0 & 0 & 0 & 0 \\ 0 & 0 & 0 & 0 & 0 \\ 0 & 0 & \lambda_H & \frac{\beta_2(S+L)}{N} & \frac{\beta_2\eta(S+L)}{N} \\ 0 & 0 & 0 & 0 & 0 \\ \epsilon_1\lambda_H & 0 & 0 & \theta\lambda_T & 0 \\ 0 & \epsilon_2\lambda_H & 0 & 0 & 0 \\ 0 & 0 & 0 & 0 & 0 \\ 0 & 0 & 0 & 0 & 0 \\ 0 & \frac{\beta_1 A}{N} & 0 & 0 & \omega\lambda_T \end{bmatrix}, F_2 = \begin{bmatrix} 0 & 0 & 0 & 0 & 0 & 0 \\ 0 & \frac{\beta_1 S}{N} & 0 & 0 & 0 & \frac{\beta_1 S}{N} \\ 0 & 0 & 0 & 0 & 0 & 0 \\ 0 & 0 & 0 & 0 & 0 & 0 \\ \frac{\beta_2(S+L)}{N} & \frac{\beta_2(S+L)}{N} & \frac{\beta_2(S+L)}{N} & 0 & \eta\frac{\beta_2(S+L)}{N} & 0 \\ 0 & 0 & 0 & 0 & 0 & 0 \\ 0 & 0 & 0 & 0 & 0 & 0 \\ 0 & 0 & 0 & 0 & 0 & 0 \\ 0 & 0 & 0 & 0 & 0 & 0 \\ 0 & 0 & 0 & 0 & 0 & 0 \\ 0 & 0 & 0 & 0 & 0 & 0 \\ 0 & \frac{\beta_1 A}{N} & 0 & 0 & 0 & \frac{\beta_1 A}{N} \end{bmatrix}. \quad (26)$$

The matrix $V = [\; V_1 \quad V_2 \;]$, where

$$
V_1 = \begin{bmatrix}
\lambda_T + \lambda_T + \mu & 0 & \frac{\beta_1 S}{N} & 0 & \frac{\beta_2 S}{N} & \eta \frac{\beta_2 S}{N} \\
0 & (k + \alpha + \epsilon_1 \lambda_H + \mu) & -\gamma p & -\sigma & 0 & 0 \\
0 & -k & (\gamma \epsilon_2 \lambda_H + d_1 + \mu) & 0 & \frac{\epsilon_2 \beta_2 I}{N} & \eta \frac{\epsilon_2 \beta_2 I}{N} \\
0 & -\alpha & -(1-p)\gamma & \sigma + \mu + \lambda_H & \frac{\beta_2 L}{N} & \frac{\beta_2 L}{N} \\
0 & 0 & \theta \frac{\beta_1 H}{N} & 0 & \theta \lambda_T + \delta + \omega_1 + \mu & 0 \\
0 & 0 & \frac{\omega \beta_1 A}{N} & 0 & -\delta & U \\
0 & 0 & 0 & 0 & 0 & 0 \\
0 & 0 & 0 & 0 & 0 & 0 \\
0 & 0 & 0 & 0 & -\omega_1 & -\omega_2 \\
0 & 0 & 0 & 0 & 0 & 0 \\
0 & 0 & 0 & 0 & 0 & 0
\end{bmatrix}, \quad (27)
$$

$$
V_2 = \begin{bmatrix}
\frac{\beta_2 S}{N} & \frac{\beta_2 S}{N} & \frac{\beta_2 S}{N} & 0 & \eta \frac{\beta_2 S}{N} & 0 \\
0 & 0 & 0 & 0 & 0 & 0 \\
\epsilon_2 \frac{\beta_2 I}{N} & \epsilon_2 \frac{\beta_2 I}{N} & \epsilon_2 \frac{\beta_2 I}{N} & 0 & \eta \epsilon_2 \frac{\beta_2 I}{N} & 0 \\
\frac{\beta_2 L}{N} & \frac{\beta_2 L}{N} & \frac{\beta_2 L}{N} & 0 & \eta \frac{\beta_2 L}{N} & 0 \\
0 & \theta \frac{\beta_1 H}{N} & 0 & 0 & 0 & \theta \frac{\beta_1 H}{N} \\
0 & \frac{\omega \beta_1 A}{N} & 0 & -\varphi & 0 & \frac{\omega \beta_1 A}{N} \\
M & -\psi \gamma & -\theta_1 & 0 & 0 & 0 \\
-\epsilon & P & 0 & 0 & 0 & 0 \\
-\theta_2 & (1-\psi)\gamma \phi & -(\theta_1 + \theta_3 + \omega_5 + \mu) & 0 & 0 & 0 \\
-\omega_3 & -\omega_4(1-\phi) & -\omega_5 & (\varphi + \mu) & -\omega_6 & -\omega_7 \\
0 & 0 & -\theta_3 & 0 & (\omega_6 + d_2 + \mu) & -\tau \\
-(\sigma_1 + \delta_1) & -\psi_1 & 0 & 0 & 0 & (\omega_7 + \tau + d_3 + \mu)
\end{bmatrix}. \quad (28)
$$

The letters $M$ and $P$ in the matrix represented as: $M = \epsilon + \omega_3 + \theta_2 + \sigma_1 + \delta_1 + \mu$ and $P = \gamma(\psi + \phi - \psi\phi) + \omega_4(1 - \phi) + \psi_1 + \mu + d_1$.

The spectral radius of the matrix $FV^{-1}$ are:

$$R_1 = \frac{\beta_2}{A}\left[1 + \frac{\eta(\delta C + \omega_1 \varphi)}{CB - \varphi\omega_2}\right] = R_H \qquad \text{and}$$

$$R_2 = \beta_1 \left[\frac{k(\sigma + \mu)}{(k+\alpha+\mu)(\gamma+d_1+\mu)(\sigma+\mu) - kp\gamma\mu - \alpha\sigma(\gamma+d_1+\mu) - k\sigma\gamma}\right] = R_T.$$

(29)

Hence, the basic reproduction number of (2) is expressed by:
$R_0 = max\{R_H, R_T\}$ is justified in [56].

## 3.6 Local stability of DFE point

**Theorem 4** *The DFE of the full HIV-TB model (2) is LAS if $R_0 < 1$, and unstable if $R_0 > 1$.*

**Poof.**

The Jacobian matrix of the model at the DFE point is;
$J\left(\frac{\pi}{\mu}, 0,0,0,0,0,0,0,0,0,0,0\right) = \begin{bmatrix} J_1 & J_2 \end{bmatrix}$, with

$$J_1(\varepsilon_0) = \begin{bmatrix}
-\mu & 0 & -\beta_1\frac{\pi}{\mu N(t)} & 0 & -\beta_2\frac{\pi}{\mu N(t)} & -\beta_2\eta\frac{\pi}{\mu N(t)} \\
0 & -(k+\alpha+\mu) & \gamma p + \beta_1\frac{\pi}{\mu N(t)} & \sigma & 0 & 0 \\
0 & k & -(\gamma+d_1+\mu) & 0 & 0 & 0 \\
0 & \alpha & (1-p)\gamma & -(\sigma+\mu) & 0 & 0 \\
0 & 0 & 0 & 0 & \beta_2\frac{\pi}{\mu N(t)} - (\delta+\omega_1+\mu) & \beta_2\eta\frac{\pi}{\mu N(t)} \\
0 & 0 & 0 & 0 & \delta & -(d_2+\omega_2+\mu) \\
0 & 0 & 0 & 0 & 0 & 0 \\
0 & 0 & 0 & 0 & 0 & 0 \\
0 & 0 & 0 & 0 & 0 & 0 \\
0 & 0 & 0 & 0 & \omega_1 & \omega_2 \\
0 & 0 & 0 & 0 & 0 & 0 \\
0 & 0 & 0 & 0 & 0 & 0
\end{bmatrix},$$

(30)

$$J_2(\varepsilon_0) = \begin{bmatrix}
-\beta_2\frac{\pi}{\mu N(t)} & -(\beta_2+\beta_1)\frac{\pi}{\mu N(t)} & -\beta_2\frac{\pi}{\mu N(t)} & 0 & -\beta_2\eta\frac{\pi}{\mu N(t)} & -\beta_1\frac{\pi}{\mu N(t)} \\
0 & \beta_1\frac{\pi}{\mu N(t)} & 0 & 0 & 0 & \beta_1\frac{\pi}{\mu N(t)} \\
0 & 0 & 0 & 0 & 0 & 0 \\
0 & 0 & 0 & 0 & 0 & 0 \\
\beta_2\frac{\pi}{\mu N(t)} & \beta_2\frac{\pi}{\mu N(t)} & \beta_2\frac{\pi}{\mu N(t)} & 0 & \eta\beta_2\frac{\pi}{\mu N(t)} & 0 \\
0 & 0 & 0 & \varphi & 0 & 0 \\
-M & \psi\gamma & \theta_1 & 0 & 0 & 0 \\
\epsilon & -P & 0 & 0 & 0 & 0 \\
\theta_2 & (1-\psi)\phi\gamma & -(\theta_1+\theta_3+\omega_5+\mu) & 0 & 0 & 0 \\
\omega_3 & \omega_4(1-\phi) & \omega_5 & -(\varphi+\mu) & \omega_6 & \omega_7 \\
0 & 0 & \theta_3 & 0 & -(\omega_6+d_2+\mu) & \tau \\
(\sigma_1+\delta_1) & \psi_1 & 0 & 0 & 0 & -(\omega_7+d_3+\tau+\mu)
\end{bmatrix}.$$

Afterwards, we get the eigenvalues of $J$ likes below.

$$
\begin{vmatrix}
-\mu - \lambda & 0 & -\beta_1 \frac{\pi}{\mu N(t)} & 0 & -\beta_2 \frac{\pi}{\mu N(t)} & -\beta_2 \eta \frac{\pi}{\mu N(t)} \\
0 & -(k+\alpha+\mu)-\lambda & \gamma p + \beta_1 \frac{\pi}{\mu N(t)} & \sigma & 0 & 0 \\
0 & k & -(\gamma+d_1+\mu)-\lambda & 0 & 0 & 0 \\
0 & \alpha & (1-p)\gamma & -(\sigma+\mu)-\lambda & 0 & 0 \\
0 & 0 & 0 & 0 & \beta_2\frac{\pi}{\mu N(t)}-A-\lambda & \beta_2\eta\frac{\pi}{\mu N(t)} \\
0 & 0 & 0 & 0 & \delta & -B-\lambda \\
0 & 0 & 0 & 0 & 0 & 0 \\
0 & 0 & 0 & 0 & 0 & 0 \\
0 & 0 & 0 & 0 & 0 & 0 \\
0 & 0 & 0 & 0 & \omega_1 & \omega_2 \\
0 & 0 & 0 & 0 & 0 & 0 \\
0 & 0 & 0 & 0 & 0 & 0
\end{vmatrix}
$$

$$
\begin{vmatrix}
-\beta_2\frac{\pi}{\mu N(t)} & -(\beta_2+\beta_1)\frac{\pi}{\mu N(t)} & -\beta_2\frac{\pi}{\mu N(t)} & 0 & -\beta_2\eta\frac{\pi}{\mu N(t)} & -\beta_1\frac{\pi}{\mu N(t)} \\
0 & \beta_1\frac{\pi}{\mu N(t)} & 0 & 0 & 0 & \beta_1\frac{\pi}{\mu N(t)} \\
0 & 0 & 0 & 0 & 0 & 0 \\
0 & 0 & 0 & 0 & 0 & 0 \\
\beta_2\frac{\pi}{\mu N(t)} & \beta_2\frac{\pi}{\mu N(t)} & \beta_2\frac{\pi}{\mu N(t)} & 0 & \eta\beta_2\frac{\pi}{\mu N(t)} & 0 \\
0 & 0 & 0 & \varphi & 0 & 0 \\
-M-\lambda & \psi\gamma & \theta_1 & 0 & 0 & 0 \\
\epsilon & -P-\lambda & 0 & 0 & 0 & 0 \\
\theta_2 & (1-\psi)\phi\gamma & -(\theta_1+\theta_3+\omega_5+\mu)-\lambda & 0 & 0 & 0 \\
\omega_3 & \omega_4(1-\phi) & \omega_5 & -C-\lambda & \omega_6 & \omega_7 \\
0 & 0 & \theta_3 & 0 & -(\omega_6+d_2+\mu)-\lambda & \tau \\
(\sigma_1+\delta_1) & \psi_1 & 0 & 0 & 0 & -(\omega_7+d_3+\tau+\mu)-\lambda
\end{vmatrix} = 0.
$$

(31)

After huge calculations, we get the following result.

$$(\mu + \lambda)(J + \lambda)(H + \lambda)[\epsilon\psi\gamma(G + \lambda) + \epsilon\theta_1(1 - \psi)\phi\gamma + (P + \lambda)(\theta_1\theta_2 - (M + \lambda)(G + \lambda))]$$

$$[\beta_2 D - A - \lambda)((B + \lambda)(C + \lambda) - \varphi\omega_2) + \beta_2\eta D(\delta(C + \lambda) + \varphi\omega_1)]$$

$$[k((\gamma p + \beta_1 D)(\sigma + \mu + \lambda) + \sigma(1 - p)\gamma) - (F + \lambda)((E + \lambda)(\sigma + \mu + \lambda) - \alpha\sigma)] = 0, \quad (32)$$

where $A = \delta + \omega_1 + \mu, B = \omega_2 + d_2 + \mu, C = \varphi + \mu, D = \dfrac{\pi}{\mu N}, E = k + \alpha + \mu, F = \gamma + d_1 + \mu,$

$G = \theta_1 + \theta_3 + \omega_5 + \mu, H = \omega_6 + d_2 + \mu, J = \omega_7 + d_3 + \tau + \mu.$

The equation becomes: $(\mu + \lambda)(J + \lambda)(H + \lambda) = 0$ implies $\lambda = -\mu < 0$ or $\lambda = -J < 0$ or $\lambda = -H < 0$, $[\epsilon\psi\gamma(G + \lambda) + \epsilon\theta_1(1 - \psi)\phi\gamma + (P + \lambda)(\theta_1\theta_2 - (M + \lambda)(G + \lambda))] = 0$, $[(\beta_2 D - A - \lambda)((B + \lambda)(C + \lambda) - \varphi\omega_2) + \beta_2\eta D(\delta(C + \lambda) + \varphi\omega_1)] = 0$, and $[k((\gamma p + \beta_1 D)(\sigma + \mu + \lambda) + \sigma(1 - p)\gamma) - (F + \lambda)((E + \lambda)(\sigma + \mu + \lambda) - \alpha\sigma)] = 0$.

Hence

$$[(\beta_2 D - A - \lambda)((B + \lambda)(C + \lambda) - \varphi\omega_2) + \beta_2\eta D(\delta(C + \lambda) + \varphi\omega_1)] = 0. \quad (33)$$

The Eq (33) is derived as:

$$\frac{1}{A(BC - \varphi\omega_2)}[\lambda^3 + \lambda^2\left(\frac{\mu NA + (B + C)\mu N - \beta_2\pi}{\mu N}\right) +$$

$$\lambda\left(\frac{\mu N(BC - \varphi\omega_2) + (B + C)\mu NA - \beta_2\pi(B + C + \eta\delta)}{\mu N}\right)] + (1 - R_H) = 0. \quad (34)$$

Again,

$$[k((\gamma p + \beta_1 D)(\sigma + \mu + \lambda) + \sigma(1 - p)\gamma) - (F + \lambda)((E + \lambda)(\sigma + \mu + \lambda) - \alpha\sigma)] = 0. \quad (35)$$

Thus, the Eq (35) is expressed as:

$$\frac{1}{(k + \alpha + \mu)(\gamma + d_1 + \mu)(\sigma + \mu) - kp\gamma\mu - \alpha\sigma(\gamma + d_1 + \mu) - k\sigma\gamma}[\lambda^3 +$$

$$\lambda^2(\alpha + \sigma + k + \gamma + d_1 + 3\mu) + \lambda(\alpha + k + \gamma + d_1 + 2\mu)(\sigma + \mu) - kp\gamma - \frac{\alpha\sigma\beta_1\pi}{\mu N}] + (1 - R_T) = 0. \quad (36)$$

Applying the Routh-Hurwitz stability criteria, the two polynomial expressions (34) and (36) have roots called eigenvalues. The eigenvalues have negative real part if and only if the two constant terms $(1 - R_H) > 0$ and $(1 - R_T) > 0$. Therefore, $R_H < 1$ and $R_T < 1$ gives $R_0 < 1$, this completed the proof.

## 3.7 Global stability of DFE point

**Theorem 5** *The fixed point $U_0 = (X^*, 0)$ is GAS, if $R_0 < 1$ (LAS) and the two conditions $(H_1)$ and $(H_2)$ are satisfied.*

We explored the theorem using the technique in [57]. The model (2) can be expressed by: $\frac{dX}{dt} = F(X, Z), \frac{dZ}{dt} = G(X, Z), G(X, 0) = 0$, where the vectors $X$ and $Z$ are representing the uninfected and infected compartments.

So, $X = S, Z = (E, L, T, I, H, A, H_E, H_I, H_L, A_L, A_I)$ and the conditions $(H_1)$ and $(H_2)$ are:

$(H_1), \frac{dX}{dt} = F(X, 0), X^*$ is GAS.

$(H_2)$, $\frac{dZ}{dt} = QZ - G^*(X,Z)$, where $G^*(X,Z) \geq 0$ for $(X,Z) \in R_{12}^+$ and $Q$ is a Metzler matrix (the non diagonal entries of $Q$ are non-negative). Hence, $Q = [\, Q_a \quad Q_b \,]$, where

$$
Q_a = \begin{bmatrix}
-(k+\alpha+\mu) & \gamma p + \beta_1 \frac{\pi}{\mu N(t)} & \sigma & 0 & 0 \\
k & -(\gamma+d_1+\mu) & 0 & 0 & 0 \\
\alpha & (1-p)\gamma & -(\sigma+\mu) & 0 & 0 \\
0 & 0 & 0 & \beta_2 \frac{\pi}{\mu N(t)} - (\delta+\omega_1+\mu) & \beta_2 \eta \frac{\pi}{\mu N(t)} \\
0 & 0 & 0 & \delta & -(d_2+\omega_2+\mu) \\
0 & 0 & 0 & 0 & 0 \\
0 & 0 & 0 & 0 & 0 \\
0 & 0 & 0 & 0 & 0 \\
0 & 0 & 0 & \omega_1 & \omega_2 \\
0 & 0 & 0 & 0 & 0 \\
0 & 0 & 0 & 0 & 0
\end{bmatrix}, \quad \text{and}
$$

$$
\tag{37}
$$

$$
Q_b = \begin{bmatrix}
0 & \beta_1 \frac{\pi}{\mu N(t)} & 0 & 0 & 0 & \beta_1 \frac{\pi}{\mu N(t)} \\
0 & 0 & 0 & 0 & 0 & 0 \\
0 & 0 & 0 & 0 & 0 & 0 \\
\beta_2 \frac{\pi}{\mu N(t)} & \beta_2 \frac{\pi}{\mu N(t)} & \beta_2 \frac{\pi}{\mu N(t)} & 0 & \eta\beta_2 \frac{\pi}{\mu N(t)} & 0 \\
0 & 0 & 0 & \varphi & 0 & 0 \\
-M & \psi\gamma & \theta_1 & 0 & 0 & 0 \\
\epsilon & -P & 0 & 0 & 0 & 0 \\
\theta_2 & (1-\psi)\phi\gamma & -(\theta_1+\theta_3+\omega_5+\mu) & 0 & 0 & 0 \\
\omega_3 & \omega_4(1-\phi) & \omega_5 & -(\varphi+\mu) & \omega_6 & \omega_7 \\
0 & 0 & \theta_3 & 0 & -(\omega_6+d_2+\mu) & \tau \\
(\sigma_1+\delta_1) & \psi_1 & 0 & 0 & 0 & -(\omega_7+d_3+\tau+\mu)
\end{bmatrix}.
$$

The non diagonal entries of $Q$ are non-negative.

$G(X, Z) = QZ - G^*(X, Z)$, where

$$G^*(X, Z) = \begin{bmatrix} \beta_1\left(1 - \frac{S}{N(t)}\right)[I + H_I + A_I] \\ 0 \\ 0 \\ \beta_2\left(1 - \frac{S}{N(t)}\right)[H + H_E + H_L + H_I + \eta A_I + \eta A_L] \\ 0 \\ 0 \\ 0 \\ 0 \\ 0 \\ 0 \\ 0 \\ 0 \end{bmatrix}. \tag{38}$$

Since $0 \le S \le N$, then $G^*(X, Z) \ge 0$ and the model (2) is globally asymptotically stable.

## 3.8 EE point of HIV-TB model

The EE point of (2) occurs when TB and HIV/AIDS co-infection persist in the community. This can be calculated by the following way.

$$\begin{cases} \frac{dS}{dt} = \pi - (\lambda_H + \lambda_T + \mu)S = 0, \\ \frac{dE}{dt} = \lambda_T S + \gamma p I + \sigma L - (k + \alpha + \epsilon_1\lambda_H + \mu)E = 0, \\ \frac{dI}{dt} = kE - (\gamma + \epsilon_2\lambda_H + d_1 + \mu)I = 0, \\ \frac{dL}{dt} = (1 - p)\gamma I + \alpha E - (\sigma + \lambda_H + \mu)L = 0, \\ \frac{dH}{dt} = \lambda_H(S + L) - (\theta\lambda_T + \delta + \omega_1 + \mu)H = 0, \\ \frac{dA}{dt} = \delta H + \varphi T - (\omega_2 + \omega\lambda_T + \mu + d_2)A = 0, \\ \frac{dH_E}{dt} = \epsilon_1\lambda_H E + \theta\lambda_T H + \psi\gamma H_I + \theta_1 H_L - (\epsilon + \omega_3 + \theta_2 + \sigma_1 + \delta_1 + \mu)H_E = 0, \\ \frac{dH_I}{dt} = \epsilon_2\lambda_H I + \epsilon H_E - (\psi\gamma + (1 - \psi)\phi\gamma + (1 - \varphi)\omega_4 + \psi_1 + \mu + d_1)H_I = 0, \\ \frac{dH_L}{dt} = (1 - \psi)\phi\gamma H_I + \theta_2 H_E - (\theta_1 + \theta_3 + \omega_5 + \mu)H_L = 0, \\ \frac{dT}{dt} = \omega_1 H + \omega_2 A + \omega_3 H_E + \omega_4(1 - \varphi)H_I + \omega_5 H_L + \omega_6 A_L + \omega_7 A_I - (\varphi + \mu)T = 0, \\ \frac{dA_L}{dt} = \tau A_I + \theta_3 H_L - (\omega_6 + \mu + d_2)A_L = 0, \\ \frac{dA_I}{dt} = (\sigma_1 + \delta_1)H_E + \psi_1 H_I + \omega\lambda_T A - (\omega_7 + \mu + d_3 + \tau)A_I = 0. \end{cases} \tag{39}$$

Let $\lambda_{T*} = \frac{\beta_1[I_* + H_{I*} + A_{I*}]}{N}$ and $\lambda_{H*} = \frac{\beta_2[H_* + H_{E*} + H_{L*} + H_{I*} + \eta(A_* + A_{L*})]}{N}$, then we get

$$S_* = \frac{\pi}{(\lambda_{T*} + \lambda_{H*} + \mu)}, \; E_* = \frac{\lambda_{T*}S_* + \gamma pI_* + \sigma L_*}{k + \alpha + \epsilon_1\lambda_{H*} + \mu}, \; I_* = k\frac{E_*}{\gamma + \epsilon_2\lambda_{H*} + d_1 + \mu}, \; L_* = \frac{(1-p)\gamma I_* + \alpha E_*}{\sigma + \lambda_{H*} + \mu}, \; H_* = \frac{\lambda_{H*}(S_* + L_*)}{\theta\lambda_{T*} + \delta + \omega_1 + \mu},$$

$$A_* = \frac{\delta H_* + \varphi T_*}{\omega_2 + \omega\lambda_{T*} + \mu + d_2}, \; H_{E*} = \frac{\epsilon_1\lambda_{H*}E_* + \theta\lambda_{T*}H + \psi\gamma H_{I*} + \theta_1 H_{L*}}{\epsilon + \omega_3 + \theta_2 + \sigma_1 + \delta_1 + \mu}, \; H_{I*} = \frac{\epsilon_2\lambda_{H*}I_* + \epsilon H_{E*}}{\psi\gamma + (1-\psi)\varphi\gamma + (1-\varphi)\omega_4 + \psi_1 + \mu + d_1},$$

$$H_{L*} = \frac{(1-\psi)\varphi\gamma H_{I*} + \theta_2 H_{E*}}{\theta_1 + \theta_3 + \omega_5 + \mu}, \; T_* = \frac{\omega_1 H_* + \omega_2 A_* + \omega_3 H_{E*} + \omega_4(1-\varphi)H_{I*} + \omega_5 H_{L*} + \omega_6 A_{L*} + \omega_7 A_{I*}}{\varphi + \mu}, \; A_{L*} = \frac{\tau A_{I*} + \theta_3 H_{L*}}{\omega_6 + \mu + d_2},$$

$$A_{I*} = \frac{(\sigma_1 + \delta_1)H_{E*} + \psi_1 H_{I*} + \omega\lambda_{T*}A_*}{\omega_7 + \mu + d_3 + \tau}.$$

Thus, the EE point of HIV-TB co-epidemic model is symbolized by:

$E^* = (S_*, E_*, I_*, L_*, H_*, H_{E*}, H_{L*}, H_{I*}, T_*, A_*, A_{L*}, A_{I*}).$

**Lemma 3** *A unique endemic equilibrium point $E_*$ exist if $R_0 > 1$.*

**Proof.** If the disease is exists in the society, So, the model equation (2) becomes:

$$\begin{cases} \frac{dE}{dt} = \lambda_T S + \gamma pI + \sigma L - (k + \alpha + \epsilon_1\lambda_H + \mu)E > 0, \\[2mm] \frac{dI}{dt} = kE - (\gamma + \epsilon_2\lambda_H + d_1 + \mu)I > 0, \\[2mm] \frac{dL}{dt} = (1-p)\gamma I + \alpha E - (\sigma + \lambda_H + \mu)L > 0, \\[2mm] \frac{dH}{dt} = \lambda_H(S + L) - (\theta\lambda_T + \delta + \omega_1 + \mu)H > 0, \\[2mm] \frac{dA}{dt} = \delta H + \varphi T - (\omega_2 + \omega\lambda_T + \mu + d_2)A > 0, \\[2mm] \frac{dH_E}{dt} = \epsilon_1\lambda_H E + \theta\lambda_T H + \psi\gamma H_I + \theta_1 H_L - (\epsilon + \omega_3 + \theta_2 + \sigma_1 + \delta_1 + \mu)H_E > 0, \\[2mm] \frac{dH_I}{dt} = \epsilon_2\lambda_H I + \epsilon H_E - (\psi\gamma + (1-\psi)\varphi\gamma + (1-\phi)\omega_4 + \psi_1 + \mu + d_1)H_I > 0, \\[2mm] \frac{dH_L}{dt} = (1-\psi)\phi\gamma H_I + \theta_2 H_E - (\theta_1 + \theta_3 + \omega_5 + \mu)H_L > 0, \\[2mm] \frac{dT}{dt} = \omega_1 H + \omega_2 A + \omega_3 H_E + \omega_4(1-\phi)H_I + \omega_5 H_L + \omega_6 A_L + \omega_7 A_I - (\varphi + \mu)T > 0, \\[2mm] \frac{dA_L}{dt} = \tau A_I + \theta_3 H_L - (\omega_6 + \mu + d_2)A_L > 0, \\[2mm] \frac{dA_I}{dt} = (\sigma_1 + \delta_1)H_E + \psi_1 H_I + \omega\lambda_T A - (\omega_7 + \mu + d_3 + \tau)A_I > 0. \end{cases} \quad (40)$$

Now, from the first three inequalities of (40), we have

$$\begin{cases} (k + \alpha + \epsilon_1\lambda_H + \mu)E < \lambda_T S + \gamma pI + \sigma L, \\[2mm] (\gamma + \epsilon_2\lambda_H + d_1 + \mu)I < kE, \\[2mm] (\sigma + \lambda_H + \mu)L < (1-p)\gamma I + \alpha E. \end{cases} \quad (41)$$

From the fact that $\frac{S}{N(t)} \leq 1$. Thus, the system (41) gives:

$$\begin{cases} (k + \alpha + \mu)E < \beta_1 I + \gamma pI + \sigma L, \\[2mm] (\gamma + d_1 + \mu)I < kE, \\[2mm] (\sigma + \mu)L < (1-p)\gamma I + \alpha E. \end{cases} \quad (42)$$

Next, adding the first and the third inequalities of (42) simultaneously to eliminate the term $L$, we get the following result.

$$\begin{cases} ((k + \alpha + \mu)(\sigma + \mu) - \alpha\sigma)E < ((\beta_1 + \gamma p)(\sigma + \mu) + \sigma(1-p)\gamma)I, \\[2mm] (\gamma + d_1 + \mu)I < kE. \end{cases} \quad (43)$$

Thus, the system (43) gives:

$$(k + \alpha + \mu)(\gamma + d_1 + \mu)(\sigma + \mu) - \alpha\sigma(\gamma + d_1 + \mu) - k\sigma\gamma + k\sigma p\gamma - kp\gamma\mu - k\alpha\gamma p < k\beta_1(\sigma + \mu),$$

$$(k + \alpha + \mu)(\gamma + d_1 + \mu)(\sigma + \mu) - \alpha\sigma(\gamma + d_1 + \mu) - k\sigma\gamma - kp\gamma\mu - k\alpha\gamma p < k\beta_1(\sigma + \mu),$$

$$1 < \frac{k\beta_1(\sigma + \mu)}{(k + \alpha + \mu)(\gamma + d_1 + \mu)(\sigma + \mu) - \alpha\sigma(\gamma + d_1 + \mu) - k\sigma\gamma - kp\gamma\mu - k\alpha\gamma p} = R_T,$$

$$R_T > 1.$$

(44)

Similarly, the other inequalities in the system (40) verified the condition $R_H > 1$. This indicated that a unique EE point exists if $R_0 = max\{R_H, R_T\} > 1$.

### 3.9 Impact of HIV/AIDS on TB disease

To explore the impact of HIV/AIDS on TB disease and vase versa, we described $R_T$ interims of $R_H$.

Hence, $R_H = \frac{\beta_2\pi}{\mu A}\left[1 + \frac{\eta(\delta C + \omega_1\varphi)}{CB - \varphi\omega_2}\right] \Rightarrow \mu = \frac{\beta_2\pi}{R_H A}\left[1 + \frac{\eta(\delta C + \omega_1\varphi)}{CB - \varphi\omega_2}\right]$.

Then, substituting equation $\mu$ in $R_T$ gives:

$$R_T = \frac{\beta_1\pi}{\frac{\beta_2\pi}{R_H A}[1 + \frac{\eta(\delta C + \omega_1\varphi)}{CB - \varphi\omega_2}]}\left[\frac{k(\sigma + \mu)}{(k + \alpha + \mu)(\gamma + d_1 + \mu)(\sigma + \mu) - kp\gamma\mu - \alpha\sigma(\gamma + d_1 + \mu) - k\sigma\gamma}\right].$$

$$= \frac{\beta_1 R_H A(CB - \varphi\omega_2)}{\beta_2[CB - \varphi\omega_2 + \eta(\delta C + \omega_1\varphi)]}\left[\frac{k(\sigma + \mu)}{(k + \alpha + \mu)(\gamma + d_1 + \mu)(\sigma + \mu) - kp\gamma\mu - \alpha\sigma(\gamma + d_1 + \mu) - k\sigma\gamma}\right].$$

Now, we did the impact of the two diseases interaction by:

$$\frac{\partial R_T}{\partial R_H} = \frac{k\beta_1 A(CB - \varphi\omega_2)(\sigma + \mu)}{\beta_2[CB - \varphi\omega_2 + \eta(\delta C + \omega_1\varphi)][(k + \alpha + \mu)(\gamma + d_1 + \mu)(\sigma + \mu) - kp\gamma\mu - \alpha\sigma(\gamma + d_1 + \mu) - k\sigma\gamma]} > 0.$$

(45)

Here, the Eq (45) displays that HIV/AIDS disease accelerates the rate of infection of TB cases and vice versa.

### 3.10 Bifurcation analysis

In order to discuss the nature of bifurcation at the threshold value $R_0 = 1$, we used the central manifold theory [62]. To apply the technique, the next shifts of variables are made.

Let $S = x_1, E = x_2, I = x_3, L = x_4, H = x_5, A = x_6, H_E = x_7, H_I = x_8, H_L = x_9, T = x_{10}, A_L = x_{11},$ and $A_I = x_{12}$.

Thus, the system (2) becomes:

$$
\begin{cases}
\frac{dx_1}{dt} = \pi - (\lambda_H + \lambda_T + \mu)x_1, \\[4pt]
\frac{dx_2}{dt} = \lambda_T x_1 + \gamma p x_3 + \sigma x_4 - (k + \alpha + \epsilon_1 \lambda_H + \mu)x_2, \\[4pt]
\frac{dx_3}{dt} = k x_2 - (\gamma + \epsilon_2 \lambda_H + d_1 + \mu)x_3, \\[4pt]
\frac{dx_4}{dt} = (1 - p)\gamma x_3 + \alpha x_2 - (\sigma + \lambda_H + \mu)x_4, \\[4pt]
\frac{dx_5}{dt} = \lambda_H(x_1 + x_4) - (\theta\lambda_T + \delta + \omega_1 + \mu)x_5, \\[4pt]
\frac{dx_6}{dt} = \delta x_3 + \varphi x_{10} - (\omega_2 + \omega\lambda_T + \mu + d_2)x_6, \\[4pt]
\frac{dx_7}{dt} = \epsilon_1 \lambda_H x_2 + \theta\lambda_T x_5 + \psi\gamma x_8 + \theta_1 x_9 - (\epsilon + \omega_3 + \theta_2 + \sigma_1 + \delta_1 + \mu)x_7, \\[4pt]
\frac{dx_8}{dt} = \epsilon_2 \lambda_H x_3 + \epsilon x_7 - (\psi\gamma + (1 - \psi)\phi\gamma + (1 - \varphi)\omega_4 + \psi_1 + \mu + d_1)x_8, \\[4pt]
\frac{dx_9}{dt} = (1 - \psi)\phi\gamma x_8 + \theta_2 x_7 - (\theta_1 + \theta_3 + \omega_5 + \mu)x_9, \\[4pt]
\frac{dx_{10}}{dt} = \omega_1 x_5 + \omega_2 x_6 + \omega_3 x_7 + \omega_4(1 - \varphi)x_8 + \omega_5 x_9 + \omega_6 x_{11} + \omega_7 x_{12} - (\varphi + \mu)x_{10}, \\[4pt]
\frac{dx_{11}}{dt} = \tau x_{12} + \theta_3 x_9 - (\omega_6 + \mu + d_2)x_{11}, \\[4pt]
\frac{dx_{12}}{dt} = (\sigma_1 + \delta_1)x_7 + \psi_1 x_8 + \omega\lambda_T x_6 - (\omega_7 + \mu + d_3 + \tau)x_{12},
\end{cases}
\tag{46}
$$

where $\lambda_T = \frac{\beta_1[x_3 + x_8 + x_{12}]}{N}$ and $\lambda_H = \frac{\beta_2[x_5 + x_7 + x_8 + x_9 + \eta(x_6 + x_{11})]}{N}$ .

The Jacobian matrix $J$ of (46) at DFE point is already articulated.

$J(\frac{\pi}{\mu}, 0, 0, 0, 0, 0, 0, 0, 0, 0, 0, 0) = [J_1 \quad J_2]$, with

$$
J_1(\varepsilon_0) =
\begin{bmatrix}
-\mu & 0 & -\beta_1 \frac{\pi}{\mu N(t)} & 0 & -\beta_2 \frac{\pi}{\mu N(t)} & -\beta_2 \eta \frac{\pi}{\mu N(t)} \\
0 & -(k + \alpha + \mu) & \gamma p + \beta_1 \frac{\pi}{\mu N(t)} & \sigma & 0 & 0 \\
0 & k & -(\gamma + d_1 + \mu) & 0 & 0 & 0 \\
0 & \alpha & (1 - p)\gamma & -(\sigma + \mu) & 0 & 0 \\
0 & 0 & 0 & 0 & \beta_2 \frac{\pi}{\mu N(t)} - (\delta + \omega_1 + \mu) & \beta_2 \eta \frac{\pi}{\mu N(t)} \\
0 & 0 & 0 & 0 & \delta & -(d_2 + \omega_2 + \mu) \\
0 & 0 & 0 & 0 & 0 & 0 \\
0 & 0 & 0 & 0 & 0 & 0 \\
0 & 0 & 0 & 0 & 0 & 0 \\
0 & 0 & 0 & 0 & \omega_1 & \omega_2 \\
0 & 0 & 0 & 0 & 0 & 0 \\
0 & 0 & 0 & 0 & 0 & 0
\end{bmatrix},
$$

$$
J_2(\varepsilon_0) = \begin{bmatrix}
-\beta_2\frac{\pi}{\mu N(t)} & -(\beta_2+\beta_1)\frac{\pi}{\mu N(t)} & -\beta_2\frac{\pi}{\mu N(t)} & 0 & -\beta_2\eta\frac{\pi}{\mu N(t)} & -\beta_1\frac{\pi}{\mu N(t)} \\
0 & \beta_1\frac{\pi}{\mu N(t)} & 0 & 0 & 0 & \beta_1\frac{\pi}{\mu N(t)} \\
0 & 0 & 0 & 0 & 0 & 0 \\
0 & 0 & 0 & 0 & 0 & 0 \\
\beta_2\frac{\pi}{\mu N(t)} & \beta_2\frac{\pi}{\mu N(t)} & \beta_2\frac{\pi}{\mu N(t)} & 0 & \eta\beta_2\frac{\pi}{\mu N(t)} & 0 \\
0 & 0 & 0 & \varphi & 0 & 0 \\
-M & \psi\gamma & \theta_1 & 0 & 0 & 0 \\
\epsilon & -P & 0 & 0 & 0 & 0 \\
\theta_2 & (1-\psi)\phi\gamma & -(\theta_1+\theta_3+\omega_5+\mu) & 0 & 0 & 0 \\
\omega_3 & \omega_4(1-\phi) & \omega_5 & -(\varphi+\mu) & \omega_6 & \omega_7 \\
0 & 0 & \theta_3 & 0 & -(\omega_6+d_2+\mu) & \tau \\
(\sigma_1+\delta_1) & \psi_1 & 0 & 0 & 0 & -(\omega_7+d_3+\tau+\mu)
\end{bmatrix}.
$$

Hereafter, we can calculate the right eigenvectors of $J(\varepsilon_0)$ symbolized by:

$u = (u_1, u_2, u_3, u_4, u_5, u_6, u_7, u_8, u_9, u_{10}, u_{11}, u_{12})^T$ corresponding to the zero eigenvalues as follows.

$$
[J_1(\varepsilon_0) \quad J_2(\varepsilon_0)] \begin{pmatrix} u_1 \\ u_2 \\ u_3 \\ u_4 \\ u_5 \\ u_6 \\ u_7 \\ u_8 \\ u_9 \\ u_{10} \\ u_{11} \\ u_{12} \end{pmatrix} = \begin{pmatrix} 0 \\ 0 \\ 0 \\ 0 \\ 0 \\ 0 \\ 0 \\ 0 \\ 0 \\ 0 \\ 0 \\ 0 \end{pmatrix}. \tag{47}
$$

The Eq (47) becomes;

$$
\begin{cases}
-\mu u_1 - \beta_1 \frac{\pi}{\mu N(t)} u_3 - \beta_2 \frac{\pi}{\mu N(t)} u_5 - \beta_2 \eta \frac{\pi}{\mu N(t)} u_6 - \beta_2 \frac{\pi}{\mu N(t)} u_7, \\[4pt]
-(\beta_2 + \beta_1) \frac{\pi}{\mu N(t)} u_8 - \beta_2 \frac{\pi}{\mu N(t)} u_9 - \beta_2 \eta \frac{\pi}{\mu N(t)} u_{11} - \beta_1 \frac{\pi}{\mu N(t)} u_{12} = 0, \\[4pt]
-(k + \alpha + \mu) u_2 + (\gamma p + \beta_1 \frac{\pi}{\mu N(t)}) u_3 + \sigma u_4 + \beta_1 \frac{\pi}{\mu N(t)} u_8 + \beta_1 \frac{\pi}{\mu N(t)} u_{12} = 0, \\[4pt]
k u_2 - (\gamma + d_1 + \mu) u_3 = 0, \\[4pt]
\alpha u_2 + (1 - p)\gamma u_3 - (\sigma + \mu) u_4 = 0, \\[4pt]
(\beta_2 \frac{\pi}{\mu N(t)} - (\delta + \omega_1 + \mu)) u_5 + \beta_2 \eta \frac{\pi}{\mu N(t)} u_6 + \beta_2 \frac{\pi}{\mu N(t)} u_7 + \\[4pt]
\beta_2 \frac{\pi}{\mu N(t)} u_8 + \beta_2 \frac{\pi}{\mu N(t)} u_9 + \eta \beta_2 \frac{\pi}{\mu N(t)} u_{11} = 0, \\[4pt]
\delta u_5 - (d_2 + \omega_2 + \mu) u_6 + \varphi u_{10} = 0, \\[4pt]
-M u_7 + \psi \gamma u_8 + \theta_1 u_9 = 0, \\[4pt]
\epsilon u_7 - P u_8 = 0, \\[4pt]
\theta_2 u_7 + (1 - \psi)\phi \gamma u_8 - (\theta_1 + \theta_3 + \omega_5 + \mu) u_9 = 0, \\[4pt]
\omega_1 u_5 + \omega_2 u_6 + \omega_3 u_7 + \omega_4 (1 - \phi) u_8 + \omega_5 u_9 - (\varphi + \mu) u_{10} + \omega_6 u_{11} + \omega_7 u_{12} = 0, \\[4pt]
\theta_3 u_9 - (\omega_6 + d_2 + \mu) u_{11} + \tau u_{12} = 0, \\[4pt]
(\sigma_1 + \delta_1) u_7 + \psi_1 u_8 - (\omega_7 + d_3 + \tau + \mu) u_{12} = 0.
\end{cases}
\tag{48}
$$

Solving system (48) we get:

$u_1 = -\left(\frac{\beta_1 \frac{\pi}{\mu N(t)} u_3 + \beta_2 \frac{\pi}{\mu N(t)} u_5 + \beta_2 \eta \frac{\pi}{\mu N(t)} u_6 + \beta_2 \frac{\pi}{\mu N(t)} u_7 + (\beta_2 + \beta_1) \frac{\pi}{\mu N(t)} u_8 + \beta_2 \frac{\pi}{\mu N(t)} u_9 + \beta_2 \eta \frac{\pi}{\mu N(t)} u_{11} + \beta_1 \frac{\pi}{\mu N(t)} u_{12}}{\mu}\right) < 0.$

$u_2 = \frac{(\gamma p + \beta_1 \frac{\pi}{\mu N(t)}) u_3 + \sigma u_4 + \beta_1 \frac{\pi}{\mu N(t)} u_8 + \beta_1 \frac{\pi}{\mu N(t)} u_{12}}{(k + \alpha + \mu)} > 0.$

$u_3 = \frac{k u_2}{(\gamma + d_1 + \mu)} > 0. u_4 = \frac{\alpha u_2 + (1 - p)\gamma u_3}{(\sigma + \mu)} > 0.$

$u_5 = \frac{\beta_2 \eta \frac{\pi}{\mu N(t)} u_6 + \beta_2 \frac{\pi}{\mu N(t)} u_7 + \beta_2 \frac{\pi}{\mu N(t)} u_8 + \beta_2 \frac{\pi}{\mu N(t)} u_9 + \eta \beta_2 \frac{\pi}{\mu N(t)} u_{11}}{(\delta + \omega_1 + \mu) - \beta_2 \frac{\pi}{\mu N(t)}} > 0$, since $(\delta + \omega_1 + \mu) > \beta_2 \frac{\pi}{\mu N(t)}$.

$u_6 = \frac{\delta u_5 + \varphi u_{10}}{(d_2 + \omega_2 + \mu)} > 0. u_7 = \frac{\psi \gamma u_8 + \theta_1 u_9}{M} > 0. u_8 = \frac{\epsilon u_7}{P} > 0. u_9 = \frac{\theta_2 u_7 + (1 - \psi)\phi \gamma u_8}{(\theta_1 + \theta_3 + \omega_5 + \mu)} > 0. u_{10} = \frac{\omega_1 u_5 + \omega_2 u_6 + \omega_3 u_7 + \omega_4 (1 - \phi) u_8 + \omega_5 u_9 + \omega_6 u_{11} + \omega_7 u_{12}}{(\varphi + \mu)} > 0. u_{11} = \frac{\theta_3 u_9 + \tau u_{12}}{(\omega_6 + d_2 + \mu)} > 0. u_{12} = \frac{(\sigma_1 + \delta_1) u_7 + \psi_1 u_8}{(\omega_7 + d_3 + \tau + \mu)} > 0.$ Again, the left eigenvector of $J(\varepsilon_0)$ symbolized by $v = (v_1, v_2, v_3, v_4, v_5, v_6, v_7, v_8, v_9, v_{10}, v_{11}, v_{12})^T$ is calculated as:

$$Y\left(\tfrac{\pi}{\mu}, 0, 0, 0, 0, 0, 0, 0, 0, 0, 0, 0\right) = [\, Y_1 \quad Y_2 \,], \text{ with}$$

$$Y_1(\varepsilon_0) = \begin{bmatrix} -\mu & 0 & 0 & 0 & 0 & 0 \\ 0 & -(k+\alpha+\mu) & k & \alpha & 0 & 0 \\ -\beta_1\frac{\pi}{\mu N(t)} & \gamma p + \beta_1\frac{\pi}{\mu N(t)} & -(\gamma+d_1+\mu) & (1-p)\gamma & 0 & 0 \\ 0 & \sigma & 0 & -(\sigma+\mu) & 0 & 0 \\ -\beta_2\frac{\pi}{\mu N(t)} & 0 & 0 & 0 & \beta_2\frac{\pi}{\mu N(t)}-(\delta+\omega_1+\mu) & \delta \\ -\beta_2\eta\frac{\pi}{\mu N(t)} & 0 & 0 & 0 & \beta_2\eta\frac{\pi}{\mu N(t)} & -(d_2+\omega_2+\mu) \\ -\beta_2\frac{\pi}{\mu N(t)} & 0 & 0 & 0 & \beta_2\frac{\pi}{\mu N(t)} & 0 \\ -(\beta_2+\beta_1)\frac{\pi}{\mu N(t)} & \beta_1\frac{\pi}{\mu N(t)} & 0 & 0 & \beta_2\frac{\pi}{\mu N(t)} & 0 \\ -\beta_2\frac{\pi}{\mu N(t)} & 0 & 0 & 0 & \beta_2\frac{\pi}{\mu N(t)} & 0 \\ 0 & 0 & 0 & 0 & 0 & \varphi \\ -\beta_2\eta\frac{\pi}{\mu N(t)} & 0 & 0 & 0 & \eta\beta_2\frac{\pi}{\mu N(t)} & 0 \\ -\beta_1\frac{\pi}{\mu N(t)} & \beta_1\frac{\pi}{\mu N(t)} & 0 & 0 & 0 & 0 \end{bmatrix},$$

$$Y_2(\varepsilon_0) = \begin{bmatrix} 0 & 0 & 0 & 0 & 0 & 0 \\ 0 & 0 & 0 & 0 & 0 & 0 \\ 0 & 0 & 0 & 0 & 0 & 0 \\ 0 & 0 & 0 & 0 & 0 & 0 \\ 0 & 0 & 0 & \omega_1 & 0 & 0 \\ 0 & 0 & 0 & \omega_2 & 0 & 0 \\ -M & \epsilon & \theta_2 & \omega_3 & 0 & (\sigma_1+\delta_1) \\ \psi\gamma & -P & (1-\psi)\phi\gamma & \omega_4(1-\varphi) & 0 & \psi_1 \\ \theta_1 & 0 & -(\theta_1+\theta_3+\omega_5+\mu) & \omega_5 & \theta_3 & 0 \\ 0 & 0 & 0 & -(\varphi+\mu) & 0 & 0 \\ 0 & 0 & 0 & \omega_6 & -(\omega_6+d_2+\mu) & 0 \\ 0 & 0 & 0 & \omega_7 & \tau & -(\omega_7+d_3+\tau+\mu) \end{bmatrix}.$$

Thus

$$\begin{bmatrix} Y_1(\varepsilon_0) & Y_2(\varepsilon_0) \end{bmatrix} \begin{pmatrix} v_1 \\ v_2 \\ v_3 \\ v_4 \\ v_5 \\ v_6 \\ v_7 \\ v_8 \\ v_9 \\ v_{10} \\ v_{11} \\ v_{12} \end{pmatrix} = \begin{pmatrix} 0 \\ 0 \\ 0 \\ 0 \\ 0 \\ 0 \\ 0 \\ 0 \\ 0 \\ 0 \\ 0 \\ 0 \end{pmatrix}. \tag{49}$$

The Eq (49) becomes;

$$-\mu v_1 = 0 \Rightarrow v_1 = 0, v_2 = \frac{k v_3 + \alpha v_4}{(k + \alpha + \mu)} > 0, v_3 = \frac{\gamma p + \beta_1 \frac{\pi}{\mu N(t)} v_2 + (1-p)\gamma v_4}{(\gamma + d_1 + \mu)} > 0, v_4 =$$

$$\frac{\sigma v_2}{\sigma + \mu} > 0, v_5 = \frac{\delta v_6 + \omega_1 v_{10}}{(\delta + \omega_1 + \mu) - \beta_2 \frac{\pi}{\mu N(t)}} > 0, v_6 = \frac{\beta_2 \eta \frac{\pi}{\mu N(t)} v_5 + \omega_2 v_{10}}{d_2 + \omega_2 + \mu} > 0, 0.1cm v_7 =$$

$$\frac{\beta_2 \frac{\pi}{\mu N(t)} v_5 + \epsilon v_8 + \theta_2 v_9 + \omega_3 v_{10} + (\sigma_1 + \delta_1)v_{12}}{M} > 0, v_8 =$$

$$\frac{\beta_1 \frac{\pi}{\mu N(t)} v_2 + \beta_2 \frac{\pi}{\mu N(t)} v_5 + \psi \gamma v_7 + (1-\psi)\phi \gamma v_9 + \omega_4(1-\phi)v_{10} + \psi_1 v_{12}}{P} > 0, v_9 =$$

$$\frac{\beta_2 \frac{\pi}{\mu N(t)} v_5 + \theta_1 v_7 + \omega_5 v_{10} + \theta_3 v_{11}}{\theta_1 + \theta_3 + \omega_5 + \mu} > 0, v_{10} = \frac{\varphi v_6}{\varphi + \mu} > 0, v_{11} = \frac{\eta \beta_2 \frac{\pi}{\mu N(t)} v_5 + \omega_6 v_{10}}{\omega_6 + d_2 + \mu} >$$

$$0, v_{12} = \frac{\beta_1 \frac{\pi}{\mu N(t)} v_2 + \omega_7 v_{10} + \tau v_{11}}{\omega_7 + d_3 + \tau + \mu} > 0.$$ We calculated $a$ and $b$ using the formula:

$a = \sum_{k,i,j=1}^{n} v_k u_i u_j \frac{\partial^2 f_k}{\partial x_i \partial x_j}(\varepsilon_0)$, and $b = \sum_{k,i=1}^{n} v_k u_i \frac{\partial^2 f_k}{\partial x_i \partial \beta_r}(\varepsilon_0)$, for $r$ = 1 or 2 [62], where $n$ is the number of compartments, and $f_i = \frac{dx_i}{dt}$ for $i$ = 1, 2, 3, . . ., 12 in (46).

Hence, $\frac{\partial^2 f_2}{\partial x_3 \partial x_1} = \frac{\partial^2 f_2}{\partial x_8 \partial x_1} = \frac{\partial^2 f_2}{\partial x_{12} \partial x_1} = \frac{\beta_1}{N(t)}, \frac{\partial^2 f_2}{\partial x_5 \partial x_2} = \frac{\partial^2 f_2}{\partial x_7 \partial x_2} = \frac{\partial^2 f_2}{\partial x_8 \partial x_2} = \frac{\partial^2 f_2}{\partial x_9 \partial x_2} =$

$-\epsilon_1 \frac{\beta_2}{N(t)}, \frac{\partial^2 f_2}{\partial x_6 \partial x_2} = \frac{\partial^2 f_2}{\partial x_{11} \partial x_2} = -\epsilon_1 \eta \frac{\beta_2}{N(t)}, \frac{\partial^2 f_3}{\partial x_5 \partial x_3} = \frac{\partial^2 f_3}{\partial x_7 \partial x_3} = \frac{\partial^2 f_3}{\partial x_8 \partial x_3} = \frac{\partial^2 f_3}{\partial x_9 \partial x_3} =$

$-\epsilon_2 \frac{\beta_2}{N(t)}, \frac{\partial^2 f_3}{\partial x_6 \partial x_3} = \frac{\partial^2 f_3}{\partial x_{11} \partial x_3} = -\epsilon_2 \eta \frac{\beta_2}{N(t)}, \frac{\partial^2 f_4}{\partial x_5 \partial x_4} = \frac{\partial^2 f_4}{\partial x_7 \partial x_4} = \frac{\partial^2 f_4}{\partial x_8 \partial x_4} = \frac{\partial^2 f_4}{\partial x_9 \partial x_4} =$

$-\frac{\beta_2}{N(t)}, \frac{\partial^2 f_4}{\partial x_6 \partial x_4} = \frac{\partial^2 f_4}{\partial x_{11} \partial x_4} = -\eta \frac{\beta_2}{N(t)}, \frac{\partial^2 f_5}{\partial x_5 \partial x_4} = \frac{\partial^2 f_5}{\partial x_7 \partial x_4} = \frac{\partial^2 f_5}{\partial x_8 \partial x_4} = \frac{\partial^2 f_5}{\partial x_9 \partial x_4} =$

$\frac{\beta_2}{N(t)}, \frac{\partial^2 f_5}{\partial x_6 \partial x_4} = \frac{\partial^2 f_5}{\partial x_{11} \partial x_4} = \eta \frac{\beta_2}{N(t)}, \frac{\partial^2 f_5}{\partial x_5 \partial x_1} = \frac{\partial^2 f_5}{\partial x_7 \partial x_1} = \frac{\partial^2 f_5}{\partial x_8 \partial x_1} = \frac{\partial^2 f_5}{\partial x_9 \partial x_1} = \frac{\beta_2}{N(t)}, \frac{\partial^2 f_5}{\partial x_6 \partial x_1} =$

$\frac{\partial^2 f_5}{\partial x_{11} \partial x_1} = \eta \frac{\beta_2}{N(t)}, \frac{\partial^2 f_5}{\partial x_3 \partial x_5} = \frac{\partial^2 f_5}{\partial x_8 \partial x_5} = \frac{\partial^2 f_5}{\partial x_{12} \partial x_5} = -\theta \frac{\beta_1}{N(t)}, \frac{\partial^2 f_6}{\partial x_3 \partial x_6} = \frac{\partial^2 f_6}{\partial x_8 \partial x_6} = \frac{\partial^2 f_6}{\partial x_{12} \partial x_6} =$

$$-\omega_2 \frac{\beta_1}{N(t)}, \frac{\partial^2 f_7}{\partial x_5 \partial x_2} = \frac{\partial^2 f_7}{\partial x_7 \partial x_2} = \frac{\partial^2 f_7}{\partial x_8 \partial x_2} = \frac{\partial^2 f_7}{\partial x_9 \partial x_2} = \epsilon \frac{\beta_2}{N(t)}, \frac{\partial^2 f_7}{\partial x_6 \partial x_2} = \frac{\partial^2 f_7}{\partial x_{11} \partial x_2} =$$

$$\eta\epsilon \frac{\beta_2}{N(t)}, \frac{\partial^2 f_7}{\partial x_3 \partial x_5} = \frac{\partial^2 f_7}{\partial x_8 \partial x_5} = \frac{\partial^2 f_7}{\partial x_{12} \partial x_5} = \theta \frac{\beta_1}{N(t)}, \frac{\partial^2 f_8}{\partial x_5 \partial x_2} = \frac{\partial^2 f_8}{\partial x_7 \partial x_2} = \frac{\partial^2 f_8}{\partial x_8 \partial x_2} = \frac{\partial^2 f_8}{\partial x_9 \partial x_2} =$$

$$\epsilon_2 \frac{\beta_2}{N(t)}, \frac{\partial^2 f_8}{\partial x_6 \partial x_2} = \frac{\partial^2 f_8}{\partial x_{11} \partial x_2} = \eta\epsilon_2 \frac{\beta_2}{N(t)}, \frac{\partial^2 f_{12}}{\partial x_3 \partial x_6} = \frac{\partial^2 f_{12}}{\partial x_8 \partial x_6} = \frac{\partial^2 f_{12}}{\partial x_{12} \partial x_6} = \omega \frac{\beta_1}{N(t)}.$$ Then, $a =$

$$v_2 u_1 u_3 \frac{\beta_1}{N} + v_2 u_1 u_8 \frac{\beta_1}{N} + v_2 u_1 u_{12} \frac{\beta_1}{N} + v_2 u_2 u_5 \frac{-\epsilon_1 \beta_2}{N} + v_2 u_2 u_7 \frac{-\epsilon_1 \beta_2}{N} + v_2 u_2 u_8 \frac{-\epsilon_1 \beta_2}{N} +$$

$$v_2 u_2 u_9 \frac{-\epsilon_1 \beta_2}{N} + v_2 u_2 u_6 \frac{-\eta\epsilon_1 \beta_2}{N} + v_2 u_2 u_{11} \frac{-\eta\epsilon_1 \beta_2}{N} + v_3 u_3 u_5 \frac{-\epsilon_2 \beta_2}{N} + v_3 u_3 u_7 \frac{-\epsilon_2 \beta_2}{N} +$$

$$v_3 u_3 u_8 \frac{-\epsilon_2 \beta_2}{N} + v_3 u_3 u_9 \frac{-\epsilon_2 \beta_2}{N} + v_3 u_3 u_6 \frac{-\eta\epsilon_2 \beta_2}{N} + v_3 u_3 u_{11} \frac{-\eta\epsilon_2 \beta_2}{N} + v_4 u_4 u_5 \frac{-\beta_2}{N} +$$

$$v_4 u_4 u_7 \frac{-\beta_2}{N} + v_4 u_4 u_8 \frac{-\beta_2}{N} + v_4 u_4 u_9 \frac{-\beta_2}{N} + v_4 u_4 u_6 \frac{-\eta\beta_2}{N} + v_4 u_4 u_{11} \frac{-\eta\beta_2}{N} + v_5 u_4 u_5 \frac{\beta_2}{N} +$$

$$v_5 u_4 u_7 \frac{\beta_2}{N} + v_5 u_4 u_8 \frac{\beta_2}{N} + v_5 u_4 u_9 \frac{\beta_2}{N} + v_5 u_4 u_6 \frac{\eta\beta_2}{N} + v_5 u_4 u_{11} \frac{\eta\beta_2}{N} + v_5 u_1 u_5 \frac{\beta_2}{N} + v_5 u_1 u_7 \frac{\beta_2}{N} +$$

$$v_5 u_1 u_8 \frac{\beta_2}{N} + v_5 u_1 u_9 \frac{\beta_2}{N} + v_5 u_1 u_6 \frac{\eta\beta_2}{N} + v_5 u_1 u_{11} \frac{\eta\beta_2}{N} + v_5 u_5 u_3 \frac{-\theta\beta_1}{N} + v_5 u_5 u_8 \frac{-\theta\beta_1}{N} +$$

$$v_5 u_5 u_{12} \frac{-\theta\beta_1}{N} + v_6 u_6 u_3 \frac{-\omega_2 \beta_1}{N} + v_6 u_6 u_8 \frac{-\omega_2 \beta_1}{N} + v_6 u_6 u_{12} \frac{-\omega_2 \beta_1}{N} + v_7 u_5 u_3 \frac{\theta\beta_1}{N} + v_7 u_5 u_8 \frac{\theta\beta_1}{N} +$$

$$v_7 u_5 u_{12} \frac{\theta\beta_1}{N} + v_{12} u_6 u_3 \frac{\omega\beta_1}{N} + v_{12} u_6 u_8 \frac{\omega\beta_1}{N} + v_{12} u_6 u_{12} \frac{\omega\beta_1}{N} + v_7 u_2 u_5 \frac{\epsilon\beta_2}{N} + v_7 u_2 u_7 \frac{\epsilon\beta_2}{N} +$$

$$v_7 u_2 u_8 \frac{\epsilon\beta_2}{N} + v_7 u_2 u_9 \frac{\epsilon\beta_2}{N} + v_7 u_2 u_6 \frac{\epsilon\eta\beta_2}{N} + v_7 u_2 u_{11} \frac{\epsilon\eta\beta_2}{N} + v_8 u_2 u_5 \frac{\epsilon_2\beta_2}{N} +$$

$$v_8 u_2 u_7 \frac{\epsilon_2 \beta_2}{N} + v_8 u_2 u_8 \frac{\epsilon_2 \beta_2}{N} + v_8 u_2 u_9 \frac{\epsilon_2 \beta_2}{N} + v_8 u_2 u_6 \frac{\epsilon_2 \eta\beta_2}{N} + v_8 u_2 u_{11} \frac{\epsilon_2 \eta\beta_2}{N}.$$

We considered the case when $R_T > R_H$ i.e., $R_0 = R_T$ and $R_0 = 1$.

Choose $\beta_1 = \beta_1^*$ as the bifurcation parameter.

Now,
$$b = \sum_{k,i=1}^n v_k u_i \frac{\partial^2 f_k}{\partial x_i \partial \beta_1}(\varepsilon_0) \quad = v_2 u_3 \frac{\pi}{N\mu} + v_2 u_8 \frac{\pi}{N\mu} + v_2 u_{12} \frac{\pi}{N\mu}.$$

$$= v_2 \frac{\pi}{N\mu}(u_3 + u_8 + u_{12}) > 0.$$

Again,

$a = (v_2 u_1 + u_6(v_{12}\omega - v_6\omega_2))(u_3 + u_8 + u_{12})\frac{\beta_1}{N} + \Delta \frac{\beta_2}{N}(u_5 + u_7 + u_8 + u_9 + \eta(u_6 + u_{11}))$,
where $\Delta = v_5 u_3 + v_5 u_4 + v_7 u_5\theta + v_7 u_2\epsilon + v_8 u_2\epsilon_2 - v_2 u_2 - v_3 u_3 - v_4 u_4 - v_5 u_5 = u_4(v_5 - v_4) + u_5(\theta v_7 - v_5) + u_2(\epsilon_2 v_8 - v_2) + u_3(v_5 - v_3) + v_7 u_2\epsilon > 0$, since $v_5 > v_4$, $\theta v_7 > v_5$, $\epsilon_2 v_8 > v_2$, $v_5 > v_3$ as one eigenvector can be expressed by the other. In addition $v_{12}\omega > v_6\omega_6$.

Hence, both the values of $a$ and $b$ are positive.

Therefore, the model (2) displays a backward bifurcation at $R_0 = 1$. As $R_0$ approaches one, the number of co-infectious individuals jump suddenly from 0 to the large endemic equilibrium. Fig 2 shows the diagram representation of a backward bifurcation phenomena. If this bifurcation happen, the HIV-TB co-infection disease persists in the community, even though $R_0 < 1$. As shown in the Fig 2, there are three equilibria co-exist when $R_0$ is in the range $0 < R_c < R_0 < 1$, where $R_c$ is a critical value, which in Fig 2 is $R_c = 0.4835$. In this range, both the DFE and EE points are stable, while the middle equilibrium is unstable. When $R_0 < R_c$, only the DFE point exists and is stable. This justifies reducing $R_0$ below unity is only a necessary but not sufficient condition for disease elimination. Thus, always a disease eradication cannot just be achieved by making $R_0 < 1$.

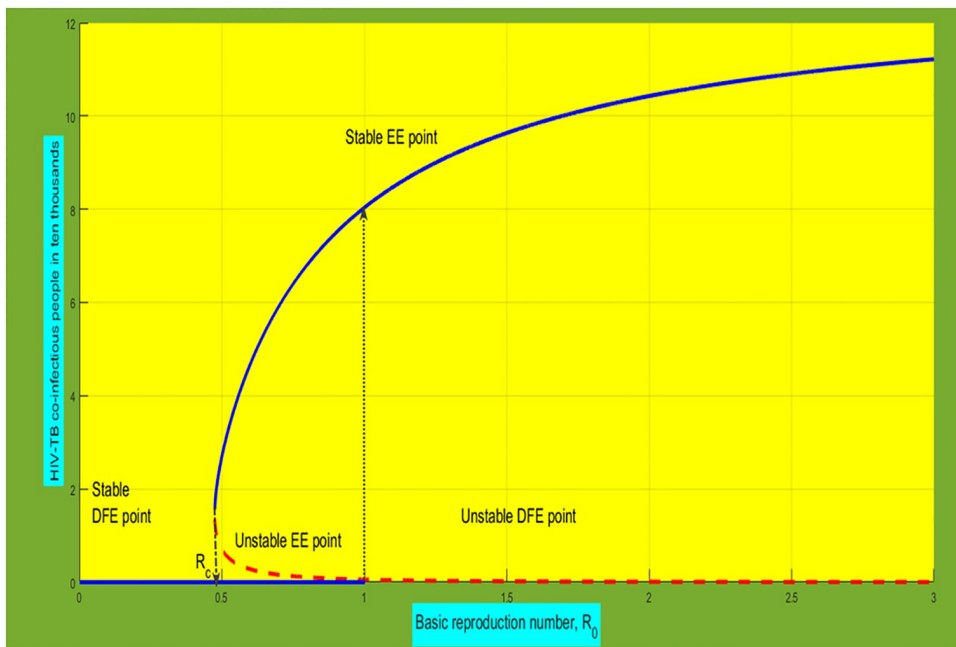

**Fig 2. Numerical simulation of backward bifurcation at the point $R_0 = 1$, where the vertical axis represents the HIV-TB co-infectious individuals.**

## 4 Model with optimal control

We used the following four (two preventive and two controlling) efforts.

1. The preventive effort of TB disease ($u_1(t)$) represents the effort of protecting susceptible individuals who are becoming infected. Such mechanisms are healthy educational campaigns and early detection as well as isolation of infectious individuals are associated with $u_1(t)$.

2. The preventive effort of HIV/AIDS disease ($u_2(t)$) implies the effort of protecting susceptible individuals who are contacting to HIV/AIDS infected people. Such mechanisms are the HIV/AIDS educational campaign and early detection of HIV infected individuals are associated with $u_2(t)$.

3. The case finding for TB disease ($u_3(t)$). The effort $u_3(t)$ illustrates the screening and then treatment of high- risk latent TB. The risk that TB infection will progress to TB disease is greatly reduced by treatment of latent TB. Since finite groups are at a high risk of growing TB disease once infected. This effort is a key mechanism for TB control strategy.

4. The treatment effort for HIV/AIDS disease ($u_4(t)$). The strategy $u_4(t)$ refers to treating HIV infected people with Antiretroviral therapy (ART). This can decrease the individual's infectiousness level by reducing their viral load and helping them to recapture their immunity to obtain a better life. This treatment can also curtail HIV-TB co-infection rate.

Thus, incorporating the above strategies in the model (2), we get the following optimal control model of HIV-TB co-epidemic.

$$
\begin{cases}
\dfrac{dS}{dt} = \pi - ((1 - u_2)\lambda_H + (1 - u_1)\lambda_T + \mu)S, \\[2mm]
\dfrac{dE}{dt} = (1 - u_1)\lambda_T S + \gamma p I + \sigma L - (k + \alpha + \epsilon_1\lambda_H + \mu)E, \\[2mm]
\dfrac{dI}{dt} = kE - (\gamma + \epsilon_2\lambda_H + d_1 + \mu)I, \\[2mm]
\dfrac{dL}{dt} = (1 - p)\gamma I + \alpha E - (\sigma + \lambda_H + \mu)L, \\[2mm]
\dfrac{dH}{dt} = (1 - u_2)\lambda_H S + \lambda_H L - (\theta\lambda_T + \delta + \omega_1(1 + u_4) + \mu)H, \\[2mm]
\dfrac{dA}{dt} = \delta H + \varphi T - (\omega_2(1 + u_4) + \omega\lambda_T + \mu + d_2)A, \\[2mm]
\dfrac{dH_E}{dt} = \epsilon_1\lambda_H E + \theta\lambda_T H + \psi\gamma H_I + \theta_1 H_L - (\epsilon + \omega_3(1 + u_4) + \\[1mm]
\qquad\qquad \theta_2(1 + u_3) + \sigma_1 + \delta_1 + \mu)H_E, \\[2mm]
\dfrac{dH_I}{dt} = \epsilon_2\lambda_H I + \epsilon H_E - (\psi\gamma + (1 - \psi)\gamma\phi + (1 - \phi)\omega_4(1 + u_4) + \\[1mm]
\qquad\qquad \psi_1 + \mu + d_1)H_I, \\[2mm]
\dfrac{dH_L}{dt} = (1 - \psi)\phi\gamma H_I + \theta_2(1 + u_3)H_E - (\theta_1 + \theta_3 + \omega_5(1 + u_4) + \mu)H_L, \\[2mm]
\dfrac{dT}{dt} = \omega_1(1 + u_4)H + \omega_2(1 + u_4)A + \omega_3(1 + u_4)H_E + \omega_4(1 + u_4)(1 - \phi)H_I + \\[1mm]
\qquad\qquad \omega_5(1 + u_4)H_L + \omega_6(1 + u_4)A_L + \omega_7(1 + u_4)A_I - (\varphi + \mu)T, \\[2mm]
\dfrac{dA_L}{dt} = (\tau)A_I + \theta_3 H_L - (\omega_6(1 + u_4) + \mu + d_2)A_L, \\[2mm]
\dfrac{dA_I}{dt} = (\sigma_1 + \delta_1)H_E + \psi_1 H_I + \omega\lambda_T A - (\omega_7(1 + u_4) + \mu + d_3 + \tau)A_I.
\end{cases}
\tag{50}
$$

The optimal controls are defined in the set $U = \{u_i(t)\colon 0 \le u_i(t) \le 1, 0 \le t \le T\}$, where $i = 1, 2, 3, 4$.

Let the objective function can be expressed as: [63, 64]:

$$
J(t) = \int_0^{t_f} [b_1 H_E(t) + b_2 H_I(t) + b_3 H_L(t) + b_4 A_L(t) + b_5 A_I(t) + \frac{1}{2}\sum_{i=1}^{4} c_i u_i^2(t)]dt, \tag{51}
$$

where $b_1$, $b_2$, $b_3$, $b_4$, and $b_5$ are the cost associated with the number of $H_E$, $H_I$, $H_L$, $A_L$, and $A_I$ compartments respectively. The constants $c_i$, $i = 1, 2, 3, 4$ are the costs of executing the strategies from $u_1$ up to $u_4$ respectively [65]. We have taken a quadratic form for determining the cost of intervention [66, 67].

Thus, we try to find the optimal controls $u_1^*$, $u_2^*$, $u_3^*$, and $u_4^*$ satisfying

$J(u_1^*, u_2^*, u_3^*, u_4^*) = \min \{J(u_1, u_2, u_3, u_4) | (u_1, u_2, u_3, u_4) \in U\}$, where $U$ is the set expressed above.

**Theorem 6** (*Existence of solutions*). *There exists an optimal control $u_1^*(t)$, $u_2^*(t)$, $u_3^*(t)$, $u_4^*(t)$ and solutions S, E, I, L, H, A, $H_E$, $H_I$, $H_L$, $A_L$, T, and $A_I$ to control induced initial value problem* (50) *that minimizes the objective functional $J(u_i(t))$, $i = 1, 2, 3, 4$ of* (51) *over the set of admissible control U.*

**Proof:**

Look the following three conditions of Fleming and Rishel's theorem [68].

1. The set of solutions of (50) and (51) that incorporate control variables in $U$ is non-empty.

2. The system of Eq (50) is a linear combination of control functions with coefficients are state variables.

3. The integrand L in Eq (51) becomes; $L(x, u, t) = b_1 H_E(t) + b_2 H_I(t) + b_3 H_L(t) + b_4 A_L(t) + b_5 A_I(t) + \frac{1}{2} \sum_{i=1}^{4} c_i u_i^2(t)$ is convex on $U$ and it also fulfills $L(x, u, t) \geq \delta_1 |(u_1, u_2, u_3, u_4)|^\beta - \delta_2$, where $\delta_1 > 0$ and $\beta > 1$.

Firstly, to proof 1, we mentioned to [69, 70]. If the solutions of (50) and (51) are bounded plus Lipschitz, then they are unique.

Thus, the total population $N(t)$ is also bounded above by $\frac{\pi}{\mu}$ and bellow by $N_0 \neq 0$. Here, each compartment in $N(t)$ is bounded. In that case the state variables are bounded and continuous. Hence, this displays that there is the boundedness of the partial derivatives with respect to the state variables with in the system [71].

This accomplishes that proof 1 holds.

Secondly, the right hand side equation of the control system (50) can be written as:

$$
\begin{pmatrix} f_1 \\ f_2 \\ f_3 \\ f_4 \\ f_5 \\ f_6 \\ f_7 \\ f_8 \\ f_9 \\ f_{10} \\ f_{11} \\ f_{12} \end{pmatrix} + \begin{pmatrix} \lambda_T S & \lambda_H S & 0 & 0 \\ -\lambda_T S & 0 & 0 & 0 \\ 0 & 0 & 0 & 0 \\ 0 & 0 & 0 & 0 \\ 0 & \lambda_H S & 0 & -\lambda_H S \\ 0 & -\lambda_H S & 0 & -\omega_1 H \\ 0 & 0 & 0 & -\omega_2 A \\ 0 & 0 & -\theta_2 H_E & -\omega_3 H_E \\ 0 & 0 & 0 & -\omega_4(1-\phi)H_I \\ 0 & 0 & \theta_2 H_E & -\omega_5 H_L \\ 0 & 0 & 0 & R \\ 0 & 0 & 0 & -\omega_6 A_L \\ 0 & 0 & 0 & -\omega_7 A_I \end{pmatrix} \cdot \begin{pmatrix} u_1 \\ u_2 \\ u_3 \\ u_4 \end{pmatrix}, \tag{52}
$$

where

$$
\begin{aligned}
f_1 &= \pi - (\lambda_H + \lambda_T + \mu)S, \\[4pt]
f_2 &= \lambda_T S + \gamma p I + \sigma L - (k + \alpha + \epsilon_1 \lambda_H + \mu)E, \\[4pt]
f_3 &= kE - (\gamma + \epsilon_2 \lambda_H + d_1 + \mu)I, \\[4pt]
f_4 &= (1 - p)\gamma I + \alpha E - (\sigma + \lambda_H + \mu)L, \\[4pt]
f_5 &= \lambda_H S + \lambda_H L - (\theta \lambda_T + \delta + \omega_1 + \mu)H, \\[4pt]
f_6 &= \delta H + \varphi T - (\omega_2 + \omega \lambda_T + \mu + d_2)A, \\[4pt]
f_7 &= \epsilon_1 \lambda_H E + \theta \lambda_T H + \psi \gamma H_I + \theta_1 H_L - (\epsilon + \omega_3 + \theta_2 + \sigma_1 + \delta_1 + \mu)H_E, \\[4pt]
f_8 &= \epsilon_2 \lambda_H I + \epsilon H_E - (\psi \gamma + (1 - \psi)\gamma \phi + (1 - \phi)\omega_4 + \psi_1 + \mu + d_1)H_I, \\[4pt]
f_9 &= (1 - \psi)\phi \gamma H_I + \theta_2 H_E - (\theta_1 + \theta_3 + \omega_5 + \mu)H_L, \\[4pt]
f_{10} &= \omega_1 H + \omega_2 A + \omega_3 H_E + \omega_4 (1 - \phi)H_I + \omega_5 H_L + \omega_6 A_L + \omega_7 A_I - (\varphi + \mu)T, \\[4pt]
f_{11} &= \tau A_I + \theta_3 H_L - (\omega_6 + \mu + d_2)A_L, \\[4pt]
f_{12} &= (\sigma_1 + \delta_1)H_E + \psi_1 H_I + \omega \lambda_T A - (\omega_7 + \mu + d_3 + \tau)A_I, \\[4pt]
R &= -(\omega_1 H + \omega_2 A + \omega_3 H_E + \omega_4 (1 - \phi)H_I + \omega_5 H_L + \omega_6 A_L + \omega_7 A_I).
\end{aligned}
\tag{53}
$$

Clearly, the system (52) is linearly dependent on $u_1$, $u_2$, $u_3$, and $u_4$. Therefore, the second condition holds.

Lastly, to verify condition 3, we need to show for any $\varrho \in (0, 1)$ such that:

$$
(1 - \varrho)L(x, u, t) + \varrho L(x, v, t) \geq L(x, t, (1 - \varrho)u, \varrho v),
\tag{54}
$$

where

$$
(1 - \varrho)L(x, u, t) + \varrho L(x, v, t) = b_1 H_E(t) + b_2 H_I(t) + b_3 H_L(t) + b_4 A_L(t) + b_5 A_I(t) + \frac{1 - \varrho}{2}\sum_{i=1}^{4} c_i u_i^2(t) + \frac{\varrho}{2}\sum_{i=1}^{4} c_i v_i^2(t).
\tag{55}
$$

$$
L(x, t, (1 - \varrho)u, \varrho v) = b_1 H_E(t) + b_2 H_I(t) + b_3 H_L(t) + b_4 A_L(t) + b_5 A_I(t) + \frac{1}{2}\sum_{i=1}^{4} c_i ((1 - \varrho)u_i + \varrho v_i)^2(t).
\tag{56}
$$

Further,

$$
(1 - \varrho)L(x, u, t) + \quad \varrho L(x, v, t) - L(x, t, (1 - \varrho)u, \varrho v)
$$

$$
= \frac{1 - \varrho}{2} \sum_{i=1}^{4} c_i u_i^2(t) + \frac{\varrho}{2} \sum_{i=1}^{4} c_i v_i^2(t) - \frac{1}{2} \sum_{i=1}^{4} c_i((1 - \varrho)u_i + \varrho v_i)^2(t),
$$

$$
= \frac{1}{2} \sum_{i=1}^{4} c_i[(1 - \varrho)u_i^2(t) + \varrho v_i^2(t) - ((1 - \varrho)u_i + \varrho v_i)^2(t)],
$$

$$
= \frac{1}{2} \sum_{i=1}^{4} c_i[\sqrt{\varrho(1 - \varrho)}u_i(t) - \sqrt{\varrho(1 - \varrho)}v_i(t)]^2],
$$

$$
= \frac{1}{2} \varrho(1 - \varrho) \sum_{i=1}^{4} c_i[u_i(t) - v_i(t)]^2] \geq 0.
$$

(57)

Hence, $L(x, u, t)$ is convex on $U$.

Hereafter, to prove the boundedness on $L$, we used the following technique.

$c_3 u_3^2 \leq c_3$, since $u_3 \in [0, 1]$.

Thus $\frac{1}{2} c_3 u_3^2 \leq \frac{c_3}{2}$. This implies $\frac{1}{2} c_3 u_3^2 - \frac{c_3}{2} \leq 0$.

The function $L(x, u, t)$ is expressed as:

$$
L(x, u, t) = b_1 H_E(t) + b_2 H_I(t) + b_3 H_L(t) + b_4 A_L(t) + b_5 A_I(t) + \frac{1}{2} \sum_{i=1}^{4} c_i u_i^2(t) \geq \frac{1}{2} \sum_{i=1}^{4} c_i u_i^2 - \frac{c_3}{2},
$$

$$
L(x, u, t) \geq \min\left(\frac{c_1}{2}, \frac{c_2}{2}, \frac{c_3}{2}, \frac{c_4}{2}\right)(u_1^2 + u_2^2 + u_3^2 + u_4^2) - \frac{c_3}{2},
$$

(58)

$$
L(x, u, t) \geq \min\left(\frac{c_1}{2}, \frac{c_2}{2}, \frac{c_3}{2}, \frac{c_4}{2}\right) \| (u_1, u_2, u_3, u_4) \|^2 - \frac{c_3}{2}.
$$

Therefore, $L(x, u, t) \geq \delta_1 ||(u_1, u_2, u_3, u_4)||^\beta - \delta_2$, where $\delta_1 = \min(\frac{c_1}{2}, \frac{c_2}{2}, \frac{c_3}{2}, \frac{c_4}{2})$ and $\delta_2 = \frac{c_4}{2}$ and $\beta = 2$.

**Theorem 7** *Let $x = (S, E, I, L, H, A, H_E, H_I, H_L, A_L, T, A_I)$ and $u = (u_1, u_2, u_3, u_4)$ is an optimal control pair, then there exists a continuously differentiable vector $\lambda_1(t), \ldots, \lambda_{12}(t)$ satisfying*

*the following*:

$$
\begin{cases}
\dfrac{d\lambda_1}{dt} = (1 - u_1)(\lambda_1 - \lambda_2)\dfrac{\beta_1(I + H_I + A_I)}{N} + (1 - u_2)(\lambda_1 - \lambda_5)\dfrac{\beta_2(H + H_E + H_I + H_L + \eta(A + A_L))}{N}, \\[2mm]
\dfrac{d\lambda_2}{dt} = \lambda_2(k + \alpha + \mu) - \lambda_3 k - \lambda_4 \alpha + (\lambda_2 - \lambda_7)\epsilon_1\left(\dfrac{\beta_2(H + H_E + H_I + H_L + \eta(A + A_L))}{N}\right), \\[2mm]
\dfrac{d\lambda_3}{dt} = (1 - u_1)(\lambda_1 - \lambda_2)\dfrac{\beta_1 S}{N} - \lambda_2\gamma p + \lambda_3\left(\gamma + \epsilon_2 \dfrac{\beta_2(H + H_E + H_I + H_L + \eta(A + A_L))}{N} + d_1 + \mu\right) \\[2mm]
\qquad - \lambda_4(1 - p)\gamma + \dfrac{\theta\beta_1 H}{N}(\lambda_5 - \lambda_7) + \dfrac{\omega\beta_1 A}{N}(\lambda_6 - \lambda_{12}), \\[2mm]
\dfrac{d\lambda_4}{dt} = -\lambda_2\sigma + \lambda_4(\sigma + \mu) + (\lambda_4 - \lambda_5)\dfrac{\beta_2(H + H_E + H_I + H_L + \eta(A + A_L))}{N}, \\[2mm]
\dfrac{d\lambda_5}{dt} = \lambda_1(1 - u_2)\dfrac{\beta_2 S}{N} + \epsilon_1\dfrac{\beta_2 E}{N}(\lambda_2 - \lambda_7) + \epsilon_2\dfrac{\beta_2 I}{N}(\lambda_3 - \lambda_8) + \dfrac{\beta_2 L}{N}(\lambda_4 - \lambda_5) \\[2mm]
\qquad + \lambda_5(1 - u_2)\dfrac{\beta_2 S}{N} - \sigma\lambda_6 - \lambda_7\theta\left(\dfrac{\beta_1(I + H_I + A_I)}{N}\right) - \lambda_{10}\omega_1(1 + u_4), \\[2mm]
\dfrac{d\lambda_6}{dt} = \lambda_1(1 - u_2)\dfrac{\eta\beta_2 S}{N} + \epsilon_1\dfrac{\eta\beta_2 E}{N}(\lambda_2 - \lambda_7) + \epsilon_2\dfrac{\eta\beta_2 I}{N}(\lambda_3 - \lambda_8) + \dfrac{\eta\beta_2 L}{N}(\lambda_4 - \lambda_5) \\[2mm]
\qquad + \lambda_5(1 - u_2)\dfrac{\eta\beta_2 S}{N} + \lambda_6(d_2 + \mu) + (\lambda_6 - \lambda_{10})\omega_2(1 + u_4) + (\lambda_6 - \lambda_{12})\omega\left(\dfrac{\beta_1(I + H_I + A_I)}{N}\right), \\[2mm]
\dfrac{d\lambda_7}{dt} = -b_1 + \lambda_1(1 - u_2)\dfrac{\beta_2 S}{N} + \epsilon_1\dfrac{\beta_2 E}{N}(\lambda_2 - \lambda_7) + \epsilon_2\dfrac{\beta_2 I}{N}(\lambda_3 - \lambda_8) + \dfrac{\beta_2 L}{N}(\lambda_4 - \lambda_5) \\[2mm]
\qquad + \lambda_5(1 - u_2)\dfrac{\beta_2 S}{N} + \lambda_7(\epsilon + \mu) + (\lambda_7 - \lambda_9)\theta_2(1 + u_3) + \\[2mm]
\qquad (\lambda_7 - \lambda_{10})\omega_3(1 + u_4) + (\lambda_7 - \lambda_{12})(\sigma_1 + \delta_1), \\[2mm]
\dfrac{d\lambda_8}{dt} = -b_2 + \lambda_1(1 - u_2)\dfrac{\beta_2 S}{N} + \epsilon_1\dfrac{\beta_2 E}{N}(\lambda_2 - \lambda_7) + \epsilon_2\dfrac{\beta_2 I}{N}(\lambda_3 - \lambda_8) + \dfrac{\beta_2 L}{N}(\lambda_4 - \lambda_5) \\[2mm]
\qquad + \lambda_5(1 - u_2)\dfrac{\beta_2 S}{N} + \lambda_6\dfrac{\omega\beta_1 A}{N} + (\lambda_8 - \lambda_7)\psi\gamma + \lambda_8(1 - \psi)\gamma\phi + \\[2mm]
\qquad \psi_1(\lambda_8 - \lambda_{12}) + (\lambda_8 - \lambda_{10})\omega_4(1 + u_4)(1 - \phi), \\[2mm]
\dfrac{d\lambda_9}{dt} = -b_3 + \lambda_1(1 - u_2)\dfrac{\beta_2 S}{N} + \epsilon_1\dfrac{\beta_2 E}{N}(\lambda_2 - \lambda_7) + \epsilon_2\dfrac{\beta_2 I}{N}(\lambda_3 - \lambda_8) + \dfrac{\beta_2 L}{N}(\lambda_4 - \lambda_5) \\[2mm]
\qquad + \lambda_5(1 - u_2)\dfrac{\beta_2 S}{N} + (\lambda_9 - \lambda_7)\theta_1 + (\lambda_9 - \lambda_{11})\theta_3 + (\lambda_9 - \lambda_{10})\omega_5(1 + u_4) + \lambda_9\mu, \\[2mm]
\dfrac{d\lambda_{10}}{dt} = -\lambda_6\varphi + \lambda_{10}(\varphi + \mu), \\[2mm]
\dfrac{d\lambda_{11}}{dt} = -b_4 + \lambda_1(1 - u_2)\dfrac{\eta\beta_2 S}{N} + \epsilon_1\dfrac{\eta\beta_2 E}{N}(\lambda_2 - \lambda_7) + \epsilon_2\dfrac{\eta\beta_2 I}{N}(\lambda_3 - \lambda_8) + \dfrac{\eta\beta_2 L}{N}(\lambda_4 - \lambda_5) \\[2mm]
\qquad + \lambda_5(1 - u_2)\dfrac{\eta\beta_2 S}{N} + (\lambda_{11} - \lambda_{10})\omega_6(1 + u_4) + \lambda_{11}(d_2 + \mu), \\[2mm]
\dfrac{d\lambda_{12}}{dt} = -b_5 + (\lambda_1 - \lambda_2)(1 - u_1)\dfrac{\eta\beta_1 S}{N} + (\lambda_1 - \lambda_5)(1 - u_2)\dfrac{\eta\beta_2 S}{N} + \epsilon_1\dfrac{\eta\beta_2 E}{N}(\lambda_2 - \lambda_7) + \\[2mm]
\qquad \epsilon_2\dfrac{\eta\beta_2 I}{N}(\lambda_3 - \lambda_8) + \dfrac{\eta\beta_2 L}{N}(\lambda_4 - \lambda_5) + (\lambda_5 - \lambda_7)\dfrac{\theta\beta_1\eta H}{N} + \lambda_6)\dfrac{\omega\beta_1\eta A}{N} + (\lambda_{12} - \lambda_{10})\omega_7(1 + u_4) \\[2mm]
\qquad + (\lambda_{12} - \lambda_{11})(\tau) + \lambda_{12}(d_3 + \mu),
\end{cases}
\tag{59}
$$

*with transversality conditions* $\lambda_i(t_f) = 0$, $i = 1, 2, 3, \ldots, 12$.

*Moreover, we get the control set $(u_1^*(t), u_2^*(t), u_3^*(t), u_4^*(t))$ characterized by*

$$u_1^*(t) = max\{0, min(1, u_1^*)\}, \quad u_2^*(t) = max\{0, min(1, u_2^*)\},$$
$$u_3^*(t) = max\{0, min(1, u_3^*)\}, \quad u_4^*(t) = max\{0, min(1, u_4^*)\};$$

*where*

$$u_1^* = \frac{\beta_1 S(\lambda_2 - \lambda_1)(I + H_I + A_I)}{c_1 N},$$

$$u_2^* = \frac{\beta_2 S(\lambda_5 - \lambda_1)(H + H_E + H_I + H_L + \eta(A + A_I))}{c_2 N},$$

$$u_3^* = \frac{(\lambda_7 - \lambda_9)\theta_2 H_E}{c_3},$$

*and*

$$u_4^* = \frac{(\lambda_5 - \lambda_{10})\omega_1 H + (\lambda_6 - \lambda_{10})\omega_2 A + (\lambda_7 - \lambda_{10})\omega_3 H_E + (\lambda_8 - \lambda_{10})\omega_4(1 - \phi)H_I + (\lambda_9 - \lambda_{10})\omega_5 H_L + (\lambda_{11} - \lambda_{10})\omega_6 A_L + (\lambda_{12} - \lambda_{10})\omega_7 A_I}{c_4}.$$

**Proof:**

By using the Pontryagin's Maximum Principle (PMP) [72], we found a Hamiltonian $\mathbb{H}$ stated as:

$\mathbb{H}(S, E, I, L, H, A, H_E, H_I, H_L, A_L, T, A_I, u, t) = L(x, u, t) + \lambda_1 \frac{dS}{dt} + \lambda_2 \frac{dE}{dt} + \lambda_3 \frac{dI}{dt} + \lambda_4 \frac{dL}{dt} + \lambda_5 \frac{dH}{dt} + \lambda_6 \frac{dA}{dt} + \lambda_7 \frac{dH_E}{dt} + \lambda_8 \frac{dH_I}{dt} + \lambda_9 \frac{dH_L}{dt} + \lambda_{10} \frac{dT}{dt} + \lambda_{11} \frac{dA_L}{dt} + \lambda_{12} \frac{dA_I}{dt}$, where $\lambda_i$, $i = 1, 2, \ldots,$ 12 are the co-state variables.

H, the Hamiltonian ($\mathbb{H}$) is described by:

$\mathbb{H} = b_1 H_E(t) + b_2 H_I(t) + b_3 H_L(t) + b_4 A_L(t) + b_5 A_I(t) + \frac{1}{2}(c_1 u_1^2 + c_2 u_2^2 + c_3 u_3^2 + c_4 u_4^2) + \lambda_1[\pi - ((1 - u_2)\lambda_H + (1 - u_1)\lambda_T + \mu)S] + \lambda_2[(1 - u_1)\lambda_T S + \gamma p I + \sigma L - (k + \alpha + \epsilon_1 \lambda_H + \mu)E] + \lambda_3[kE - (\gamma + \epsilon_2 \lambda_H + d_1 + \mu)I] + \lambda_4[(1 - p)\gamma I + \alpha E - (\sigma + \lambda_H + \mu)L] + \lambda_5[(1 - u_2)\lambda_H S + \lambda_H L - (\theta \lambda_T + \delta + \omega_1(1 + u_4) + \mu)H] + \lambda_6[\delta H + \varphi T - (\omega_2(1 + u_4) + \omega \lambda_T + \mu + d_2)A] + \lambda_7[\epsilon_1 \lambda_H E + \theta \lambda_T H + \psi \gamma H_I + \theta_1 H_L - (\epsilon + \omega_3(1 + u_4) + \theta_2(1 + u_3) + \sigma_1 + \delta_1 + \mu)H_E] + \lambda_8[\epsilon_2 \lambda_H I + \epsilon H_E - (\psi \gamma + (1 - \psi)\gamma \phi + (1 - \phi)\omega_4(1 + u_4) + \psi_1 + \mu + d_1)H_I] + \lambda_9[(1 - \psi)\phi \gamma H_I + \theta_2(1 + u_3)H_E - (\theta_1 + \theta_3 + \omega_5(1 + u_4) + \mu)H_L] + \lambda_{10}[\omega_1(1 + u_4)H + \omega_2(1 + u_4)A + \omega_3(1 + u_4)H_E + \omega_4(1 + u_4)(1 - \phi)H_I + \omega_5(1 + u_4)H_L + \omega_6(1 + u_4)A_L + \omega_7(1 + u_4)A_I - (\varphi + \mu)T] + \lambda_{11}[(\tau)A_I + \theta_3 H_L - (\omega_6(1 + u_4) + \mu + d_2)A_L] + \lambda_{12}[(\sigma_1 + \delta_1)H_E + \psi_1 H_I + \omega \lambda_T A - (\omega_7(1 + u_4) + \mu + d_3 + \tau + u_4)A_I]$

Hereafter, the second condition of the PMP sates that, $\exists$ adjoint variables $\lambda_i$, $i = 1, 2, \ldots, 12$ which fulfill the next equalities.

$$\frac{d\lambda_1}{dt} = -\frac{d\mathbb{H}}{dS}, \quad \frac{d\lambda_2}{dt} = -\frac{d\mathbb{H}}{dE}, \quad \frac{d\lambda_3}{dt} = -\frac{d\mathbb{H}}{dI}, \quad \frac{d\lambda_4}{dt} = -\frac{d\mathbb{H}}{dL},$$

$$\frac{d\lambda_5}{dt} = -\frac{d\mathbb{H}}{dH}, \quad \frac{d\lambda_6}{dt} = -\frac{d\mathbb{H}}{dA}, \quad \frac{d\lambda_7}{dt} = -\frac{d\mathbb{H}}{dH_E}, \quad \frac{d\lambda_8}{dt} = -\frac{d\mathbb{H}}{dH_I},$$

$$\frac{d\lambda_9}{dt} = -\frac{d\mathbb{H}}{dH_L}, \quad \frac{d\lambda_{10}}{dt} = -\frac{d\mathbb{H}}{dT}, \quad \frac{d\lambda_{11}}{dt} = -\frac{d\mathbb{H}}{dA_L}, \quad \frac{d\lambda_{12}}{dt} = -\frac{d\mathbb{H}}{dA_I}.$$

Thus, the above equalities gives the system (7) of first order ordinary derivative of adjoint variables $\lambda_i$, where $i = 1, 2, 3, \ldots, 12$.

Next, by optimality conditions, we get
$$\frac{d\mathbb{H}}{du_1}\Big|_{u_1=u_1^*} = 0 \quad \frac{d\mathbb{H}}{du_2}\Big|_{u_2=u_2^*} = 0$$
$$\frac{d\mathbb{H}}{du_3}\Big|_{u_3=u_3^*} = 0 \quad \frac{d\mathbb{H}}{du_4}\Big|_{u_4=u_4^*} = 0.$$

Therefore, $u_1^* = \frac{\beta_1 S(\lambda_2-\lambda_1)(I+H_I+A_I)}{c_1 N}$, $u_2^* = \frac{\beta_2 S(\lambda_5-\lambda_1)(H+H_E+H_I+H_L+\eta(A+A_I))}{c_2 N}$, $u_3^* = \frac{(\lambda_7-\lambda_9)\theta_2 H_E}{c_3}$, and

$u_4^* = \frac{(\lambda_5-\lambda_{10})\omega_1 H + (\lambda_6-\lambda_{10})\omega_2 A + (\lambda_7-\lambda_{10})\omega_3 H_E + (\lambda_8-\lambda_{10})\omega_4(1-\phi)H_I + (\lambda_9-\lambda_{10})\omega_5 H_L + (\lambda_{11}-\lambda_{10})\omega_6 A_L + (\lambda_{12}-\lambda_{10})\omega_7 A_I}{c_4}$.

These results can be expressed in $U$ as:

$$u_1(t) = \begin{cases} 0, & \text{if} \quad u_1^* \le 0 \\ u_1^*, & \text{if} \quad 0 < u_1^* < 1 \\ 1, & \text{if} \quad u_1^* \ge 1 \end{cases}, \quad u_2(t) = \begin{cases} 0, & \text{if} \quad u_2^* \le 0 \\ u_2^*, & \text{if} \quad 0 < u_2^* < 1 \\ 1, & \text{if} \quad u_2^* \ge 1 \end{cases},$$

$$u_3(t) = \begin{cases} 0, & \text{if} \quad u_3^* \le 0 \\ u_3^*, & \text{if} \quad 0 < u_3^* < 1 \\ 1, & \text{if} \quad u_3^* \ge 1 \end{cases}, \quad u_4(t) = \begin{cases} 0, & \text{if} \quad u_4^* \le 0 \\ u_4^*, & \text{if} \quad 0 < u_4^* < 1 \\ 1, & \text{if} \quad u_2^* \ge 1 \end{cases}$$

## 5 Numerical simulations

Till now, the TB-HIV co-infection model with or without optimal control is analysed analytically. In this portion, we discussed the numerical results to confirm our analytical findings. The simulation gives a clear image about the involvement of control functions on the disease transmission dynamics. We proposed two or more intervention strategies at a time to minimize both the disease and the cost burden.

We collected the HIV-TB co-infection confirmed cases in Ethiopia from July 2021-2023 as shown in Fig 3. We estimated the initial values of each state variables and the parameter's

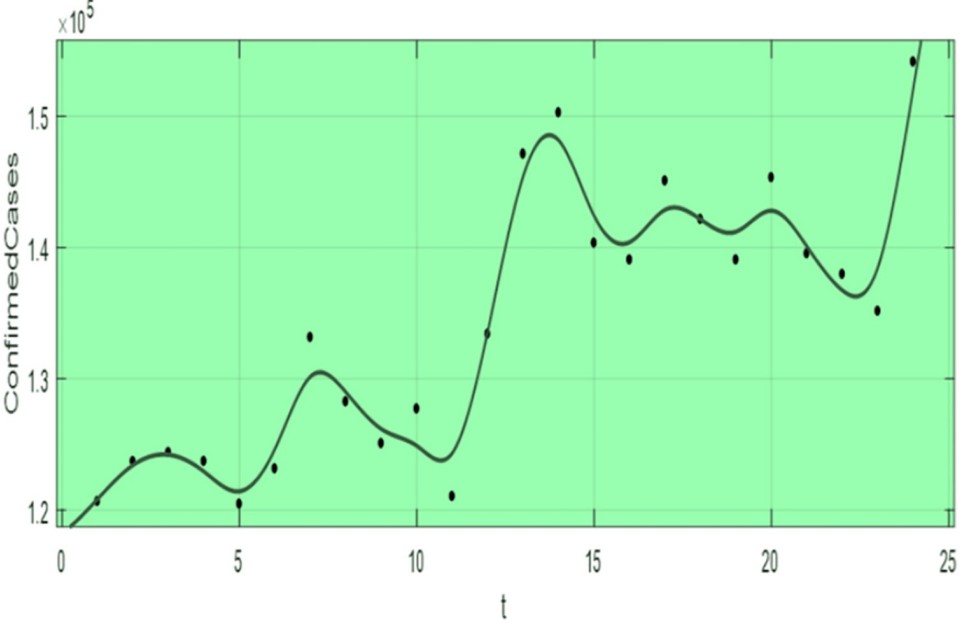

**Fig 3. HIV-TB co-infection confirmed cases versus time t from July 2021-2023.**

value, where the model solutions that are best fit to the real data reported. The populations in classes $E = 1.19 \times 10^6$, $I = 3.73 \times 10^5$, and $L = 2.18 \times 10^7$, collected from National TB and Leprosy strategic plan in Ethiopia [27]. Again, we estimated $H = 890\,311$, $A = 305\,770$, $H_E = 213\,451$, $H_I = 250\,853$, $H_L = 290\,008$, $T = 1\,253\,420$, $A_L = 207\,457$, $A_I = 120\,598$ from federal health ministry of Ethiopia, CDC, and WHO annual report [7, 21, 28].

The susceptible people is obtained by $S = N − (E + I + L + H + A + H_E + H_I + T + H_L + A_L + A_I)$ where N = 102 468 037, then $S = 75\,573\,169$. The recruitment people who are entered to class $S$ is calculated by $\pi = b \times N$, where the birth rate $b = 30.97\,/1\,000$. Hence, $\pi = 3\,173\,435.1$.

To estimate the rest constant parameters in the model (2), we formulated the model as:

$$z' = f(t, z, \theta), z(t_0) = z_0. \tag{60}$$

Here, $z$ is the state variable and $\theta$ is the parameter value to be determined.
Define a least squares objective function:

$$S(\theta) = \sum_{i=1}^{Ds} (z(i) − \bar{z}(i))^2, \tag{61}$$

where $z(i)$ is the solution of (60), $\bar{z}(i)$ is the real data, and $Ds$ is the data sample size. We get the optimum parameter values by minimizing the objective function:

$$\begin{cases} \min_{\theta} S(\theta) \\ \text{Subject to} \quad z' = f(t, z, \theta), z(t_0) = z_0. \end{cases} \tag{62}$$

The algorithm is presented below:

**Algorithm 1:**

**Step 1.** *Guess initial parameter values $a_0$. Set $a = a_0$.*

**Step 2.** *Using MATLAB version 2013a ode45 routine, solve Eq (60) using a to find the solution $z(i)$.*

**Step 3.** *Evaluate error using Eq (61).*

**Step 4.** *Use a to minimize Eq (62) using an optimization algorithm nlinfit to find the parameters $\hat{a}$ with 95% confidence interval. Update $a = \hat{a}$.*

**Step 5.** *Check for the convergence. If the convergence is not satisfied go to **Step 2**.*

**Step 6.** *On convergence, set $a = \hat{a}$.*

Using the above algorithm the values of the parameters are estimated and presented in Table 2. However, due to lack of real data, the values of some parameters are taken from other related literatures. The time duration of the study is $t_f = 10$ years.

In addition, we assessed the value of the coefficient parameters ($b_1 = 0.65$, $b_2 = 0.55$, $b_3 = 0.28$, $b_4 = 1$, and $b_5 = 1.72$) depend on the way of constants found in [18]. Furthermore, we assumed that the value of weight constants based on the importance level of one intervention over the other. These are $c_1 = 10^4$, $c_2 = 10^4$, $c_3 = 2 \times 10^4$, and $c_4 = 2 \times 10^4$. Some data may also taken as just for numerical purpose. Nevertheless, obtaining sufficient data about the vital elements of the TB-HIV/AIDS co-infection model was one big challenge of the study.

We used MATLAB software to validate the analytical results. Here, we discussed the features of the state trajectories with or without optimal control. The control profiles of each strategy are also plotted. We proposed four strategies based on the suggested intervention approaches. They designed as combination of two or more strategies at a time. However, only one intervention at a time is not an effective method [73, 77]. Thus, we have seen these

**Table 2. Symbols and values of the parameters.**

| Parameters | Values | References | Parameters | Values | References |
|---|---|---|---|---|---|
| $\pi$ | $3.1734 \times 10^6$ | Fitted | $\beta_2$ | 0.18 | [73] |
| $\beta_1$ | 0.00151 | [49] | $\phi$ | 0.701 | Assumed |
| $\mu$ | 0.0058 | Fitted | $\sigma$ | 0.0013 | [61] |
| $\theta$ | 0.3 | Fitted | $\gamma$ | 0.546 | [61] |
| $\epsilon_1$ | 0.004 | [27] | $\epsilon_2$ | 0.001 | Fitted |
| $\epsilon$ | 0.5 | Fitted | $d_2$ | 0.016 | Fitted |
| $p$ | 0.168 | [49] | $\delta$ | 0.62 | Fitted |
| $d_1$ | 0.0003 | [27] | $d_3$ | 0.002 | Fitted |
| $\alpha$ | 0.153 | Fitted | $k$ | 0.023 | Fitted |
| $\omega$ | 1.17 | [74] | $\sigma_1$ | 0.015 | Fitted |
| $\delta_1$ | 0.03 | Fitted | $\theta_1$ | 0.0026 | Fitted |
| $\psi$ | 0.336 | Assumed | $\theta_2$ | 0.153 | Fitted |
| $\psi_1$ | 0.88 | [75] | $\varphi$ | 0.08 | Assumed |
| $\tau$ | 0.6 | Fitted | $\theta_3$ | 0.1 | Fitted |
| $\eta$ | 1.05 | [76] | $\omega_4, \omega_7$ | 0.4 | Fitted |
| $\omega_3$ | 0.302 | Fitted | $\omega_i, i = 1, 2, 5, 6$ | 0.2 | Fitted |

strategies can highly minimize TB-HIV co-infection disease in the country Ethiopia and the elaborations are continued in the next part.

## 5.1 Preventive effort of TB disease and treatment of HIV/AIDS disease

We used prevention of TB disease combined with HIV/AIDS treatment as an optimal strategy (i.e.$u_i \neq 0$, for $i = 1, 4$, whereas $u_2 = 0$, and $u_3 = 0$). The plot $(A − E)$ of Figs 4–6 illustrates the impact of this optimal approach on HIV/AIDS-TB co-infected individuals. This strategy can

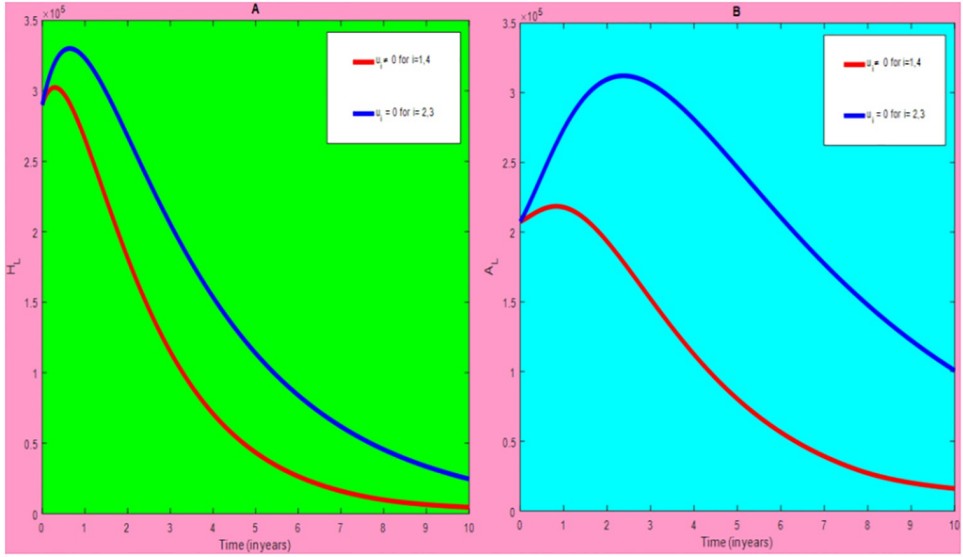

**Fig 4. Infected individuals in $H_L$ and $A_L$ class when applying combined efforts of prevention of TB and treatment of HIV/AIDS optimally.**

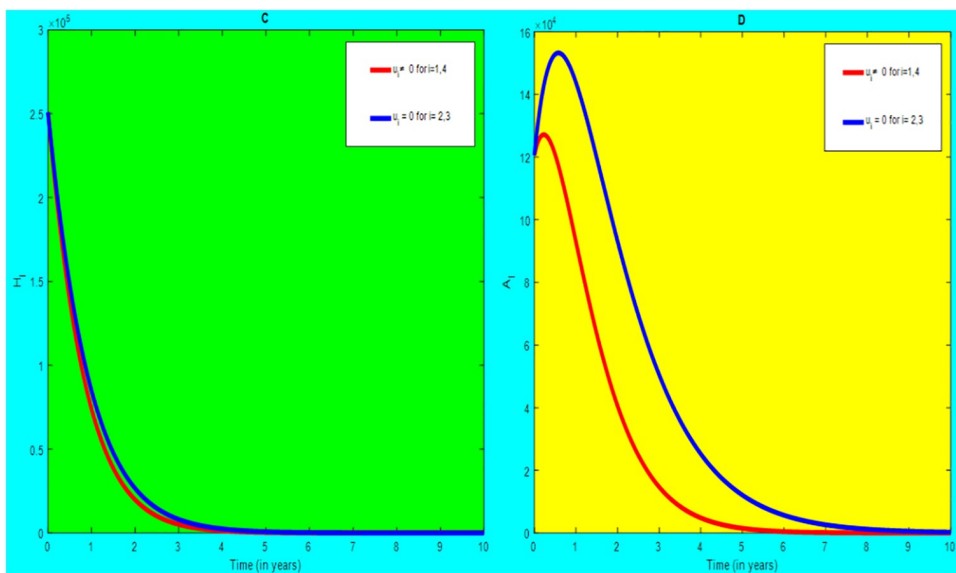

**Fig 5. Infected individuals in $H_I$ and $A_I$ class when applying combined efforts of prevention of TB and treatment of HIV/AIDS optimally.**

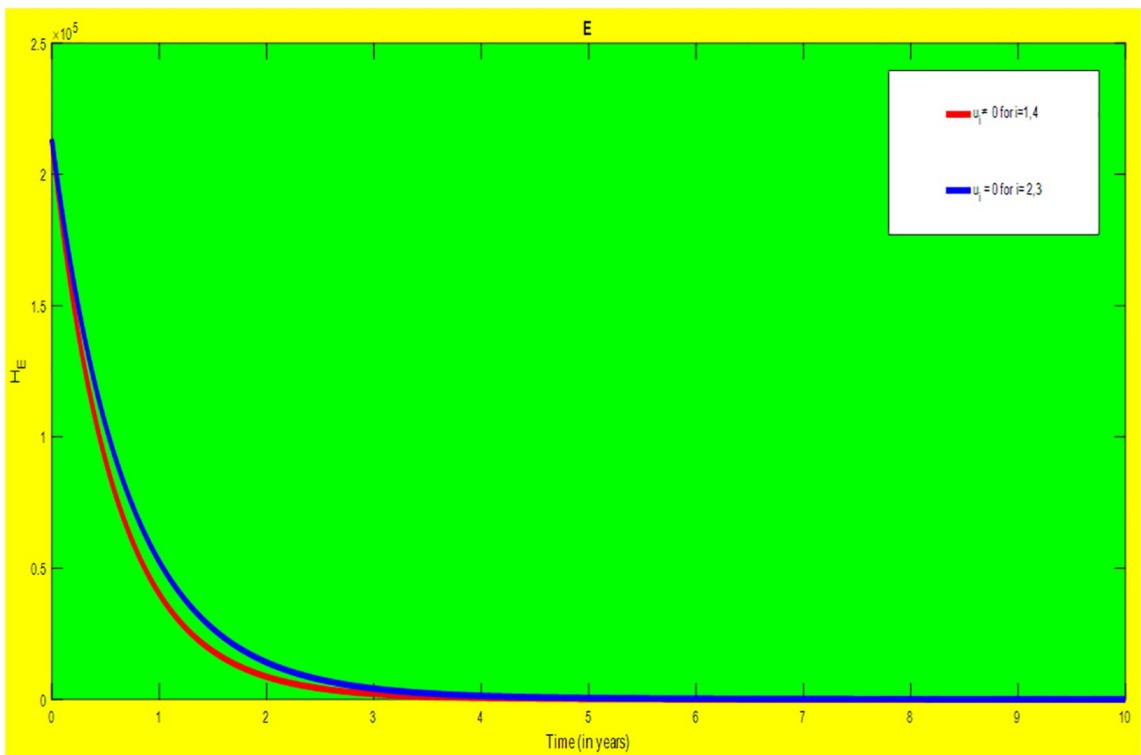

**Fig 6. Infected individuals in $H_E$ class when applying combined efforts of prevention of TB and treatment of HIV/AIDS optimally.**

be used to decrease the number of low risk latent TB individuals co-infected by HIV with pre-AIDS and AIDS symptoms dramatically rather than without optimal approach.

As shown in the Fig 4A and 4B, If optimal control is not applied, the co-infected individual in $H_L$ and $A_L$ is increased at the beginning of the year and reach at a peak value at $3.2999 \times 10^5$ and $3.1217 \times 10^5$ respectively. Hereafter, the disease burden decreased significantly. However, in case of optimal control the prevention effort has an impact on individuals under $H_L$ class by reducing the high risk latent individuals. In addition, the more co-infected people are moved to the treated class due to HIV treatment.

In Fig 5C, when no optimal control is used the co-infectious people in $H_I$ class is decreased due to the influence of constant treatment rate $w_4$ for HIV and successful TB treatment rate of $1 - \psi$. Conversely, in Fig 5D, the co-infectious people in $A_I$ class are increased at the beginning of the years and reach at a peak value $1.5322 \times 10^4$. Hereafter, the disease burden decreased significantly. This is because more TB infected people co-infected with HIV are completed their TB treatment at a constant rate $\tau$ and with constant HIV treatment rate $w_7$. However, the time-based optimal approach seems negligible in the $H_I$ compartment but later on we can observe its impact. This combination strategy has a high effect on the co-infected people by active TB and HIV with AIDS symptoms. For that reason, the prevention effort can minimize the susceptible individuals to become TB infected under high-risk latent stage progress to active stage. Additionally, the more co-infectious people are joined to treated class due to HIV treatment.

In Fig 6E, when optimal control is not applied, the co-infected people in $H_E$ class is decreased due to the influence of constant treatment rate $w_3$ for HIV and treatment rate of high risk latent TB $\theta_2$. Nevertheless, when optimal control is used the disease burden is decreased rather than without optimal control. The impact of this strategy is visible around five years but seems negligible after a while.

Therefore, this strategy can minimize/eradicate the HIV-TB co-infection disease burden in the country, Ethiopia.

## 5.2 Preventive effort of HIV/AIDS disease and case finding TB

We used prevention of HIV/AIDS disease combined with case finding effort of TB as an alternative mechanism (i.e. $u_i \neq 0$, for $i = 2, 3$, whereas $u_1 = 0$, and $u_4 = 0$). The plot $(A - E)$ of Figs 7–9 illustrates the impact of this optimal approach on HIV/AIDS-TB co-infected individuals.

As shown in the Fig 7A and 7B, if no optimal control is applied, the co-infected individual in $H_L$ and $A_L$ are increased at the beginning of the year. The disease burden decreased dramatically after a while. Nevertheless, in case of optimal control, the co-infected individuals in these two sub-classes are not more decreased as compared from the first strategy. Since, there are more HIV-infected people who have recovered from TB but remain low-risk latent due to case-finding effort. Hence, the co-infected populations in the $H_L$ class are increased and reach a peak value $3.6876 \times 10^5$. Conversely, the optimal strategy seems negligible for the first around 1 year in $A_L$ class but it gets a significant influence far ahead.

In Fig 8C, the impact of optimal control strategy seems negligible in the $H_I$ class but it has a visible impact to some extent. While, In Fig 8D, the co-infected people in $A_I$ compartment are raised for the first few months and reach a high value $1.5322 \times 10^5$. After this, the number of individuals in the class $A_I$ decreased radically.

In Fig 9E, the co-infected people in $H_E$ class are decreased when optimal control is applied. In this compartment, the influence of this strategy is better than the first approach. Since, the number of HIV people co-infected with TB at a high-risk latent stage is more decreased due to case-finding effort.

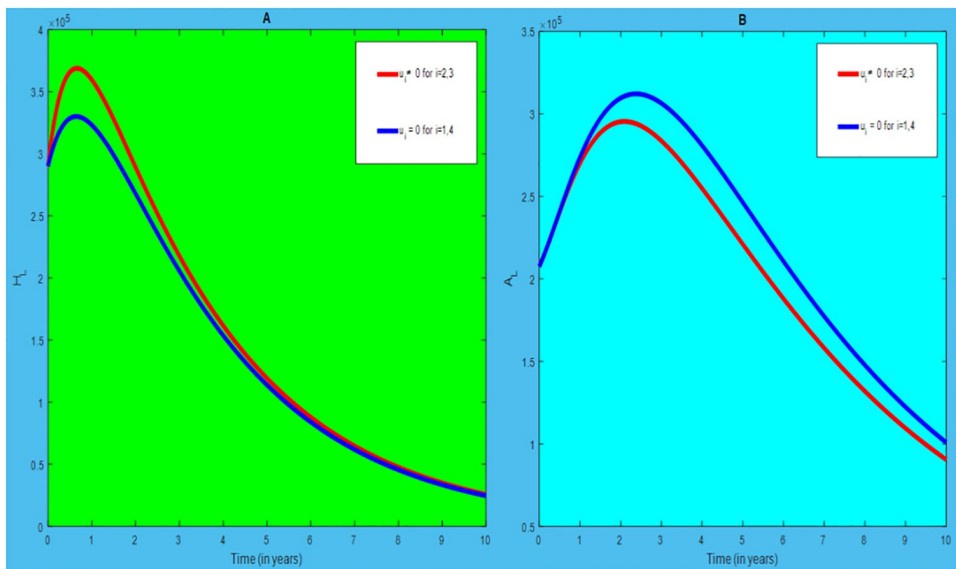

**Fig 7. Infected individuals in $H_L$ and $A_L$ class when applying combined efforts of prevention of HIV/AIDS and case finding TB optimally.**

Hence, this strategy can minimize/eradicate the HIV-TB co-infection disease burden.

## 5.3 Case finding for TB disease and HIV treatment

We used case finding for TB disease combined with HIV/AIDS treatment as an alternative measure (i.e. $u_i \neq 0$, for $i = 3, 4$, whereas $u_1 = 0$, and $u_2 = 0$). The plot $(A - E)$ of Figs 10–12 illustrates the impact of this optimal approach on HIV/AIDS-TB co-infected individuals. All plots showed that the model with and without optimal control plays a great role to minimize the disease burden.

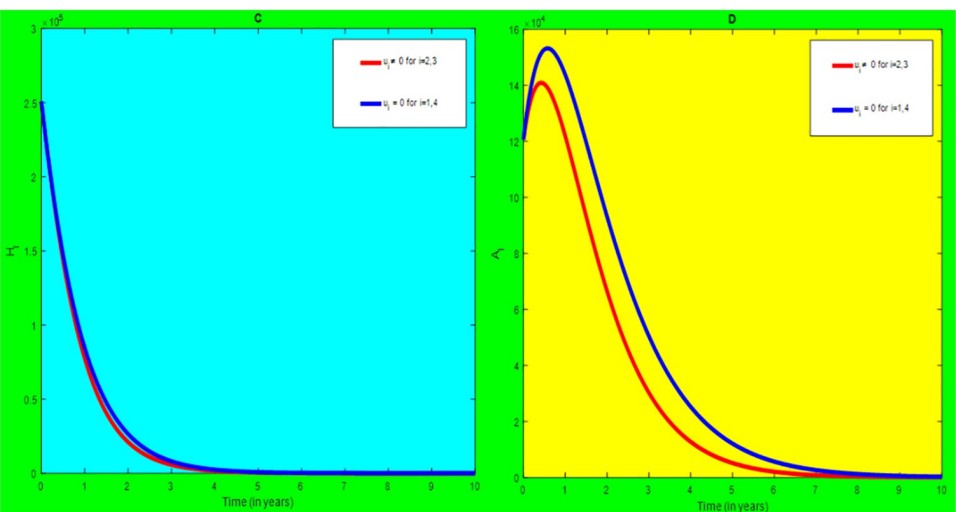

**Fig 8. Infected individuals in $H_I$ and $A_I$ class when applying combined efforts of prevention of HIV/AIDS and case finding TB optimally.**

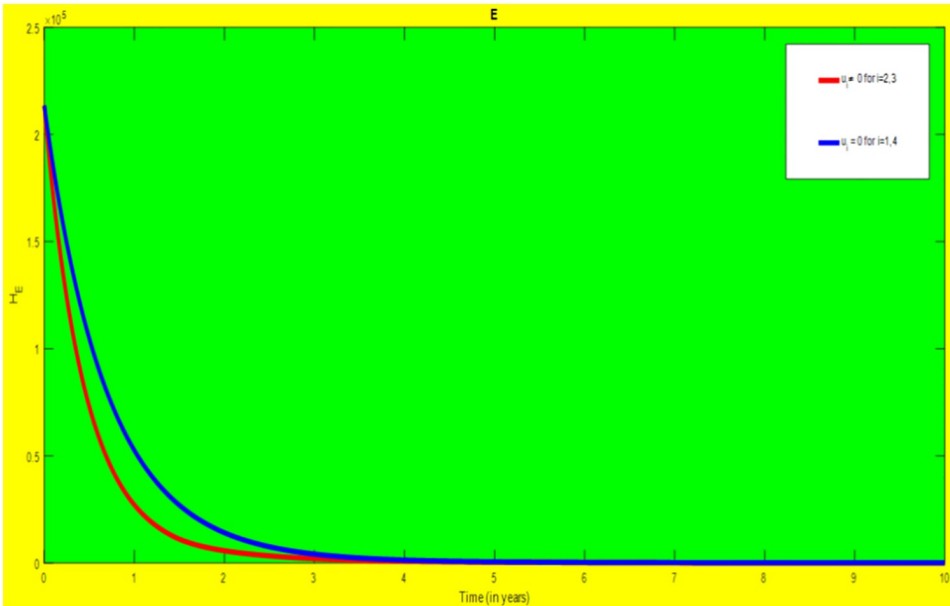

**Fig 9. Infected individuals in $H_E$ class when applying combined efforts of prevention of HIV/AIDS and case finding TB optimally.**

In the Fig 10A, the numerical results displayed that this strategy is better than the second strategy due to HIV treatment effort. The disease burden seems raised at the first of a few months but later, it decreased intensely. The optimal strategy is also a more effective approach on the co-infected class $A_I$ as shown in the Fig 10B. The graphical result shows the combination optimal approach can reduce the disease burden effectively.

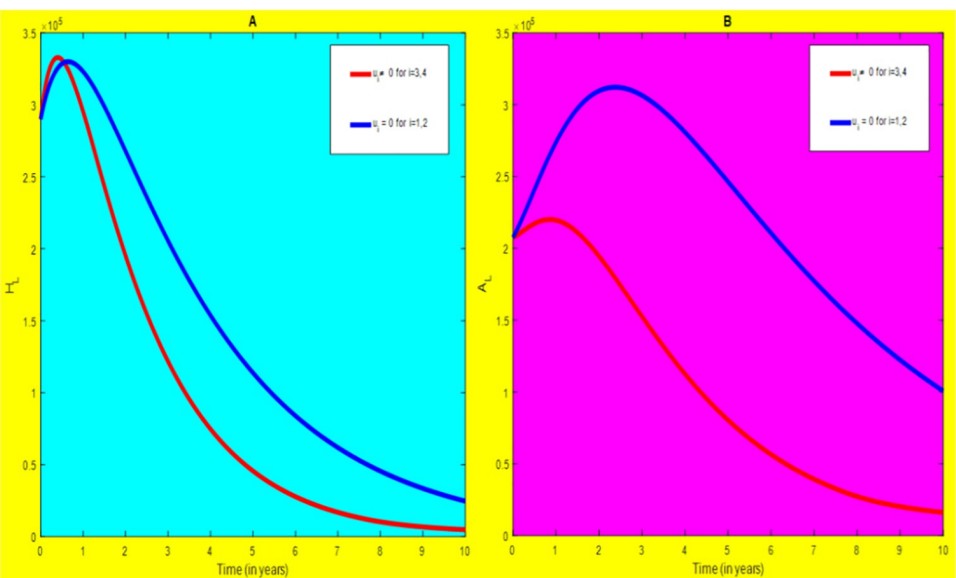

**Fig 10. Infected individuals in $H_L$ and $A_L$ class when applying combined efforts of case finding TB and HIV treatment optimally.**

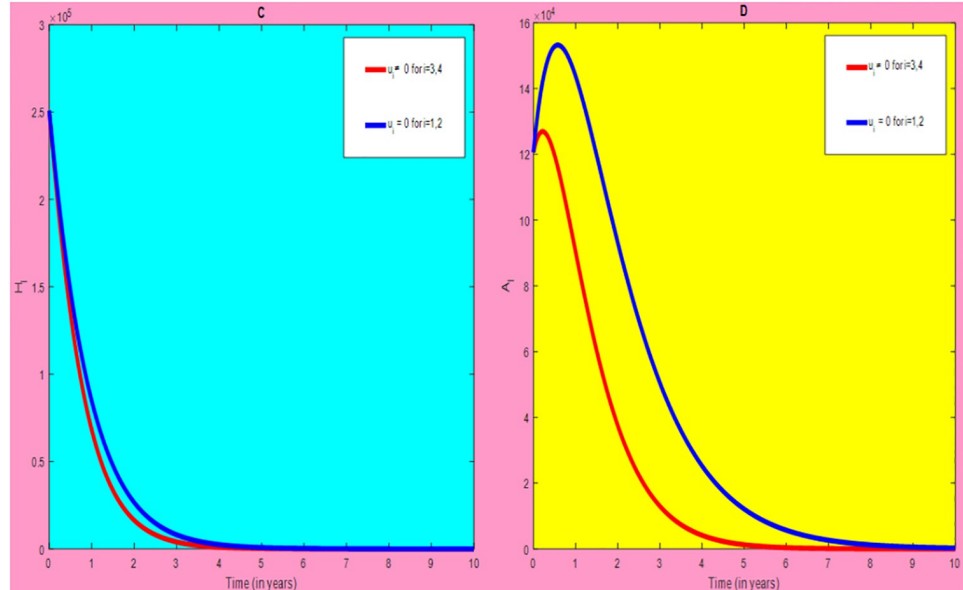

**Fig 11. Infected individuals in $H_I$ and $A_I$ class when applying combined efforts of case finding TB and HIV treatment optimally.**

In Fig 11C, one can observe that this combination optimal control has enhanced the impact on the co-infected class $H_I$ rather than the second strategy. The same effect is as shown in the sub-population $A_I$ Fig 11D. Since, there are more co-infected people are moved to the treated class due to HIV treatment effort.

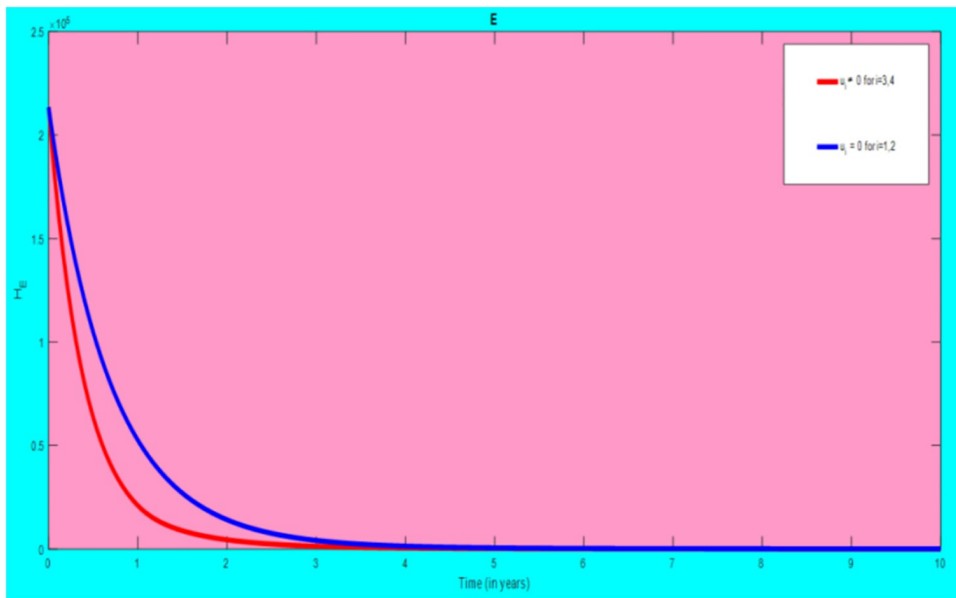

**Fig 12. Infected individuals in $H_E$ class when applying combined efforts of case finding TB and HIV treatment optimally.**

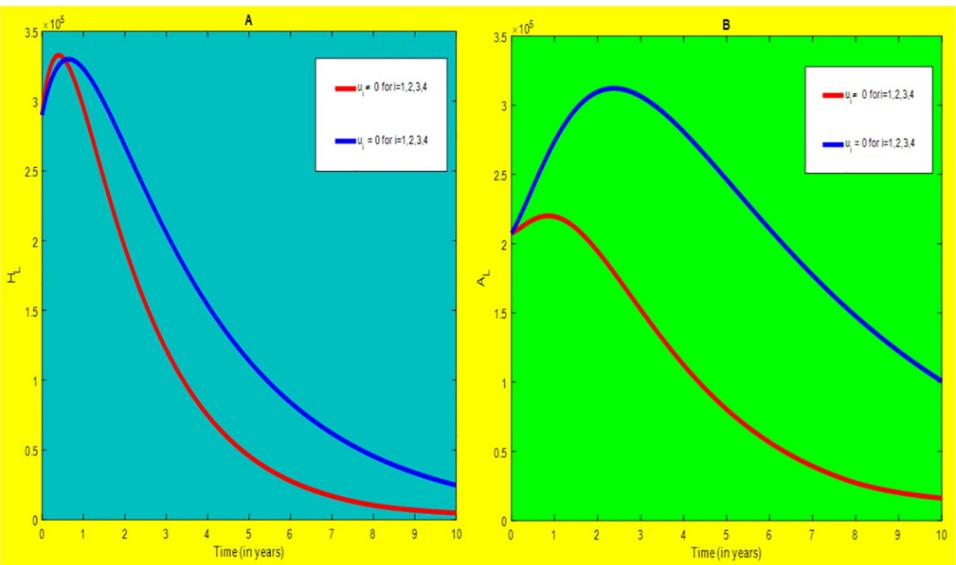

**Fig 13. Infected individuals in $H_L$ and $A_L$ class when applying all strategies optimally.**

In Fig 12E, the number of co-infected people in $H_E$ class are reduced when optimal control is used. In this class, the effect of this strategy is also better than the first approach.

Thus, the optimal control of case ending effort of TB and HIV/AIDS treatment has great impact to reduce the disease burden.

## 5.4 Using all the intervention efforts

We used all intervention efforts optimally as an alternative mechanism (i.e. $u_i \neq 0$, for $i = 1, 2, 3, 4$). The plot $(A - E)$ of Figs 13–15 displays the effect of this approach on HIV/AIDS TB co-

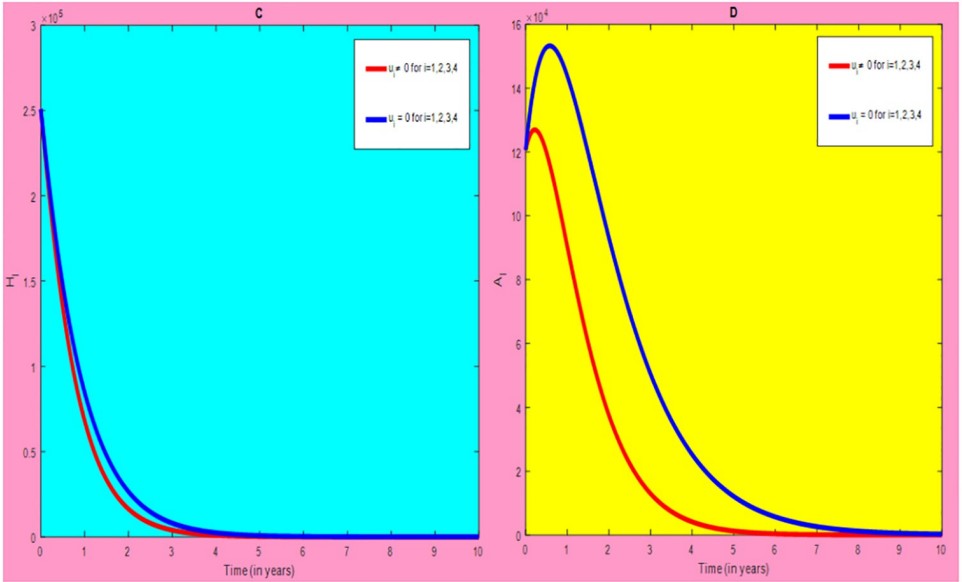

**Fig 14. Infected individuals in $H_I$ and $A_I$ class when applying all strategies optimally.**

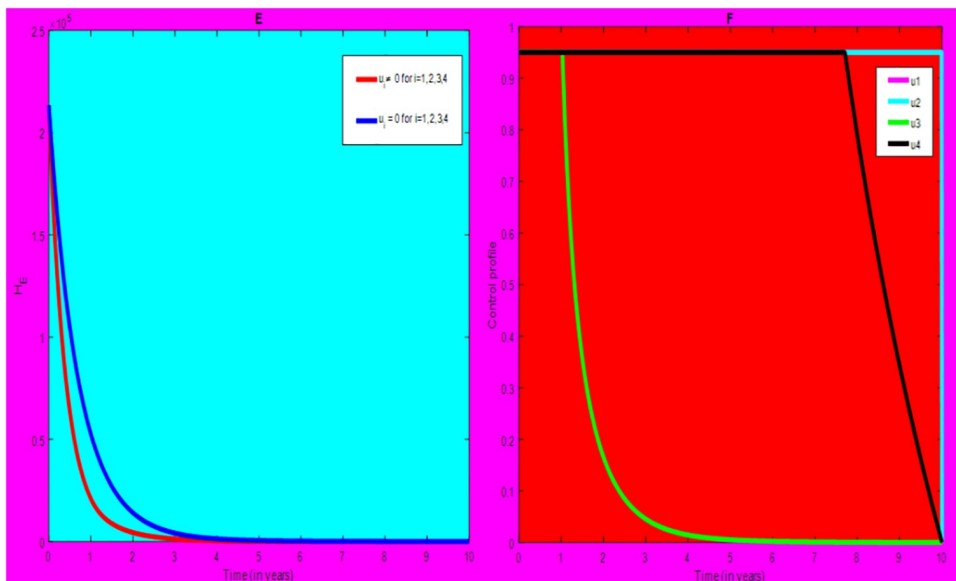

**Fig 15. Infected individuals in $H_E$ class when applying all strategies optimally and control profiles.**

infected people. All graphical results show that the model with and without optimal control plays a great role to minimize the disease burden. The effect of this mechanism on co-infected individuals seems like to the third strategy, but the visible differences appeared on the amount of cost for implementation. This will be discussed in the cost-effectiveness section, however the trajectories are plotted in Fig 16.

The control profiles that generate this simulation result are as shown in the Fig 15F. The pink color of the trajectory $u_1$ is concealed by the cyan color of the trajectory $u_2$. The control plots $u_1$ and $u_2$ have displayed the maximum efforts required for the entire duration. The

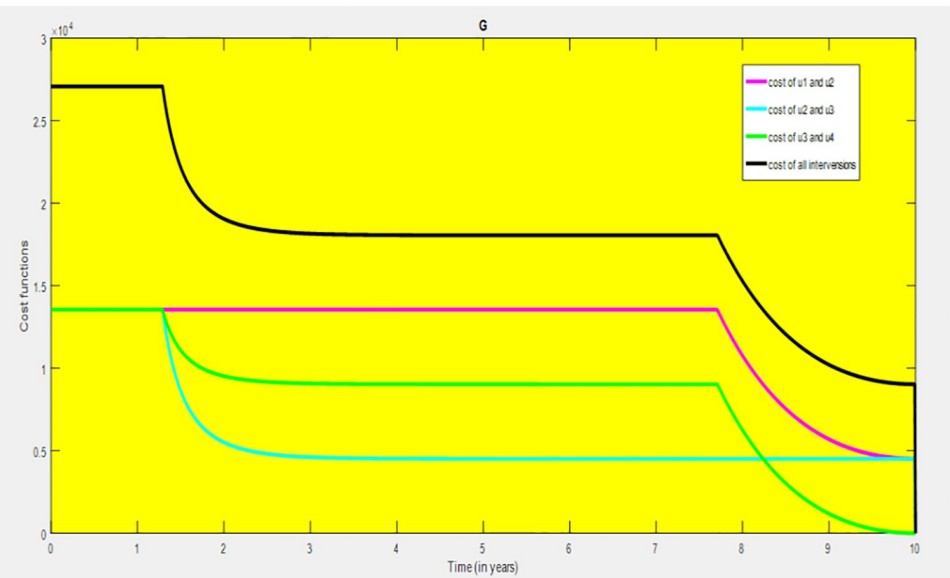

**Fig 16. Trajectories of cost functions.**

**Table 3. Total infected averted (increasing order) and total cost.**

| Plans | Description | Total infected averted | Total cost (USD) |
|---|---|---|---|
| B | Preventive of HIV & case finding TB | $9.661 \times 10^5$ | $6.0870 \times 10^6$ |
| C | Case finding TB & HIV treatment | $1.0603 \times 10^6$ | $8.6911 \times 10^6$ |
| D | All interventions | $1.0613 \times 10^6$ | $1.7715 \times 10^7$ |
| A | Preventive of TB & HIV treatment | $1.0616 \times 10^6$ | $1.2038 \times 10^7$ |

control plot $u_3$ shows that the high case-finding for TB is needed for the first around 1 year and extremely reduced later. The control plot $u_4$ shows that high treatment of HIV is desired for around 7.8 years and reduced after a while. Finally, all controls are dropped to zero due to the proposed strategies being expected to be over at the end of the time forecast.

## 6 Cost-effectiveness analysis

Here, we presented the cost-effectiveness rank of one implemented strategy over the other. We achieved this by (Baba and Makinde [78]); they had declared that

Incremental Cost-Effectiveness Ratio (ICER) $= \frac{\text{Difference in costs between strategies}}{\text{Difference in health effects between strategies}}$.

We got the total number of infected averted which is the difference between the total infectious people with and without control. We applied this technique by ranked increasing order of effectiveness with respect to the infected averted. Besides to this, the total cost is also mentioned in Table 3.

We compared the strategy of B and C by computing the ICER:

$$ICER(B) = \frac{6.0870 \times 10^6}{9.661 \times 10^5} = 6.3$$

and

$$\text{ICER(C)} = \frac{8.6911 \times 10^6 - 6.0870 \times 10^6}{1.0603 \times 10^6 - 9.661 \times 10^5} = 27.64.$$

The comparison displayed that $ICER(C) > ICER(B)$, which shows that strategy C is strongly dominated and does not consume limited resource.

Hence, we should remove strategy C from the set of choices.

Next, we compare strategy B and D.

Already we calculated $ICER(B) = 6.3$ $ICER(D) = \frac{1.7715 \times 10^7 - 6.0870 \times 10^6}{1.0613 \times 10^6 - 9.661 \times 10^5} = 122.14$.

The comparison showed that strategy D is more costly and less effective than strategy B. Hence, we should remove strategy D from the set of choices.

Finally, we compared strategy B and A.

Now, $ICER(A) = \frac{1.2038 \times 10^7 - 6.0870 \times 10^6}{1.0616 \times 10^6 - 9.661 \times 10^5} = 62.314$.

This shows that, we should remove strategy A from the set of options.

Therefore, strategy B is the most cost-effective strategy rather than the rest alternatives.

## 7 Conclusion

The co-epidemic of the two diseases HIV/AIDS and TB is serious in the country Ethiopia as one disease accelerating the rate of infection of the other and vice versa. The disease burden leads to the country having health and economic challenges. Thus, we expect that a lot of new research findings would be incorporated into the body of knowledge. This study addressed the concept of disease transmission dynamics and further controlling efforts.

Hence, we developed a new mathematical model for HIV-TB co-infection transmission dynamics. The model has 12 state variables, which is highly non-linear and it was challenged for qualitative analysis. We discussed the points such as positive invariance of the solution set, equilibria points, basic reproduction number ($R_0$), stability of equilibria points, and probability of bifurcation. We found that the DFE point is locally asymptotically stable when $R_0 < 1$ and unstable when $R_0 > 1$, whereas the EE point is locally asymptotically stable when $R_0 > 1$. At the threshold value ($R_0 = 1$), the backward bifurcation is occurred. We also developed an optimal control model, which is an extension of the original one via incorporated prevention effort of both diseases, case finding effort of TB disease and HIV treatment. These interventions through a co-epidemic model which is build on high risk and low risk latent TB cases those are not fully recovered is a palpable gap in previous works. Hence, the existence of optimal control solutions as well as their characterization is presented. We used the distinguished Pontryagin's Maximum Principle (PMP) which states an essential condition for optimal solutions. Using the MATLAB software, we applied the Runge-Kutta method of order four (RK4-method) to validate the analytical results of the optimal control model. The numerical results for the model without optimal control illustrated that decreasing the contact rate of TB and increasing HIV treatment rate of co-infected people have high contribution to suppress the disease transmission. Whereas, the simulation result of the model with optimal controls confirmed that the combination of two or more strategies at a time can effectively minimize co-infected individuals under $H_E$, $H_I$, $H_L$, $A_L$, $A_I$ classes. Likewise, these approaches can also decrease the cost incurred during interventions. We have seen that all strategies have certain limitations; however, the couple of preventive effort of HIV/AIDS and case finding for TB disease at a time is the best cost-effective option. Those model results are scrutinized using the real data collected from Ethiopia, which is the novelty of this work and makes it different from other approaches in the literature. As upcoming work, the model can be developed via considering vertical transmission of HIV/AIDS transmission dynamics. The model can also extend into a Tri-epidemics model when the global pandemic COVID-19 will be considered; and a fractional order model to explore the memory effects of the biological systems are suggested for further investigation. Moreover, this study will aid in the fight against tuberculosis, HIV/AIDS, and their co-infection policy makers and other concerned organizations. Finally, our study recommends further attention or emphasis should be given to mathematical modelling for exploring transferable diseases.

## Author Contributions

**Conceptualization:** Tigabu Kasie Ayele, Emile Franc Doungmo Goufo.

**Data curation:** Tigabu Kasie Ayele.

**Formal analysis:** Tigabu Kasie Ayele, Emile Franc Doungmo Goufo.

**Investigation:** Tigabu Kasie Ayele, Emile Franc Doungmo Goufo, Stella Mugisha.

**Methodology:** Tigabu Kasie Ayele.

**Software:** Tigabu Kasie Ayele.

**Supervision:** Emile Franc Doungmo Goufo, Stella Mugisha.

**Validation:** Tigabu Kasie Ayele, Emile Franc Doungmo Goufo.

**Visualization:** Tigabu Kasie Ayele.

**Writing – original draft:** Tigabu Kasie Ayele.

**Writing – review & editing:** Tigabu Kasie Ayele, Emile Franc Doungmo Goufo.

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
