## [Decision Letter · Decision Letter 0]

14 Aug 2024

PONE-D-24-26378Co-infection mathematical model for HIV/AIDS and tuberculosis with optimal control in EthiopiaPLOS ONE

Dear Dr. Ayele,

Thank you for submitting your manuscript to PLOS ONE. After careful consideration, we feel that it has merit but does not fully meet PLOS ONE’s publication criteria as it currently stands. Therefore, we invite you to submit a revised version of the manuscript that addresses the points raised during the review process.

**ACADEMIC EDITOR:**The authors should address the remeaning reviewers' comments. **(1) **Each section should have its writing aligned.

We look forward to receiving your revised manuscript.

Kind regards,

Joshua Kiddy K. Asamoah, PhD

Academic Editor

PLOS ONE

4. Please update your submission to use the PLOS LaTeX template. The template and more information on our requirements for LaTeX submissions can be found at http://journals.plos.org/plosone/s/latex.

Reviewers' comments:

Reviewer's Responses to Questions

**Comments to the Author**

1. Is the manuscript technically sound, and do the data support the conclusions?

Reviewer #1: Partly

Reviewer #2: Yes

2. Has the statistical analysis been performed appropriately and rigorously? 

Reviewer #1: N/A

Reviewer #2: N/A

3. Have the authors made all data underlying the findings in their manuscript fully available?

Reviewer #1: No

Reviewer #2: No

4. Is the manuscript presented in an intelligible fashion and written in standard English?

Reviewer #1: No

Reviewer #2: Yes

5. Review Comments to the Author

Reviewer #1: The paper requires a major improvement in order to meet the standards of the journal. My comments are found below.

1. The abstract fails to justify the need for this research study. Also, the first statement in the Abstract is unnecessary. Remove it and reconstruct the abstract with necessary information. Be clear with the reason behind this work, how to go about it and the key results obtained.

2. The novelty of this research is not clearly stated. Please do that.

3. The authors claim that "However, none of these authors have studied the extension of model [68] with HIV/AIDS cohorts and incorporating optimal control efforts". The referred article [68] was extended in 2022 Zhong-Kai et. al. by where the authors studied the HIV/AIDS-TB coinfection model with optimal control by further considering age structure. Authors may find the article details here "Guo, Z. K., Huo, H. F., & Xiang, H. (2022). Optimal control of TB transmission based on an age structured HIV-TB co-infection model. Journal of the Franklin Institute, 359(9), 4116-4137".

4. The authors said also that due to lack of data they picked some parameter values from literature. How true is this? Are the authors claiming that there is no data on HIV/AIDS or TB or their coinfection?

5. In table 2, the authors have indicated that some parameters were fitted. Which year's population for Ethiopia was used for the estimation? Where from the value of N? What is the source of the data used for the estimation? Kindly consider coming out with an updated parameter estimation section.

6. Authors should note that table labels should be placed on top of the table.

7. The article has enormous grammatical errors. Please read through the entire work to correct them.

There are several graphs with no proper discussions on them. Please remove redundant graphs or rewrite the numerical discussions. Numerical simulation discussions should practical.

8. All model assumptions should be supported with literature.

9. Though I see a section on positivity and boundedness. Unfortunately, I did not see any work on positivity. Please provide that.

10. Authors should be clear with the bifurcation analysis. Does the model undergo a forward or backward bifurcation. Where is the bifurcation graph?

11. Under the control model, is early detection not more of a control mechanism than preventive?

12. Conclusion of the work should be updated. The current form is too long and does not highlight the essential aspects of the work. What are the gaps in the work?

13. The introduction and model formulation sections require improvement. Please do that.

Reviewer #2: I read the paper in details, the paper has a potential for publication provided the authors attends to the following suggestions

1. The abstract should contain answers to the following questions: Why is this particular study important? What are the important results from this study different from others? What conclusions can be drawn from the results in this study?

2. The literature review of epidemic models in this direction is too weak and must be improved meticulously, since there are many works in this direction.

3. Support important model assumptions with biological evidences. Also state if the model is a new one or a modification of an existing model.

4. What is the biological significance of the Basic reproduction number, as it regards to disease control? Also discuss Theorem 1 and why it important.

5. What are the epidemiological significance of theorems 1, 2? Be more detailed in discussing the many mathematical analyses. I advise the authors to check some recent papers with useful approach for analysis of stability analysis useful, such as:

a. Mathematical analysis of a novel fractional order vaccination model for Tuberculosis incorporating susceptible class with underlying ailment. International Journal of Modelling and Simulation

b.Mathematical Modelling of Tuberculosis Outbreak in an East African Country Incorporating Vaccination and Treatment. Computation 2023, 11, 143.

c. A mathematical analysis of the two-strain tuberculosis model dynamics with exogenous re-infection. Healthcare Analytics. 4 (2023) 100266

d. A mathematical model for the co-dynamics of COVID-19 and tuberculosis. Mathematics and Computers in Simulation. 27(2023), 499-520

e. . Mathematical model for control of tuberculosis epidemiology. Journal of Applied Mathematics and Computing, (2022)

f. Analysis and Dynamics of Tuberculosis Outbreak: A Mathematical Modelling Approach Advances in System Science and Applications 22(4), 144-161 (2022).

g. Analysis and Dynamics of Tuberculosis Outbreak: A Mathematical Modelling Approach Advances in System Science and Applications 22(4), 144-161 (2022).

and others where detailed discussions have been done for mathematical analysis in epidemiological models. Try and connect your results more to health, in accordance with the scope of the journal.

The Conclusion should be greatly improved and must capture important findings from the study.

Provide detailed future research directions in the conclusion.

The entire manuscript should be thoroughly checked for typos and also the typesetting should be greatly improved to attract wider readership. Make the paper more interesting to readers

6. PLOS authors have the option to publish the peer review history of their article (what does this mean?). If published, this will include your full peer review and any attached files.

Reviewer #1: No

Reviewer #2: **Yes: **Olumuyiwa James Peter

---

## [Author Response · Author response to Decision Letter 0]

27 Sep 2024

We sincerely thank the editor and reviewers for taking the time to review our manuscript and providing constructive feedback to improve our manuscript. We have revised the manuscript accordingly by following the reviewers’ suggestion. The letter response to reviewers is attached in file section. The revised manuscript is attached next to the old manuscript is uploaded.

---

## [Decision Letter · Decision Letter 1]

9 Oct 2024

Co-infection mathematical model for HIV/AIDS and tuberculosis with optimal control in Ethiopia

PONE-D-24-26378R1

Dear Dr. Ayele,

We’re pleased to inform you that your manuscript has been judged scientifically suitable for publication and will be formally accepted for publication once it meets all outstanding technical requirements.

Kind regards,

Joshua Kiddy K. Asamoah, PhD

Academic Editor

PLOS ONE

Additional Editor Comments (optional):

Reviewers' comments:

Reviewer's Responses to Questions

**Comments to the Author**

1. If the authors have adequately addressed your comments raised in a previous round of review and you feel that this manuscript is now acceptable for publication, you may indicate that here to bypass the “Comments to the Author” section, enter your conflict of interest statement in the “Confidential to Editor” section, and submit your "Accept" recommendation.

Reviewer #1: All comments have been addressed

Reviewer #2: (No Response)

2. Is the manuscript technically sound, and do the data support the conclusions?

Reviewer #1: Yes

Reviewer #2: (No Response)

3. Has the statistical analysis been performed appropriately and rigorously? 

Reviewer #1: Yes

Reviewer #2: (No Response)

4. Have the authors made all data underlying the findings in their manuscript fully available?

Reviewer #1: Yes

Reviewer #2: (No Response)

5. Is the manuscript presented in an intelligible fashion and written in standard English?

Reviewer #1: Yes

Reviewer #2: (No Response)

6. Review Comments to the Author

Reviewer #1: I have carefully read through the responses and corrections, and I think the authors have accurately responded to my comments. The paper is improved in its current form. Thank you.

Reviewer #2: The authors have addressed all the comments satisfactorily, the manuscript can be accepted in the current form

7. PLOS authors have the option to publish the peer review history of their article (what does this mean?). If published, this will include your full peer review and any attached files.

Reviewer #1: No

Reviewer #2: **Yes: **Olumuyiwa James Peter

---

## [Editor Report · Acceptance letter]

22 Oct 2024

PONE-D-24-26378R1 

PLOS ONE

Dear Dr. Ayele, 

I'm pleased to inform you that your manuscript has been deemed suitable for publication in PLOS ONE. Congratulations! Your manuscript is now being handed over to our production team.

Kind regards, 

on behalf of

Dr. Joshua Kiddy K. Asamoah 

Academic Editor

PLOS ONE